# A stochastic event-based approach for flood estimation in catchments with mixed rainfall/snowmelt flood regimes

Valeriya Filipova[1], Deborah Lawrence[2], and Thomas Skaugen[2]

[1]University of Southeast Norway,INHM, Gullbringvegen 36, 3800 Bø, Norway, e-mail: valeriya.filipova@usn.no
[2]Norwegian Water Resources and Energy Directorate,P.O. Box 5091 Maj., N-0301 Oslo, Norway, e-mail: dela@nve.no

**Correspondence:** Valeriya Filipova (valeriya.filipova@usn.no)

**Abstract.** The estimation of extreme floods is associated with high uncertainty, in part due to the limited length of streamflow records. Traditionally, statistical flood frequency analysis and an event-based model (PQRUT) using a single design storm have been applied in Norway. We here propose a stochastic PQRUT model, as an extension of the standard application of the event-based PQRUT model, by considering different combinations of initial conditions, rainfall and snowmelt, from which a distribution of flood peaks can be constructed. The stochastic PQRUT was applied for 20 small and medium-sized catchments in Norway and the results give good fits to observed peak-over-threshold series. A sensitivity analysis of the method indicates that the soil saturation level is less important than the rainfall input and the parameters of the PQRUT model for flood peaks with return periods higher than 100 years, and that excluding the snow routine can change the seasonality of the flood peaks. Estimates for the 100- and 1000-year return level based on the stochastic PQRUT model are compared with results for a) statistical frequency analysis, and b) a standard implementation of the event-based PQRUT method. The differences in flood estimates between the stochastic PQRUT and the statistical flood frequency analysis are within 50% in most catchments. However, the differences between the stochastic PQRUT and the standard implementation of the PQRUT model are much higher, especially in catchments with a snowmelt flood regime.

## 1 Introduction

The estimation of low-probability floods is required for the design of high-risk structures such as dams, bridges, levees, etc. For example, floods with a 100-year return period are sometimes required for the design of levees and the design and safety evaluation of high-risk dams requires the estimation of flood hydrographs for the 1000-year return period and, in some cases, floods with magnitudes of up to the Probable Maximum Flood (PMF). An overview of design flood standards for reservoir engineering in different countries is provided in Ren et al. (2017). Flood mapping also usually requires input hydrographs for flood events with return periods of up to 1000 years. Methods for estimating these floods can be generally classified into three groups: 1) statistical flood frequency analysis; 2) the single design event simulation approach; and 3) derived flood frequency simulation methods.

At gauged sites, statistical flood frequency analysis involves fitting a distribution function to the annual maxima or peak over threshold flood events and calculating the quantile of interest. When return periods that are longer than the observed

record length are needed, the process requires extrapolation of the fitted statistical distribution. This introduces a high degree of uncertainty due to the number of limited observations relative to the estimated quantile (e.g. Katz et al., 2002). Significant progress has been made in methods for reducing this uncertainty by incorporating historic or paleo-flood data (Parkes and Demeritt, 2016), where available. Another way to "extend" the hydrological record in order to reduce the uncertainty is to

combine data series from several different gauges by identifying pooling groups or hydrologically similar regions, where this is possible. It has been found, however, that the identification of such hydrological regions can be difficult in practice (Nyeko-Ogiramoi et al., 2012). The application of statistical flood frequency analysis in ungauged basins is also problematic. As the physical processes in the catchments are usually not directly considered in the analysis, estimating the flood quantiles in ungauged basins using regression or geostatistical methods can produce average RMSNE (root mean square normalised error)

values of between 27 and 70% (Salinas et al., 2013), or even higher for the 100-year return period. In addition, the complete hydrograph is often needed in practice. Although multivariate analysis of flood events (e.g. flood peaks, volumes and durations) can be used to generate hydrographs for specific return periods, the methods are not easily applied (Gräler et al., 2013).

    The second method for extreme flood estimation is the design event approach in which single realizations of initial conditions and precipitation are used as input in an event-based hydrological model. Another feature of the approach is that when event-

based models are used, a critical duration defined as the duration of the storm that results in the highest peak flow needs to be included. Two advantages of this method over statistical flood frequency analysis are that rainfall records are often widely available (e.g in the form of gridded datasets) and that the event hydrograph is generated in addition to the magnitude of the flood peak. This approach has been traditionally used due to its simplicity (e.g. Kjeldsen, 2007; Wilson et al., 2011). However, its application often involves the assumption that the simulated flood event has the same return period as the rainfall used as

input in the hydrological model. This assumption is not realistic and, depending on the initial conditions, the return period of the rainfall and the corresponding runoff can differ by orders of magnitude (e.g. Salazar et al., 2017). A reason for this is that flood events are often caused by a combination of factors, such as a high degree of soil saturation in the catchment, heavy rainfall and seasonal snowmelt. A joint probability distribution, therefore, needs to be considered if one is to fully describe the relationship between the return period of rainfall and of runoff.

The third possible approach is the derived flood frequency method in which the distribution function of peak flows is derived from the distribution of other random variables such as rainfall depth and duration and different soil moisture states. Although a statistical distribution of flow values or their plotting positions are used to calculate the required quantiles, as in conventional flood frequency analysis, a hydrological model can be used to simulate an unlimited number of discharge values under differing conditions, thus extending and enhancing the observed discharge record. Under very stringent assumptions,

the derived distribution can actually be solved analytically by using a simple rainfall-runoff model (e.g. a unit hydrograph), assuming independence between rainfall intensity and duration and considering only a few initial soil moisture states. However, because of these simplifying assumptions, the method can produce poor results (Loukas, 2002). For this reason, methods based on simulation techniques are most often used, and these range from continuous simulations to event-based simulations with Monte Carlo methods.

In the continuous simulation approach, a stochastic weather generator is used to simulate long synthetic series of rainfall and temperature, which serve as input in a continuous rainfall-runoff model. The resulting long series of simulated discharge is then used to estimate the required return periods, usually using plotting positions (e.g. Calver and Lamb, 1995; Camici et al., 2011; Haberlandt and Radtke, 2014). A disadvantage of these methods is that they are computationally demanding, as long

continuous periods need to be simulated to estimate the extreme quantiles. Several newer methods, therefore, use a continuous weather generator coupled with an event-based hydrological model. For example, the hybrid-CE (causative event) method uses a continuous rainfall-runoff simulation to determine the inputs to an event-based model (Li et al., 2014). Another disadvantage, however, of continuous simulation models is that stochastic weather generators require the estimation of a large number of parameters (e.g. Onof et al., 2000; Beven and Hall, 2014). In addition, models such as the modified Barlett- Lewis rectan-

gular pulse model have limited capacity to simulate extreme rainfall depths, which can lead to an underestimation of runoff (Kim et al., 2017). In order to avoid the limitations of continuous weather generators, the semi-continuous method SCHADEX (Paquet et al., 2013) uses a probabilistic model for centred rainfall events (Multi-Exponential Weather Pattern-based distribution; Garavaglia et al. (2010)), identified as over threshold values that are larger than the adjacent rainfall values. Using this approach, millions of synthetic rainfall events can be generated, assigned a probability estimated from the MEWP model and

inserted directly into the historic precipitation series to replace observed rainfall events. In this manner, the SCHADEX method is similar to the hybrid-CE methods because a long-term hydrological simulation is used to characterize observed hydrological conditions and synthetic events are only inserted into the precipitation record for periods selected from the observed record. Despite the many advantages of the hybrid-CE and SCHADEX methods over continuous simulation methods, they neverthe-less require sufficient data for the calibration of the hydrological model, modelling of the extreme precipitation distribution and

for ensuring that an exhaustive range of initial hydrological conditions are sampled during the simulations.

Another method for derived flood frequency analysis is the joint probability approach (e.g. Muzik, 1993; Loukas, 2002; Svensson et al., 2013; Rahman et al., 2002). In this approach, Monte Carlo simulation is used to generate a large set of initial conditions and meteorological variables, which serve as input to an event-based hydrological model. This approach requires that the important variables are first identified and any correlations between the variables are quantified. Most often, the random

variables that are considered are related to properties of the rainfall (intensity, duration, frequency) and to the soil moisture deficit. Some of these methods, such as the Stochastic Event Flood Model (SEFM, (Schaefer and Barker, 2002)), similarly to SCHADEX, require the use of a simulation based on a historical period to generate data series of state variables from which the random variables are sampled. Although the contribution of snowmelt can be important in some areas, it is rarely incorporated as it requires the generation of a temperature sequence for the event that is consistent with the rainfall sequence

used and a snow water equivalent as an initial condition. The assumption of a fixed rate of snowmelt which is based on typical temperatures, as is often used in Norway for the single event-based design method, can introduce a bias in the estimates. The joint probability of both rainfall and snowmelt needs to be considered to obtain a probability neutral value (Nathan and Bowles, 1997). One of the few methods that incorporates snowmelt is the SEFM which has been applied in several USBR (US Bureau of Reclamation) studies and uses the semi-distributed HEC-1 hydrological model (Schaefer and Barker, 2002).

Considering that, most often, simple event-based hydrological models are used (e.g. unit hydrograph), the joint probability

approach is particularly advantageous in ungauged catchments or data-poor catchments, where the use of parsimonious models is preferred.

Even though methods for derived flood frequency analysis are becoming more commonly used in practice as they can provide better estimates of the high flood quantiles (e.g. Australian Rainfall and Runoff 2016 (Ball, 2015), SCHADEX method), this kind of method has not yet been established in Norway. The purpose of this study is hence to develop a derived flood frequency method using a stochastic event-based approach to estimate design floods, including those with a significant contribution from snowmelt. In this way, the results for any return period can be derived, taking into account the probability of a range of possible initial conditions. A sensitivity analysis is then performed to understand the uncertainty in the stochastic PQRUT model and establish the relative roles of several factors, including the rainfall model, snowmelt, the initial soil moisture parameters of the model and the length of the simulation. The results are then compared with results from an event-based modelling method based on a single design precipitation sequence and assumed initial conditions and with statistical flood frequency analysis of the observed annual maximum series for a set of catchments in Norway for the 100- and 1000 -year return periods.

## 2   Stochastic event-based flood model

The stochastic event-based model proposed here involves the generation of several hydrometeorological variables: precipitation depth and sequence, the temperature sequence during the precipitation event, the initial discharge, and the antecedent soil moisture conditions and snow water equivalent. A simple 3-parameter flood model PQRUT (Andersen et al., 1983) is used to simulate the streamflow hydrograph for a set of randomly selected conditions based on these hydrometeorological variables. After this procedure is completed 100,000 times for each season, the results are combined and a flood frequency curve is constructed from all of the simulations using their plotting positions. As the method requires initial values for soil moisture and snow water equivalent, i.e variables which generally cannot be sampled directly from climatological data and which depend on the sequence of precipitation and temperature over longer periods, the Distance Distribution Dynamics (DDD) hydrological model (Skaugen and Onof, 2014) was calibrated and run for a historical period to produce a distribution of possible values for testing the approach. The method (also shown in Fig. 1) can be outlined in summary form as follows:

1. Extract flood events for a given catchment and identify the critical storm duration

   For each season:

2. Aggregate the precipitation data to match the critical duration for the catchment

3. Extract POT precipitation events and fit a GP distribution

4. Fit probability distributions for the initial discharge, soil moisture deficit and SWE values for the season

5. Generate precipitation depth from the fitted GP distribution

6. Disaggregate the precipitation depth to a 1 hour time step by matching the dates of the identified POT flood events (from step 3) to dataseries of precipitation with an hourly timestep

7. Sample a temperature sequence by matching the dates of the identified POT flood events (from step 1) to the dataseries of temperature with an hourly timestep

8. Sample initial conditions for snow water equivalent (*SWE*), soil moisture deficit and initial discharge from their distributions (step 5), accounting for co-variation using a multivariate normal distribution

9. Simulate streamflow values using the calibrated PQRUT model for the sample event

10. Repeat steps 5.-9. 100 000 times

11. Estimate the annual exceedance probability from the total of 400 000 (i.e.100 000 for each season) samples using plotting positions

The study area and data requirements for the proposed method are described in section 2.1, while section 2.2 describes the method for determining the critical duration, and section 2.3 and 2.4 describe the generation of antecedent conditions and meteorological data series. The hydrological model is presented in section 2.5, the method for constructing the flood frequency curve is outlined in section 2.6 and the sensitivity analysis is presented in section 2.7.

## 2.1 Study Area and Data requirements

### 2.1.1 Catchment selection and available streamflow data

The study area in Norway, consisting of a dataset of 20 catchments located throughout the whole country, is shown in Fig 2. All catchments have at least 10 years of hourly discharge data, and in all cases the length of the daily flow record is considerably longer than 10 years. All selected catchments are members of the Norwegian Bench Mark dataset (Fleig et al., 2013), which ensures that the data series are unaffected by significant streamflow regulation and have discharge data of sufficiently high quality suitable for the analyses of flood statistics. The catchment size was restricted to small and medium-sized catchments (maximum area is 854 km$^2$), as the structure of the 3-parameter PQRUT model does not take into account the longer-term storage processes which can contribute to delaying the runoff response during storm events. Previous applications of PQRUT in Norway indicate that this shortcoming is most problematic for larger catchments. Discharge datasets with both daily and hourly time steps were obtained from the national archive of streamflow data held by NVE (https://www.nve.no/). The catchments were delineated and their geomorphological properties were extracted using the NEVINA tool NVE (2015), except for *Q*, which was calculated using the available streamflow data and *P*, which was calculated using available gridded data (further details are given in 2.1.2 below). In order to illustrate the application of the method, we have selected three catchments which can be considered representative for different flood regimes in Norway: Krinsvatn in western Norway, Øvrevatn in northern Norway and Hørte in southern Norway (Fig. 2).

Table 1 summarises the climatological and geomorphological properties of these three catchments, including: area (*A* in km$^2$), mean annual runoff (*Q* in mm year-1), mean annual precipitation (*P* in mm year-1), mean elevation (*Hm50*), percentage forest-covered area (*For*), percentage marsh-covered area (*M*), percentage area with sparse vegetation above tree line (*B*),

'effective' lake percentage (*Lk*), catchment steepness (*Hl*) and the mean annual temperature in the catchment (*Temp*). The effective lake percent (*Lk*) is used to describe the ability of water bodies to attenuate peak flows such that lake areas which are closer to the catchment outlet have a higher weight than those near the catchment divide. It is calculated as $\frac{\sum A_i \times a_i}{A^2} \times 100$, where $a_i$ is the area of lake i, $A_i$ is the catchment area upstream of lake i and $A$ is the total catchment area. The dominant land cover for Krinsvatn and Øvrevatn is sparse vegetation over tree line, while the land cover for Hørte is mainly forest. The effective lake percentage *Lk* is insignificant for Hørte and Øvrevatn, but for Krinsvatn, the *Lk* is higher and the area covered by marsh, *M*, is 9%. The catchment steepness ( *Hl*) (defined as ( *Hm75*- *Hm25*)/ *L*, where *L* is the catchment length and *Hm25* and *Hm75* are the 25 and 75 quantiles of the catchment elevation) is highest for Hørte (18.7 m/km) and lowest for Krinsvatn (5.4 m/km). The catchment Krinsvatn, being located near the western coast of Norway, has a much higher mean annual precipitation (*P*), i.e. an average of 2354 mm year[-1], in comparison with Hørte (1261 mm year-1) and Øvrevatn (1558 mm year[-1]). The dominant flood regime for Krinsvatn is primarily rainfall-driven high flows, as the catchment is located in a coastal area and is characterised by high precipitation values and an average annual temperature of around 4° C. The highest observed floods, however, also have a contribution from snowmelt. The season of the AMAX (annual maxima flood) is the winter period, i.e. December – February, although high flows can occur throughout the year. Hørte has a mixed flood regime with most of the AMAX flood events in the period September–November, but in some years annual flood events occur in the period March–May and are associated with rainfall events during the snowmelt season. Øvrevatn has a predominantly snowmelt flood regime with most AMAX flood events occurring in the period June – August, due to the lower temperatures in the region such that precipitation falls as snow during much of the year.

### 2.1.2   Available meteorological data

Data for temperature and precipitation with daily time resolution were obtained from seNorge.no. This dataset is derived by interpolating station data on a 1 km$^2$ grid and is corrected for wind losses and elevation (Mohr, 2008). In addition, meteorological data with a sub-daily time step is needed for calibrating the PQRUT model, as many of the catchments have fast response times. For this, precipitation and temperature data with a three-hour resolution, representing a disaggregation of the 24-hour gridded seNorge.no data using the HIRLAM hindcast series (Vormoor and Skaugen, 2013), were used. The HIRLAM atmospheric model for northern Europe has a 0.1 degree resolution (around 10 km$^2$), and we used a temporal distribution of three hours. The HIRLAM data set was first downscaled to match the spatial resolution of the seNorge data, and the precipitation of the HIRLAM data was rescaled to match the 24-hour seNorge data (Vormoor and Skaugen, 2013). Then, these rescaled values were used to disaggregate the seNorge data to a 3-hour time resolution. The method was validated against 3-hour observations, and the correlation of the method was found to be higher than that obtained by simply dividing the seNorge data into eight equal 3 -hourly values (Vormoor and Skaugen, 2013). These datasets were further disaggregated to a 1-hour time step by dividing the 3-hourly values into three equal parts to match the time resolution of the streamflow data.

### 2.1.3 Initial conditions

The stochastic PQRUT method requires time series of soil moisture deficit, *SWE* and initial discharge. These data series are used to construct probability distribution functions for generating initial conditions for the event-based simulations. Sources of these data can be e.g. remotely sensed data (see, for example, the review provided in Brocca et al. (2017)) or gridded hydrological models. In this study, the DDD hydrological model was used to simulate these data series. The DDD model is a conceptual model that includes snow, soil moisture and runoff response routines and is calibrated for individual catchments using a parsimonious set of model parameters. The snowmelt routine of DDD model uses a temperature-index method and accounts for snow storage and melting for each of 10 equal area elevation zones. The soil moisture routine is based on one dynamic storage reservoir, in which we find both the saturated- and the unsaturated zones, having capacities which vary in time. The flow percolates to the saturated zone if the water content in the unsaturated zone exceeds 0.3 of its capacity. The response routine includes routing of the water in the saturated zone using a convolution of unit hydrographs based on the distribution of distances to the nearest river channel within the catchment and from the distribution of distances within the river channel.

## 2.2 Critical Duration

When simulating flood response with an event-based model, it is important to specify the so-called critical duration (Meynink and Cordery, 1976) to ensure that the flood peak is correctly modelled. The critical duration is an important quantity which effectively links the duration and the intensity of precipitation events of a given probability. To determine the critical duration, flood events from the daily time series over a quantile threshold (in this case, 0.9) were extracted. The POT (peak over threshold) flood events were considered independent if they were separated by at least seven days of values below the threshold. The day with the maximum peak value of streamflow was then identified for each event. The peak values were tested for correlations with the precipitation on the day of the peak flow and on days -1, -2 and -3 before the peak. The critical duration was determined as the number of days in which the correlation between the precipitation and the streamflow was higher than 0.25. This threshold value was selected because it gave realistic durations for the catchments in the study area. As an alternative approach, the critical duration was also set to equal the number of days for which the correlation was significant at p=0.01. This method resulted, however, in very long durations, in some cases. A possible reason for this is that if there are only a few observations, even relatively low Pearson correlation coefficients can produce statistically significant p-values. In some catchments (mostly those having a snowmelt flood regime), no significant correlation was found between discharge and precipitation, and in this case the critical duration was determined by considering only flood events in the September-November (SON) season in which most events are caused by rainfall (in this case, during the autumn). If the critical duration was more than one day, the precipitation was aggregated to the critical duration by applying a moving window to the data series. For Hørte and Øvrevatn, the critical duration was found to be 24 hours, and for Krinsvatn it was found to be 48 hours (Fig. 3).

## 2.3 Precipitation and temperature sequence generation

In addition to the critical duration of the event, the sequence of the input data must be generated for the stochastic simulation. Snowmelt can be important in the catchments considered in this study, so both the sequence of precipitation and temperature must be considered. In order to account for seasonality, the meteorological data series were first split into standard seasons: DJF, MAM, JJA and SON. In this way, we ensure that more homogeneous samples are used to fit the statistical distributions. Precipitation events over a threshold (POT events) were identified in the precipitation dataseries and a Generalized Pareto distribution was fitted to the series of selected events. In order to choose a threshold value for event selection, two criteria were used: 1) the threshold must be higher than the 0.93 quantile, and 2) the number of selected events must be between two and three per season. Although other methods for threshold selection exist, such as the use of mean life residual plots, the described method is much simpler to apply and gives acceptable results (e.g. Coles, 2001). The selected threshold varied between the 0.93 to 0.99 quantiles, depending on the season and catchment. In addition, the exponential distribution is often fitted to POT events, as it can give more robust results than the GP distribution. Figure 4 shows the return levels calculated from the GP and Exponential distributions and the empirical return levels and demonstrates that it is appropriate to use these models. In this case, we have preferred to use the GP distribution due to the inclusion of the shape parameter for describing the behaviour of the highest quantiles. An exponential distribution, however, could also be used, as could a compound weather pattern-based distribution such as the MEWP distribution (e.g. Garavaglia et al., 2010; Blanchet et al., 2015). In addition, a temperature sequence with a 1-hour time resolution was identified from the disaggregated seNorge data, introduced in section 2.1.2, and extracted for each POT event. The precipitation depths were generated (for 100,000 events) from the fitted GP distribution for each season. Storm hyetographs were used to disaggregate the precipitation values as follows: a storm hyetograph was first sampled from the extracted hyetographs for the selected POT precipitation events (by matching the dates of the selected POT precipitation events to the disaggregated seNorge datataseries), taking into account seasonality, and the ratios between the 1-hour and the total precipitation for the event were calculated according to:

$$P_h sim = \frac{P_i}{sum(Pi)} P_d sim \tag{1}$$

where $P_h sim$ is the simulated 1-hour precipitation intensity, $P_d sim$ is the simulated daily intensity and $Pi$ is the 1-hour disaggregated SeNorge intensity. The calculated ratios were then used to rescale the simulated values.

## 2.4 Antecedent snow water equivalent, streamflow and soil moisture deficit conditions

In order to determine the underlying distributions for various antecedent conditions, the relevant quantities were extracted from simulations based on the DDD hydrological model of Skaugen and Onof (2014). The model was calibrated for the selected catchments at a daily timestep using a MCMC routine (Soetaert and Petzoldt, 2010). Outputs from DDD model runs were used to extract values for the initial streamflow, snow water equivalent (*SWE*) and soil moisture deficit, at the onset of the previously selected seasonal flood POT events. It is important to note that simulated values for the soil moisture deficit are used. However as described in Skaugen and Onof (2014), the model provides realistic values in comparison with measured groundwater levels. The POT event series used for this is the same as that used for identifying the critical duration (described in section 2.2).

After extracting the initial conditions, the correlation between the variables was tested for each season for each catchment. As the correlation between the variables is in most cases significant, the variables were jointly simulated using a truncated multivariate normal distribution. In order to achieve normality for the marginal distributions, the *SWE* and the discharge were log-transformed. In the spring and summer, the *SWE* is often very low or 0 in some catchments. If the proportion of non-zero values, p, was greater than 0.3 (around 15 observations), the values were simulated using a mixed distribution as:

$$F(X) = pG_1(x1) + (1-p)G_2(x2) \hspace{4cm} (2)$$

where $G_1$ represents the multivariate normal distribution with discharge, soil moisture deficit and *SWE* as variables (denoted as X1) and $G_2$ the bivariate normal distribution for the discharge and soil moisture (given as X2). The probability *p* for switching between the trivariate and bivariate distributions is based on the historical data for *SWE* higher than 0. In addition, because the initial conditions are not expected to include extreme values, the values of the initial conditions were truncated to be between the minimum and maximum of the observed ranges. The correlation between the observed and simulated variables is shown in Figure 5 for the Krinsvatn catchment, and although the distribution of simulated values exhibits a very good resemblance to that of the observed, there is not a perfect correspondence between the two. A reason for this may be that the variables (even after log transformation) do not exactly follow a normal distribution. We considered using copulas for the correlation structure of the initial conditions (Hao and Singh, 2016). However, as the data are limited in number (around 50 observations per season), these were much more difficult to fit. Similarly, nonparametric methods such as kernel density estimation were deemed to not be feasible due to the limited number of observations. Therefore, the multivariate normal distribution was chosen as the best alternative for modelling the joint dependency between the variables comprising the initial conditions for the stochastic modelling.

## 2.5 PQRUT model

The PQRUT (P-precipitation, Q-discharge and RUT-routing) model was used to simulate the streamflow for the selected storm events. The PQRUT model is a simple, event-based, 3-parameter model (Fig. 6) which is used for various applications, including estimating design floods and safety check floods for dams in Norway (Wilson et al., 2011). In practical applications, a design precipitation sequence of a given return period is routed through the PQRUT model, usually under the assumption of full catchment saturation. For this reason, only the hydrograph response is simulated, and there is no simulation of subsurface and other storage components, such as are found in more complex conceptual hydrological models. Of the three model parameters, $K_1$ corresponds to the fast hydrograph response of the catchment, and the parameter $K_2$ is the slower or 'delayed' hydrograph response. The parameter *Trt* is the threshold above which $K_1$ becomes active.

The PQRUT model was calibrated for the 45 highest flood events for each catchment by using the DDS (Dynamically Dimensioned Search) optimization routine (Tolson and Shoemaker, 2007) and the Kling Gupta efficiency (KGE) criterion (Gupta et al., 2009) as the objective function. An additional variable, the soil deficit, *lp*, was introduced to account for initial losses to the soil zone. The reason for this is that, even though fully saturated conditions are assumed when the model is used to estimate PMF or other extreme floods with low probabilities, the model needs to account for initial losses when actual

(more frequent) events are simulated during the calibration process. This procedure is described in more detail in Filipova et al. (2016). In addition, regional values can be used in ungauged or poorly catchments (Andersen et al., 1983; Filipova et al., 2016).

For the work presented here, the value of *lp* was set to the initial soil moisture deficit, estimated using DDD. This variable functions as an initial loss to the system, such that the input to the reservoir model is 0 until the value of *lp* is exceeded by the cumulative input rainfall. In order to model flood events involving snowmelt, a simple temperature index snow melting rate was used:

$$S = C_s(T - T_L) \tag{3}$$

where $S$ is the snow melting rate in mm/hour, *Cs* is a coefficient accounting for the relation between temperature and snowmelt properties and $T_L$ is the temperature threshold for snowmelt (here fixed at $0^0$ C). Regional values for the *Cs* parameters as a function of catchment properties, based on the ranges given in Midttømme and Pettersson (2011) were applied. In addition, the temperature threshold between rain and snow was set to $T_X = 0.5^0$C, which is typically used in Norway (Skaugen, 1998).

## 2.6 Flood frequency curves

Seasonal and annual flood frequency curves were constructed by extracting the peak discharge for each event and estimating the plotting positions of the points using the Gringorten plotting position formula:

$$P_e = \frac{(m - 0.44)}{(N + 0.12)k} \tag{4}$$

where $P_e$ is the exceedance probability of the peak, m is the rank (sorted in decreasing order) of the peak value, $N$ is the number of years, *k* is the number of events per year. The number of events per year, *k*, was set to be equal to the average number of extracted POT storm (precipitation) events per year. These simulated events were compared with the POT flood events extracted from the observations (Fig. 7). After calculating the probability of the simulated events using Eq. 4, the initial conditions and seasonality for a return period of interest can be extracted. For example, the events with return period between 90 and 110 years were extracted (representing around 80 events), and the hydrological conditions for those events were identified (Table 2). The results show that there is a large variation in the total precipitation depths and initial conditions that can produce flood events of a given magnitude and this is the reason why it is difficult to assign initial conditions in event-based models. However, it is still useful to extract the distribution of these values in order to ensure that the ranges are reasonable and the catchment processes are properly simulated. For example, the average snowmelt is negative (i.e. there is snow accumulation) for Krinsvatn, which means that in most cases snowmelt does not contribute to extreme floods. This is reasonable as the catchment is located in western Norway, where the climate is warmer (the mean temperature is around 4° C) and the mean elevation is low. The average snowmelt contribution for Øvrevatn is much higher as this catchment has a predominantly snowmelt flood regime. The soil moisture deficit for the three catchments is larger than 0, even though floods with relatively long return periods (i.e. between 90 and 100 years) are being sampled here. The seasonality of the simulated values is consistent with the seasonality of the observed annual maxima (Table 1).

## 2.7   Sensitivity analysis

A sensitivity analysis was performed for the three test catchments, Hørte, Øvrevatn and Krinsvatn, in order to determine the relative importance of the initial conditions, precipitation, the parameters of PQRUT, the effect of the random seed and length of the simulation on the flood frequency curve. To test the sensitivity of the model, we have used several different model runs and calculated the percentage difference of each of these model runs relative to the standard model setup, as shown in Fig.8. More detailed information on the setup is given in Table 3. As these catchments are located in different regions and exhibit different climatic and geomorphic characteristics, we hypothesize that the flood frequency curve will be sensitive to different parameters and hydrological states, as well as local climate and catchment characteristics. The results are summarised in Table 4.

The results for the sensitivity to the rainfall model are presented in Fig 8a. The results show that the temporal patterns of the rainfall input have a large impact (up to 50 %) on the flood frequency curve for Hørte and Krinsvatn, as these catchments have a predominantly rainfall-dominated flood regime. The impact is very little for Øvrevatn. A high sensitivity to the shape of the hyetograph was also found in Alfieri et al. (2008)). Their study shows that using rectangular hyetograph results in a significant underestimation of the flood peak while the Chicago hyetograph (e.g. Chow et al. (1988)), where the peak is in the middle of the event, resulted in overestimation. In addition, Øvrevatn and Hørte showed sensitivity (around 20%) to the choice of the statistical distribution for modelling precipitation. This means that the uncertainty in fitting the rainfall model can propagate to the final results of the stochastic PQRUT, and therefore, it is important to ensure that the choice of distribution and parameters is carefully considered. A high sensitivity to the parameters of the rainfall model was also described by Svensson et al. (2013), who tested the sensitivity of a stochastic event based model applied to four small to medium-sized catchments in the UK. Both Hørte and Krinsvatn showed relatively lower sensitivity to the threshold value for the GP distribution, compared to Øvrevatn.

In addition, all catchments are very sensitive to the parameters of the PQRUT model (Fig 8b) and there is large uncertainty in these values. Because of the higher sensitivity to the calibration of the rainfall-runoff model, a conclusion can be made that, in practice, if streamflow data is available it is important that this is used for calibrating the PQRUT model rather than relying on regionalised parameter values.

Considering the effect of the initial conditions (Fig 8c), using fully saturated conditions results in the slight overestimation overestimation of flood values for all catchments, as expected, and the impact is higher at lower return periods. In addition, Øvrevatn shows a higher sensitivity (around 26% for *Q1000*) to the initial soil moisture conditions than the other two catchments. A reason for this is that for Øvrevatn, higher soil moisture conditions are associated with higher rainfall quantiles. For example, the ratio of the maximum soil moisture to the mean rainfall is 65% while for Hørte and Krinsvatn is much less, i.e 33% and 26%, respectively. The initial discharge value does not seem to have a large impact for any of the catchments. If no snow component (no snowmelt and no snow accumulation) is used, there is not much difference in the results for Øvrevatn and Hørte, but the seasonality of the flood events is changed. For example, the season when the *Q1000* is simulated for Øvrevatn is SON instead of JJA when most of the AMAX values are observed. Due to this change of seasonality, the precipitation values that produce *Q1000* are accordingly higher (the median is around 15% higher). The soil moisture deficit, as expected, is also

somewhat higher and shows much more spread, with values up to 45 mm. In addition, Krinsvatn shows a high sensitivity to the snowmelt component (21% higher) and also a step change in the frequency curve, even though the soil moisture deficit is higher. This can also be explained by the fact that the snowmelt contribution is negative (i.e. there is snow accumulation), as can also be seen in Table 2. Other studies have also shown that the soil saturation level is not as important as the parameters of the hydrological model. For example, Brigode et al. (2014) tested the sensitivity of the SCHADEX model using a block bootstrap method. In each of these experiments, different sub-periods selected from the observation record were used in turn to calibrate the rainfall model, the hydrological model and to determine the sensitivity to the soil saturation level. The results showed that for extreme floods (1000–year return period), the model is sensitive to the calibration of the rainfall and the hydrological models, but not so much to the initial conditions.

The stochastic PQRUT model shows some sensitivity to the random seeds (Fig 8d), especially for higher return periods. This is expected as the higher quantiles are calculated using a smaller sample of simulated events. Similarly, the effect of the simulation length has a larger impact on the higher quantiles (e.g. Q1000). However, the length of the simulation will depend on the required return level, as shorter simulation length can be acceptable for lower return periods, e.g. *Q100*.

## 3  Comparison with standard methods

### 3.1  Implementation of the methods and results

The results of the stochastic PQRUT method for the 100- and 1000-year return level were compared with the results for statistical flood frequency analysis and with the standard implementation of the event-based PQRUT method (in which full saturation and snow melting rates are assumed a priori) for the twenty test catchments described in section 2.1. For the statistical flood frequency analysis, the annual maximum series were extracted from the observed daily mean streamflow series. The GEV distribution was fitted to the extracted values using the L-moments method and the return levels were estimated. In order to obtain instantaneous peak values, the return values were multiplied by empirical ratios, obtained from regression equations, as given in Midttømme and Pettersson (2011). The ratios can vary substantially from catchment to catchment, and in this study, the values are from 1.02 to 1.82, depending on the area and the flood generation process (snowmelt or precipitation). Although much more sophisticated methods could be used to obtain statistically-based return levels, the procedure used here is equivalent to that currently used in standard practice in design flood analysis in Norway. In addition, the length of the daily streamflow series justifies the use of at-site flood frequency analysis (Kobierska et al., 2017); the minimum length is 31 years, while the median is 65 years of data. However, it is expected that the uncertainty will be high when the fitted GEV distribution is extrapolated to a 1000-year return period. The 1000-year return period is used here, however, as it is required for dam safety analyses in Norway (e.g. Midttømme, et al., 2011; Table 1). More robust, but potentially less reliable, estimates could be obtained using a 2-parameter Gumbel, rather than a 3-parameter GEV distribution (Kobierska et al., 2017). The standard implementation of PQRUT involves using a precipitation sequence that combines different intensities, obtained from growth curves based on the 5-year return period value fitted using a Gumbel distribution while the ratios between the different durations are derived from empirical distribution (Førland, 1992). The precipitation intensities were combined to form a single

symmetrical storm profile with the highest intensity in the middle of the storm event whereas the storm profile is randomly sampled for the stochastic PQRUT model (Fig 9). In the application reported here, the duration of the storm event was assumed to be the same as the critical duration used for the stochastic PQRUT model. The initial discharge values were similarly fixed to the seasonal mean values, as is common in standard practice. The snowmelt contribution for the 1000-year return period was, in this case, assumed to be 30 mm/day for all catchments, which corresponds to 70% of the maximum snowmelt, estimated as 45 mm/day by using temperature-index factor of 4.5 mm/°C day and 10/°C. The snowmelt contribution for the 100-year return period was assumed to be 21 mm/day for all catchments. In addition, fully saturated conditions were assumed for both the estimation of the 100- and 1000-year return periods. A similar implementation of PQRUT for the purposes of comparing different methods has also been described in Lawrence et al. (2014). The parameters of the PQRUT were estimated by using the regional equations derived in Andersen et al. (1983), as these are still used in standard practice.

The performance of the three models was validated by using two different tests. Test 1 assessed whether the estimated values for the 100- year return period are within the confidence intervals of a GP distribution fitted to the streamflow data with a 1-hour time step. The stochastic PQRUT shows good agreement with the observations (Fig. 10), and for 18 of the 20 catchments, all the points of the derived flood frequency curve were inside the confidence intervals. As expected, for most of the catchments (16 out of 20) the return levels calculated using statistical flood frequency analysis based on the GEV distributions using daily values were within the confidence intervals. For the standard PQRUT model, the values of the 100-year return level were within the confidence interval for only six of the catchments when the regional equations for the PQRUT model were used and for only eight of the catchments when calibrated parameters are used. In test 2, the results of the flood frequency analysis and the Stochastic PQRUT methods were compared, based on a quantile score (QS) suggested by E. Paquet (personal communication). This is given in Eq 5:

$$QS = 1 - \sum_{i=1}^{N}(abs(Qmod_i - Qobs_i)(Qobs_i - Qobs_{i-1})) \tag{5}$$

In Eq. 5 the observed probabilities ($Qobs_i$) are calculated using Gringorten positions for the peak AMAX series that were derived from the daily values. The modelled probabilities that correspond to the observed events are calculated by using the statistical flood frequency analysis and the Stochastic PQRUT model, as described previously. The standard implementation of the event-based PQRUT model was not evaluated based on QS as initial conditions could not be assigned for low return periods. As this model is usually used to calculate high quantiles (*Q100* or higher), fully saturated conditions are assumed for its implementation. The results for the quantile score show similar performance, the median is approximately 0.65 for both methods. However, the results vary between catchments as shown in Fig 11. Although it is difficult to evaluate the performance of the models when the dataseries are relatively short, based on the results of test 1, we can conclude that the performance of the standard PQRUT model is poorer than the performance of the statistical flood frequency analysis and the stochastic PQRUT model for the selected catchments, while the results of test 2 indicate that both the GEV distribution and the stochastic PQRUT provide similar fits to observed quantiles.

## 3.2 Discussion

A comparison of the stochastic PQRUT with the standard methods for flood estimation shows that there is a large difference between the results of the three methods for both *Q100* and *Q1000* (Fig. 12 and 13). The boxplots (Fig. 12) show that the stochastic PQRUT method gives slightly lower results on average than the standard PQRUT model for *Q100* and *Q1000*. This is probably due to assuming fully saturated conditions when applying the standard PQRUT for *Q100*, which might not be realistic for some catchments. For example, the results for the initial conditions for the three catchments, presented in section 2.6, show that the soil moisture deficit is larger than 0 for *Q100*. Furthermore, the absolute differences between the two methods are larger in catchments with lower temperature (Fig. 12). This indicates that the performance of the standard PQRUT model is worse in catchments with a snowmelt flood regime, which may be either due to the difficulty in determining the snowmelt contribution or to the poorer performance of the regional parameters in catchments with a snowmelt flow regime. Although it shows a similar pattern, the standard PQRUT model, implemented using calibrated parameters results in much less spread than the implementation using the regionalised parameters, when compared to both the GEV distribution and the stochastic PQRUT model. This means that the hydrological model can introduce a large amount of uncertainty, as also indicated by the sensitivity analysis described in section 2.7 and previous results presented by Brigode et al. (2014).

The differences between the stochastic PQRUT model and the GEV fits are much smaller than the differences between the standard PQRUT model and the GEV fits, even when calibrated parameters are used for the PQRUT modelling. The differences are larger (i.e. the stochastic PQRUT results are lower, as shown in Fig. 13) in western Norway where *P* and *Q* are higher and for steeper catchments, i.e. with a higher value of *Hl*. A reason for this might be that the empirical ratios that are used to convert daily to peak flows in these catchments are inaccurate and possibly too high. Similarly to the boxplots, Fig. 13 also shows that the results of the stochastic PQRUT closely match the GEV distribution fits with differences within 50% for most locations. There is no clear spatial pattern in the differences between estimates based on the GEV distribution and on the standard PQRUT model, except for the catchments in mid-Norway, i.e. Trøndelag (including catchment Krinsvatn), where the GEV distribution produces higher results. However, a much larger sample of catchments is needed to assess whether there is a spatial pattern in the performance of the methods.

## 4 Conclusions

In this article, we have presented a stochastic method for flood frequency analysis based on a Monte Carlo simulation to generate rainfall hyetographs and temperature series to drive a snowmelt estimation, along with the corresponding initial conditions. A simple rainfall-runoff model is used to simulate discharge, and plotting positions are used to calculate the final probabilities. In this way, we can generate thousands of flood events and base extreme flood estimates on the empirical distribution, instead of extrapolating a statistical distribution fitted to the observed events. The approach thereby gives significant insights into the various combinations of factors that can produce floods with long return periods in a given catchment, including combinations of factors that are not necessarily well represented in observed flow series. It is thus a very useful complement to statisti-

cal flood frequency analysis and can be particularly beneficial in catchments with shorter streamflow series compared to the precipitation record as well as in ungauged catchments.

In order to apply the method, we assume that the precipitation and temperature series are not significantly correlated with the initial conditions, which allows us to simulate them as independent variables. Although we have not performed a statistical analysis, the independence between the precipitation events and the initial conditions has been verified e.g. (Paquet et al., 2013). Due to the considerable seasonal variation in the initial conditions and in the rainfall distribution, seasonal distributions were used. In addition to obtaining more homogeneous samples, this allows for a check of the seasonality of the flood events, which can be of interest in catchments with a mixed flood regime. In this study, we have used a GP distribution to model the extreme precipitation. However, if only shorter precipitation dataseries are available, the exponential distribution or even regional frequency analysis methods may provide more robust results. A limitation of the method is that PQRUT can only be used for small and medium-sized catchments, since its three parameters cannot take into account spatial variation in the snowmelt and soil saturation conditions within the catchment. However, for the catchments presented in this study (all with a catchment area under 850 km$^2$), the model produces relatively good fits to the observed peaks, even though it uses a very limited number of parameters.

In this study, initial conditions based on simulations using a hydrological model (DDD) were used. This requires that this model is calibrated for each catchment. Considering the results of the sensitivity analysis, the quality of the initial conditions is not as important as that of the precipitation data for the estimation of extreme floods (with return periods higher than 100 years). This means that if no other data is available, the output of gridded hydrological model could be considered as a source of this input data. Alternatively, remotely sensed data can be used for soil moisture and the snow water equivalent while regional values for the initial discharge can be derived. This for example, can be an option in ungauged basins.

The stochastic PQRUT model was applied to 20 catchments, located in different regions of Norway and was compared with the results of statistical flood frequency analysis and the event-based PQRUT method, which is today used in standard practice. Due to the high uncertainty in estimating extreme floods, the application of the different methods produces differing results, as is often the case in practical applications. However, in this work we have shown that the stochastic PQRUT model gives estimates which generally are more similar to those obtained using a statistical flood frequency analysis based on the observed annual maximum series than are estimates obtained using a standard implementation of PQRUT. As it is not possible to test the reliability of estimates for the 500- or 1000-year flood (due to length of the observed streamflow series relative to the return period of interest), the use of alternative methods for flood estimation, including stochastic simulations such as presented here, is an essential component of flood estimation in practice.

*Code and data availability.* The R package StochasticPQRUT (https://github.com/valeriyafilipova/StochasticPQRUT) can be installed from github and contains sample data.

*Competing interests.* On behalf of all authors, the corresponding author states that there is no conflict of interest.

*Acknowledgements.* This work has been supported by a PhD fellowship to Valeria Filipova from USN-Bø. Additional funds from the Energix FlomQ project supported by the Norwegian Research Council and EnergiNorge have partially supported the contributions of the co-authors to this work. The authors wish to thank Emmanuel Paquet (EDF) for suggesting the use of a simple quantile score for comparing the simulations with the observed higher quantiles, and two anonymous reviewers for their very detailed and helpful comments on the manuscript.

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

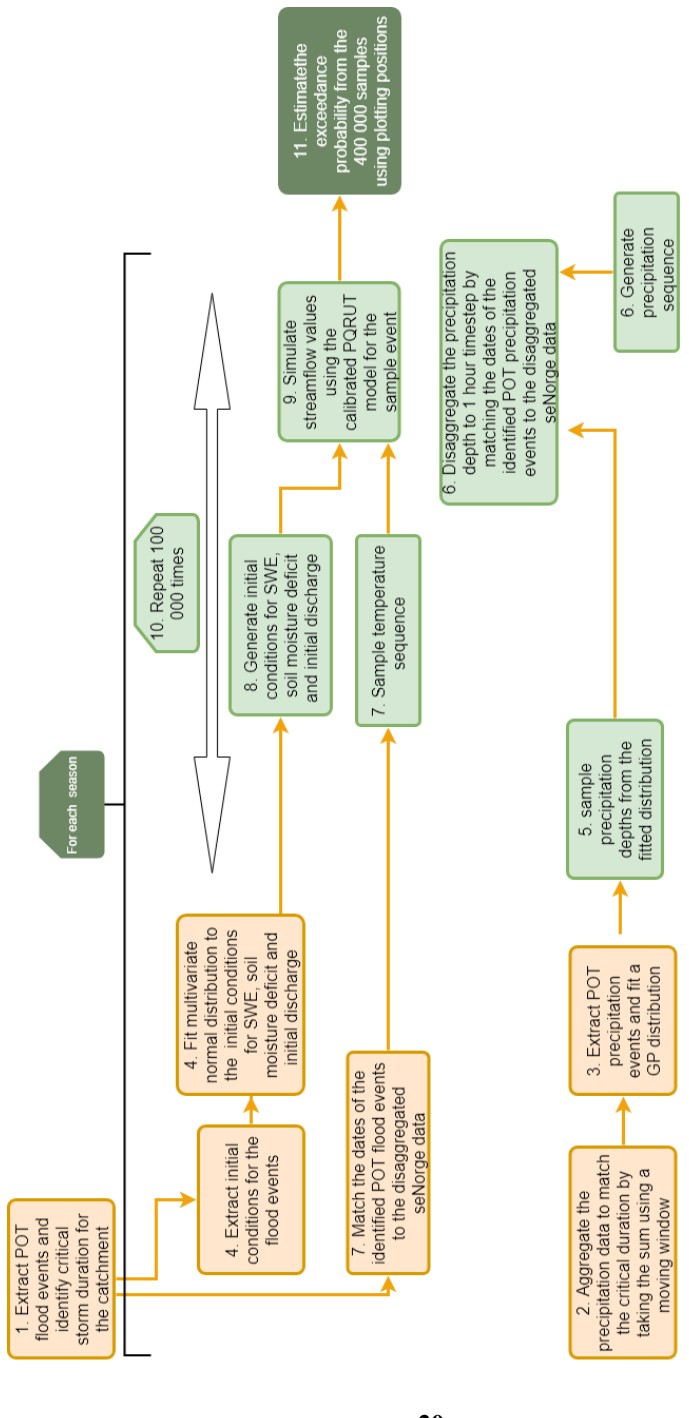

**Figure 1.** Diagram of the Stochastic PQRUT model

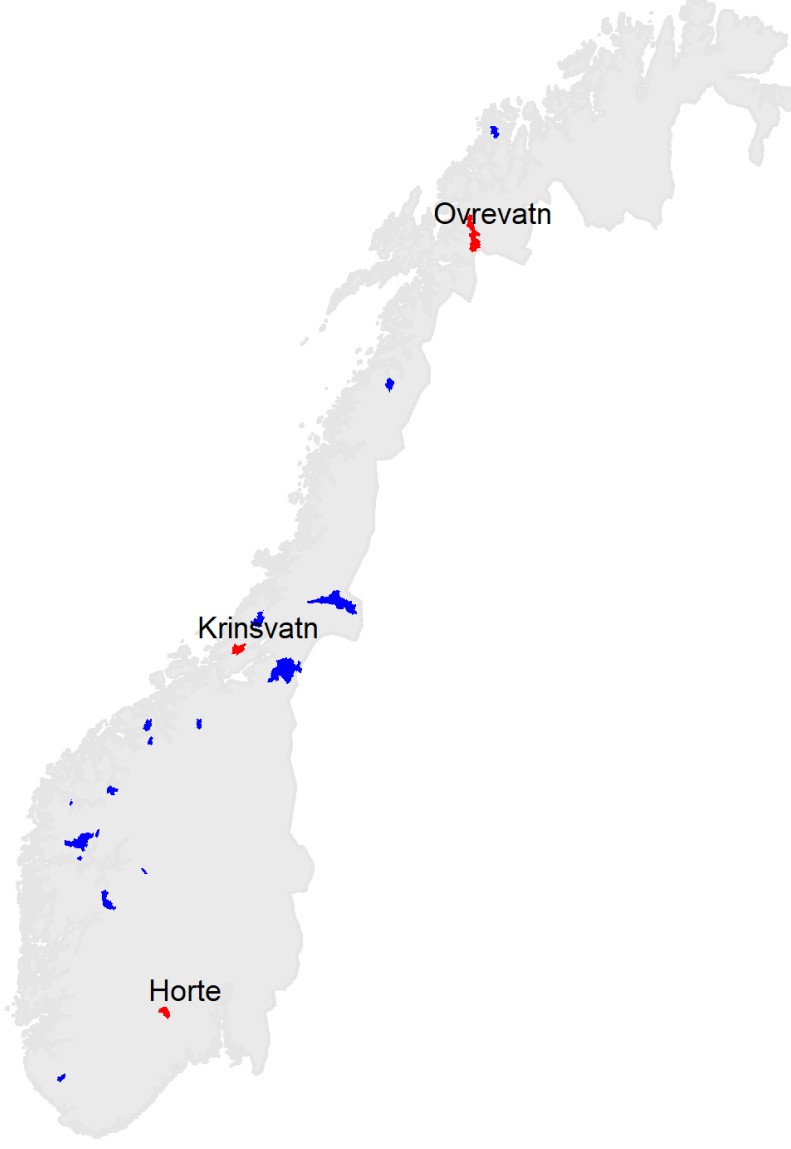

**Figure 2.** Location of the selected catchments. The catchments Hørte, Øvrevatn and Krinsvatn, for which we show the method in more detail, are plotted in red.

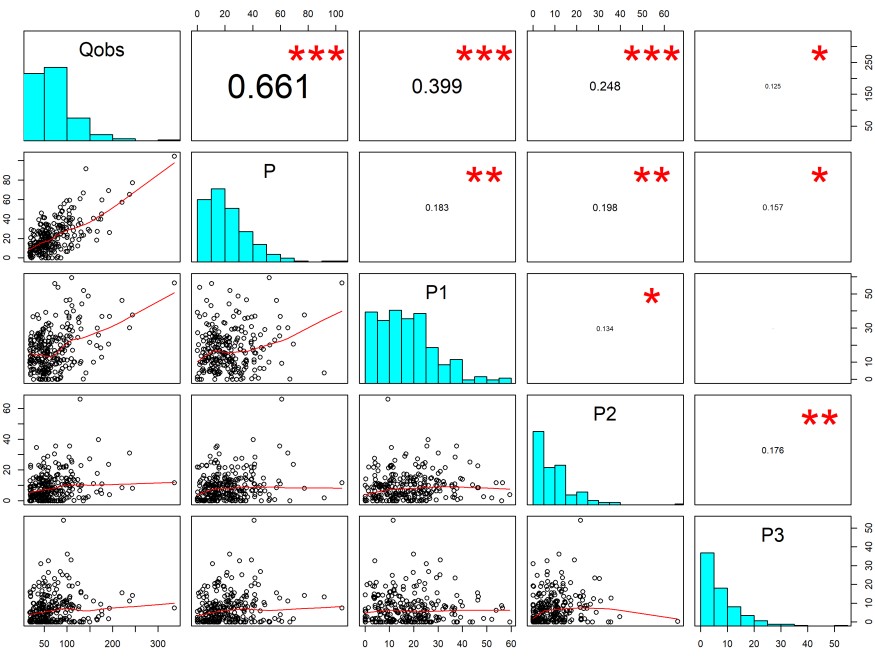

**Figure 3.** Diagram representing the process used to establish the critical duration for Krinsvatn. The stars represent the degree of significant correlation between *Qobs* and *P* on the day of the peak and -1, -2 and -3 days before the peak at p=0.01. The critical duration is set in this case to two days because the correlation between *Qobs* and *P* and *P1* is greater than 0.25.

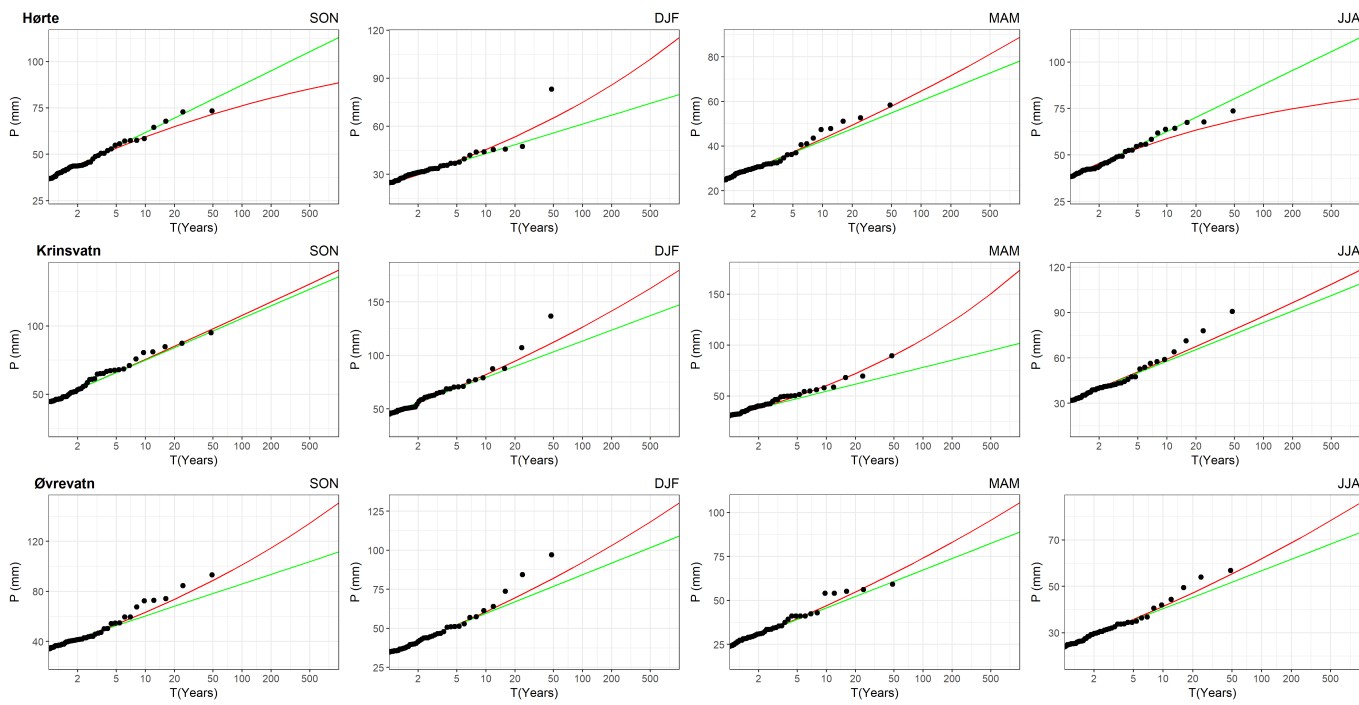

**Figure 4.** Return level plots for the fitted Generalised Pareto (GP) and exponential (EXP) distributions for peak over threshold precipitation events with duration equal to the critical duration (GP – red; EXP – green) for the three selected catchments.

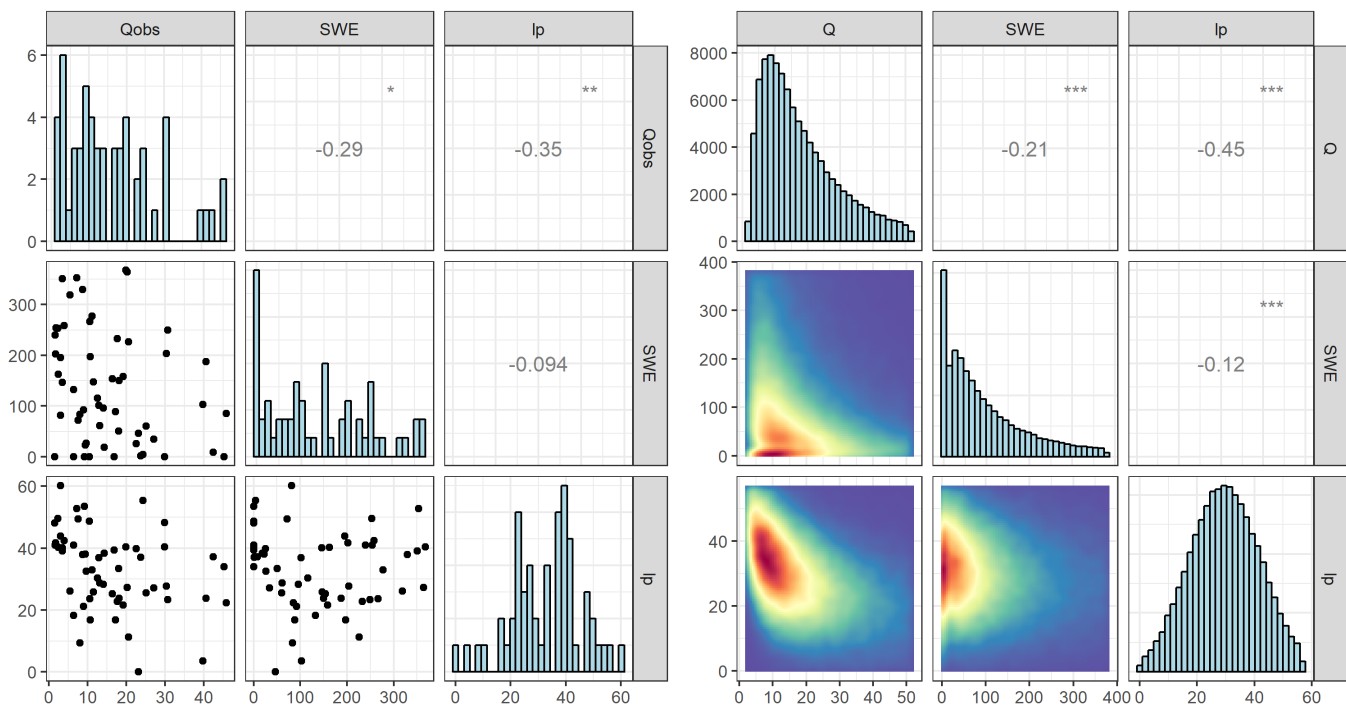

**Figure 5.** Correlation scatterplots for initial conditions (snow water equivalent (*SWE*) in mm, soil moisture deficit (*lp*) in mm, initial discharge (*Qobs*) in m3/s) for the Krinsvatn catchment for POT events (flood events over 0.9 quantile, 57 observations). Scatterplots for observed quantities are shown on the left and simulated on the right (based on 100 000 simulations). The stars represent the degree of significant correlation.

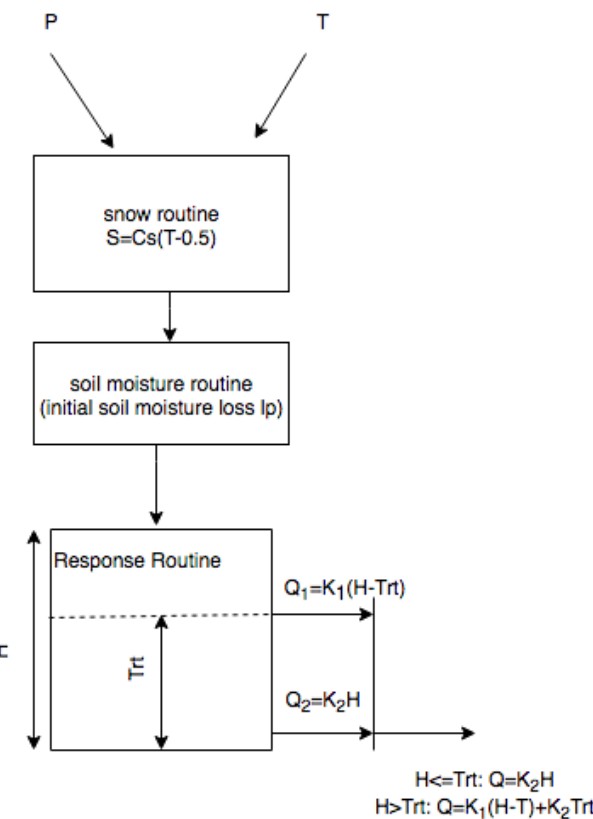

**Figure 6.** Structure of the PQRUT model

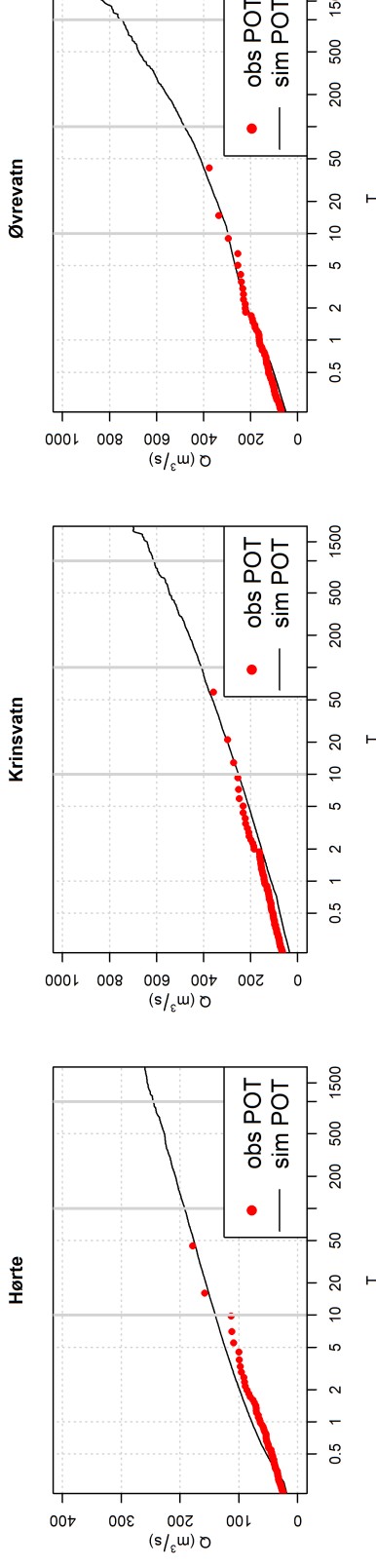

**Figure 7.** Comparison between the observed and simulated flood frequency curves of peak discharge for Hørte, Krinsvatn and Øvrevatn

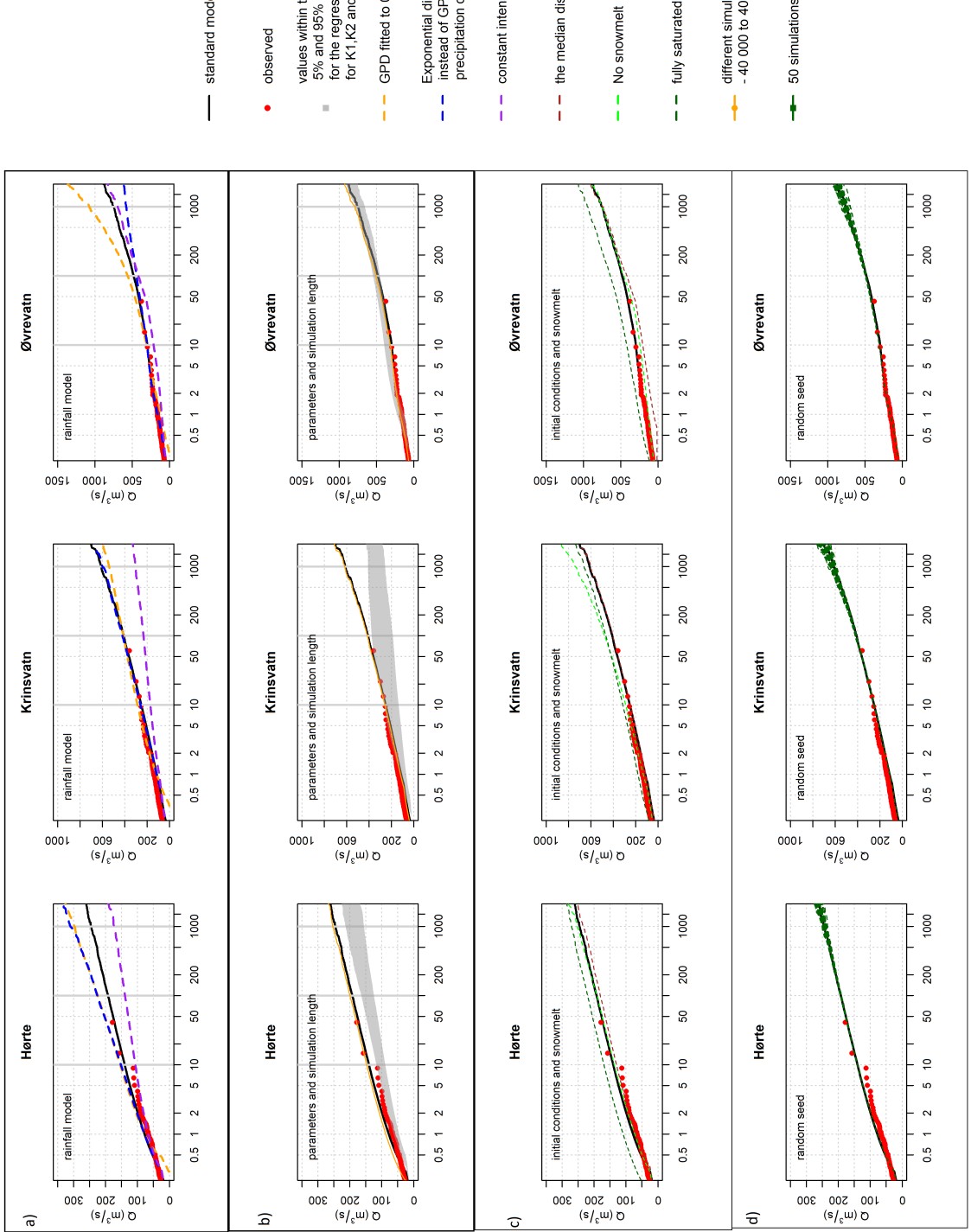

**Figure 8.** Sensitivity analysis of the Stochastic PQRUT for the following variables: a) the rainfall model , b) the effect of the simulation length and the parameters of the hydrological model , c) the initial discharge values, snowmelt conditions and catchment saturation, and the d) the random seed used for the simulations. Detailed information on the setup is given in table 3

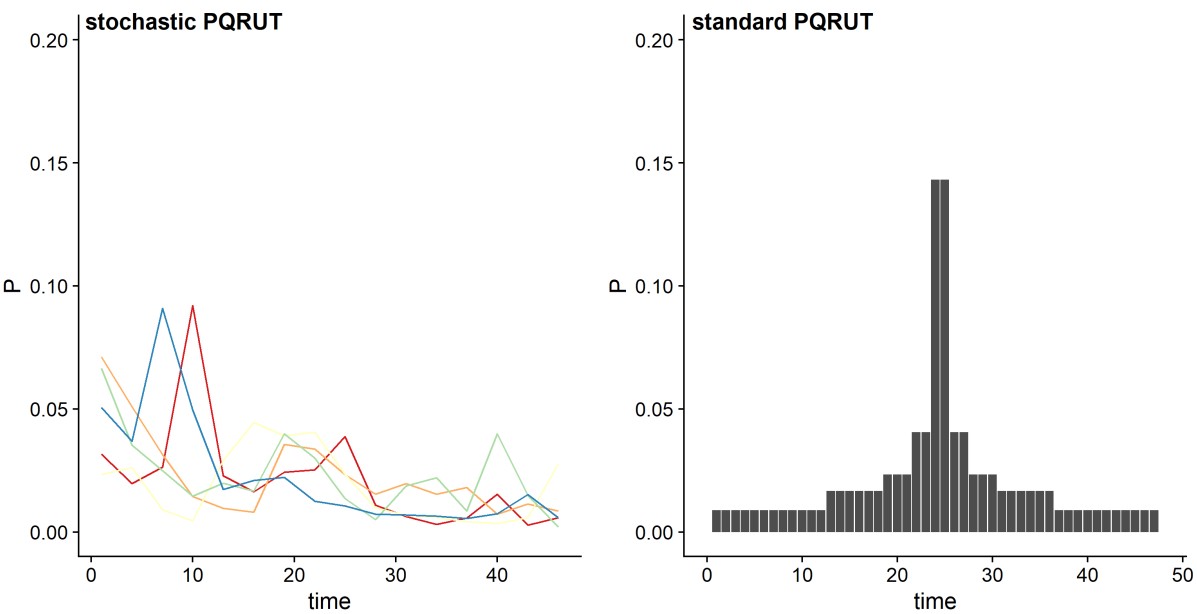

**Figure 9.** An example of storm patterns (only five are shown here) used for the simulated events in the stochastic PQRUT model (left) and the storm pattern typically used with the standard PQRUT model (right). P represents the ratio of the hourly precipitation to the total precipitation depth for the event.

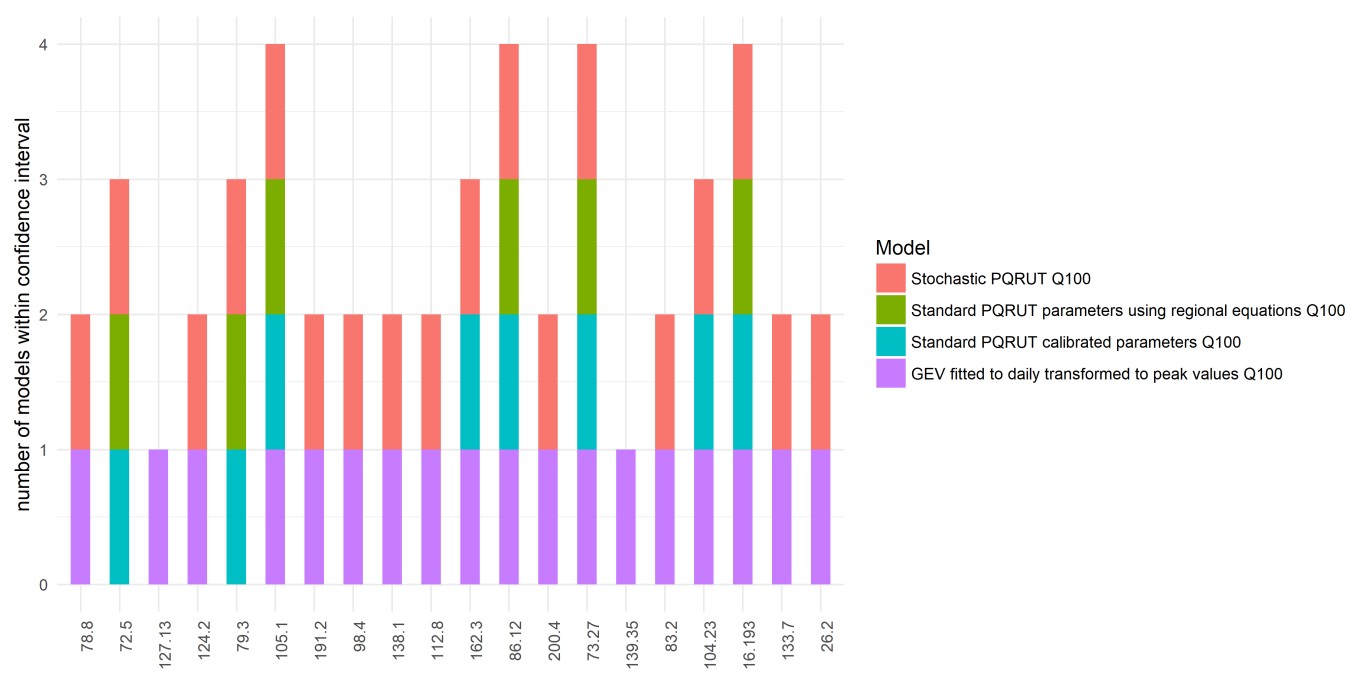

**Figure 10.** Number of models giving estimates for Q100 within the confidence intervals of the GP distribution fitted to 1-hour streamflow data.

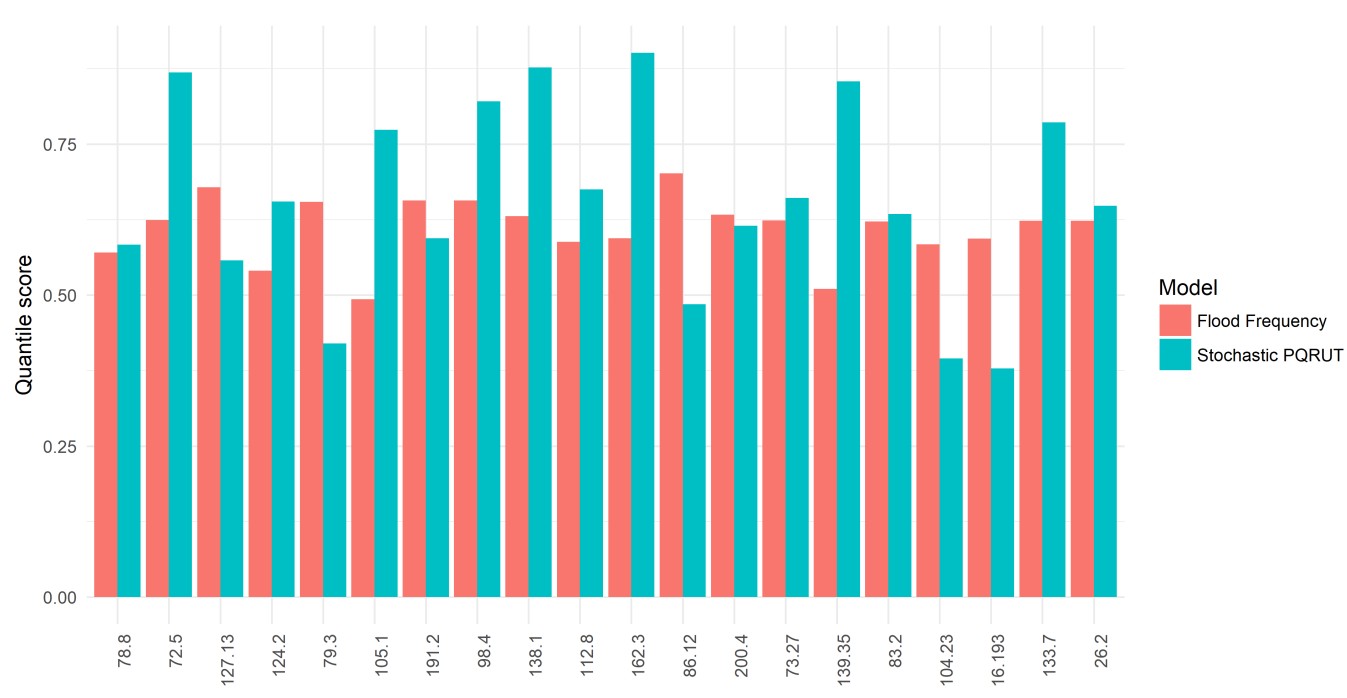

**Figure 11.** Quantile score (Eqn. 5) for estimates based on statistical flood frequency analysis using a GEV distribution (red) and on the stochastic PQRUT model (blue).

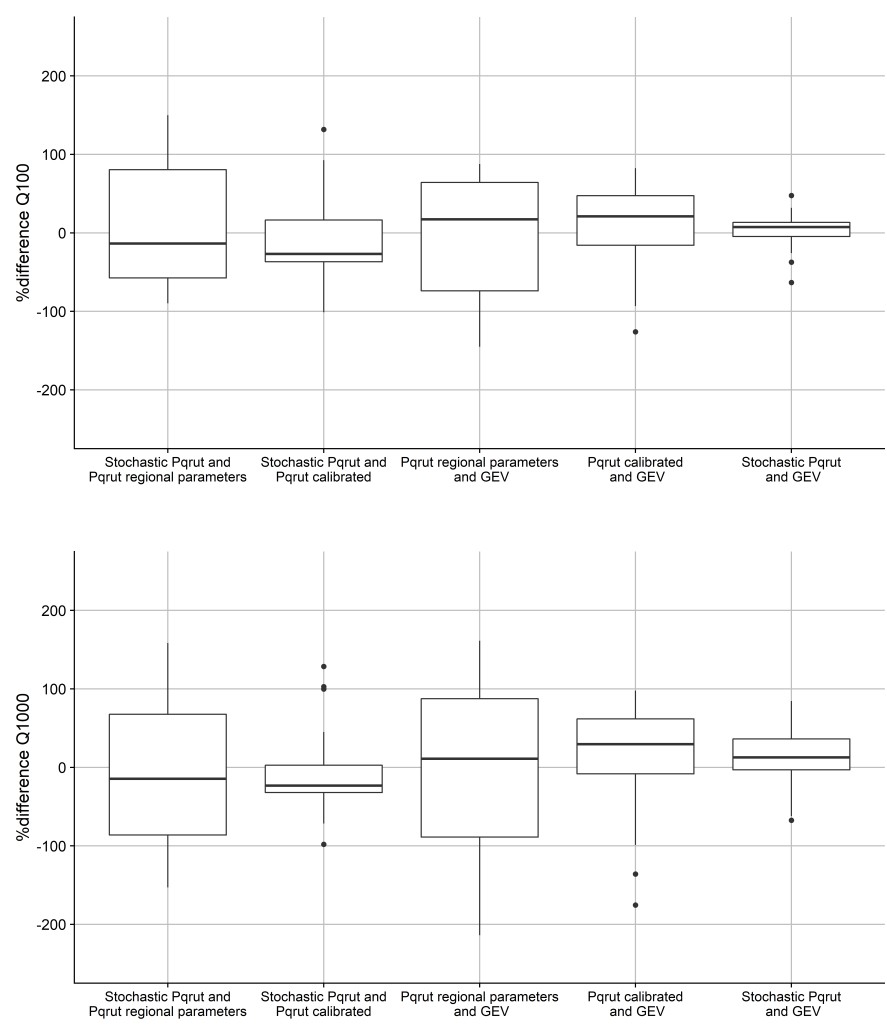

**Figure 12.** Boxplots showing the distribution of the differences (calculated by dividing the estimates by the average of all of the models) between the Stochastic PQRUT, PQRUT and GEV for the 100- and 1000 – year return level.

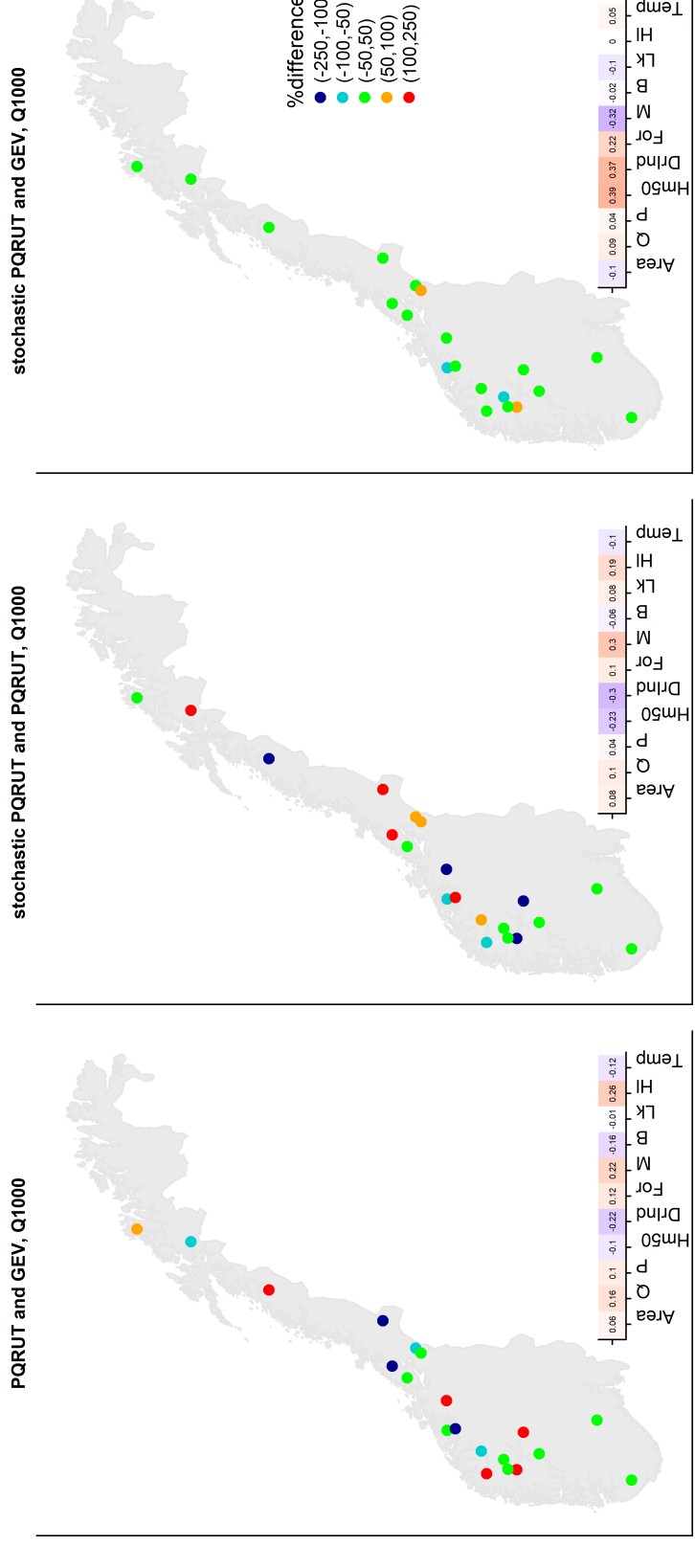

**Figure 13.** Results of the comparison between stochastic PQRUT, PQRUT and GEV for the values of the 1000-year return level. The absolute differences (calculated by dividing the estimate by the average of all models) are correlated with catchment properties. Positive correlations are given by red and negative by blue.

**Table 1.** Properties for the selected catchments, including *Area*-catchment area,*Q*-mean annual streamflow, *P*-mean annual precipitation, *M*-percent marsh,*B*-percent sparse vegetation over treeline,*Hl*-catchment steepness, *Lk*-effective lake percent and *Temp*-mean annual temperature

| Station | Area, km$^2$ | Q, mm/year | P,mm/year | Hm50,m | For,% | M,% | B,% | Lk,% | Hl, m/km | Temp,° C | Season of AMAX |
|---|---|---|---|---|---|---|---|---|---|---|---|
| Hørte | 157 | 961 | 1261 | 501 | 73 | 3 | 18 | 0.3 | 18.7 | 2.89 | SON |
| Krinsvatn | 207 | 1890 | 2354 | 348 | 20 | 9 | 57 | 1.1 | 5.4 | 4 | DJF |
| Øvrevatn | 526 | 1448 | 1558 | 564 | 35.2 | 2.5 | 52 | 0.6 | 14.8 | -0.14 | JJA |

**Table 2.** Precipitation (48 hour duration is used for Kristvatn) and initial conditions for *Q100*, range corresponds to 5 and 95 percentile. For the seasonality, the actual fraction of the simulated events is given in brackets.

| Station name | P median, mm | P range ,mm | Q median, m3/s | Q range, m3/s | Snowmelt median, mm | Snowmelt range, mm | Soil Moisture deficit median, mm | Soil Moisture deficit range, mm | Season |
|---|---|---|---|---|---|---|---|---|---|
| Krinsvatn | 140 | 107-217 | 14 | 4 - 47.1 | 0 | -31.2-18.2 | 25.5 | 5.3-50.4 | SON(0.5) DJF(0.4) |
| Hørte | 67 | 45-100.5 | 10 | 2.6- 27.1 | 0 | -37 - 22.7 | 7.8 | 1.1- 17.7 | SON (0.64) |
| Øvrevatn | 55.9 | 38-127 | 56.2 | 20.2-110.1 | 30 | 0 -49.8 | 13 | 2.6-44.2 | JJA(0.76) |

**Table 3.** Setup for the sensitivity analysis of PQRUT

| Variable | Standard model setup | Model set up for sensitivity analysis |
|---|---|---|
| Precipitation depth-threshold | threshold is selected between 0.93-0.99 quantile | GP fitted to 0.99 quantile |
| Precipitation depth-distribution | generated from GP distribution | Exponential distribution |
| Precipitation intensity | disaggregation using random historic storm events | divide into 24 equal parts |
| parameters of PQRUT | calibrated to selected storm events | use Latin Hypercube to sample 50 values within the 5% and 95% confidence intervals for the regression equations for K1, K2 and Trt. |
| length of simulation | 400 000 | Sequence of lengths were used staring at 40 000 to 400 000 by increment of 40 000 |
| Initial discharge | generated from multivariate normal distribution | use median discharge for each season |
| Snowmelt | snowmelt component (described in section 2.5) | No snowmelt component |
| Soil moisture deficit | generated from multivariate normal distribution | Fully saturated conditions Ip=0 (no initial loss) |
| effect of random seed | random seed is set at the start | 50 different runs using different random seeds |

**Table 4.** Percent difference between the model runs of the sensitivity analysis to the calibrated model

| Catchments | Hørte | Krinsvatn | Øvrevatn | Hørte | Krinsvatn | Øvrevatn | Hørte | Krinsvatn | Øvrevatn |
|---|---|---|---|---|---|---|---|---|---|
| setup\return period | Q10 | Q10 | Q10 | Q100 | Q100 | Q100 | Q1000 | Q1000 | Q1000 |
| GPD was fitted to 0.99 quantile | 10.0 | 13.5 | 2.4 | 16.5 | 1.4 | 17.0 | 22.2 | -11.3 | 48.1 |
| Exponential distribution instead of GP distribution | 3.0 | 3.0 | 0.9 | 9.1 | 3.1 | -8.5 | 18.1 | -2.0 | -22.6 |
| Disaggregate precipitation depth using uniform distribution (constant intensity) instead of using temporal patterns | -23.7 | -32.2 | -28.4 | -27.6 | -43.3 | -12.8 | -29.1 | -50.5 | -7.4 |
| 50 values, sampled using latin,hypercube within the 5% and 95% confidence intervals for the regression equations for K1,K2 and Trt (min value) | -36.7 | -46.8 | 7.1 | -35.3 | -54.4 | -5.9 | -36.7 | -58.2 | -2.9 |
| 50 values, sampled using latin,hypercube within the 5% and 95% confidence intervals for the regression equations for K1,K2 and Trt (max value) | -16.9 | 4.3 | 33.0 | -14.8 | -8.3 | 13.3 | -13.4 | -33.4 | 8.7 |
| different simulation length 40 000 to 400 000 simulations by 40 000 simulations (range) | 4.9%-7.3% | 1.6%-2.2% | -0.8% - 5.8% | 2.2%-3.5% | 0.4% - 2.3% | -4.6% - 6% | -4% - 5.4% | -0.1%-6.4% | -2% - 8.7% |
| median discharge instead of randomly generated | -7.3 | -2.4 | -31.3 | -6.0 | -1.6 | -13.1 | -5.0 | -1.2 | -1.4 |
| no snowmelt modelled | -2.3 | 14.8 | -18.7 | -0.5 | 17.3 | -4.2 | 2.9 | 21.2 | 1.2 |
| fully saturated conditions | 20.9 | 24.9 | 41.2 | 14.6 | 14.0 | 33.0 | 10.1 | 8.0 | 26.2 |
| 50 simulations with different random seeds (range) | -0.6% - 0.8% | -1.2% - 0.6% | 1.9%-3.5% | -1.3% - 0.9% | -0.2% - 3.5% | -0.6% - 3.3% | -5% - 2.6% | -3.9% -5.5% | -7.5% - 10.3% |