# Peer review of "A stochastic event-based approach for flood estimation in catchments with mixed rainfall/snowmelt flood regimes"

_Natural Hazards and Earth System Sciences, 2018_

## Referee Comment (RC1) · Anonymous Referee #1 · 9 Jul 2018

**General comments:**

This paper presents a significant enhancement of a Norwegian method for the estimation of extreme floods, based on an event-based rainfall-runoff simulation. It introduces a stochastic process for the assignment of the initial hydrological conditions before the simulated events, as well as for the intensity and the temporal dynamic of the simulated precipitation events. This method is compared to the initial method (which considers only a reference precipitation on given condition), and to a classical FFA.

The presented method is interesting, both in terms of methodology and statistical results. It is well explored, with a detailed sensitivity analysis.

However, the paper could be greatly improved by a better writing and more illustrations, particularly about the stochastic PQRUT which deserves a detailed step by step explanation of the simulation procedure (text and diagram), and also the probabilistic models for precipitation.

Regarding the sensitivity analysis, which is very important to understand the key factors and options of this new method, its writing also should be better organized and illustrated. It lacks a basic but important study of the impact of the random drawing (e.g. by performing 100 different simulations) and of the number of the simulated events on the extreme quantiles estimation. The later seems to be an issue here for high return periods.

I would recommend a significant revision of this paper, mostly to improve its structure, its writing and the illustrations provided. Detailed comments/suggestions are provided to the authors in that follows.
**Detailed comments/questions:**

Quoted sentences are written in italic.

**Abstract**

For those not familiar with PQRUT, it could be added in the abstract that the stochastic PQRUT is an extension/evolution of the "standard" PQRUT routine, applied since many years in Norway (dates and references to be provided).

The differences between the estimates can be up to 200% for some catchments, which highlights the uncertainty in these methods

This is not a good message for hydrological engineering, a less pessimistic phrasing could be " $[\ldots]$  200% for some catchments where the uncertainties of the compared methods are high and combine unfavourably".

**§1 - Introduction**

Page 1, line 14: For example, floods with a 500-year return period are sometimes used to  $[\dots]$

As most of the estimates evoked in the paper are 100 or 1000-yr. floods, and example of the use of such quantiles in Norway could useful.

Page 1, line 17: Flood mapping also usually requires input hydrographs

This is also the case for dam safety assessment.

Page 1, line 21: When longer return periods are needed

I guess the author means "longer than the record length", i.e. return period of 100 yr. or above.

Page 2, line 6: have been shown to produce average errors between 27 and 70%

Please mention on what estimation this error is computed (observed quantiles or esti-
mated ones, of which return period).

Page 2, line 32: they are computationally inefficient...

Another writing could be " [...] they are computationally demanding, as long continuous periods have to be simulated to estimate extreme quantiles".

Page 3, line 6: millions of rainfall events can be sampled from the MEWP model...

More exactly, millions of synthetic rainfall events can be generated, assigned to a probability estimated from the MEWP model, and inserted...

Page 3, line 21: it requires the generation of a temperature sequence for the event

I would add " a temperature sequence for the event, coherent with the simulated rainfall, and a snow water..."

Page 3, line 22: The assumption of a fixed rate of snowmelt [...] and a joint probability model needs to be considered

Does it mean that a fixed snowmelt is usually added without consideration to the rest of the variables which will characterize the simulated event? What kind of joint probability model should be added?

Page 24, line 24: SEFM which has been applied in several USGS studies

To my knowledge, I am not sure it is USGS (although SEFM is evoked in the USGS Bulletin 17C "Guidelines for Determining Flood Flow Frequency" of 2018), but several application of SEFM for dam safety studies have been delivered to USBR (US Bureau of Reclamation).

Page 3, line 34: due to the large uncertainty in both the event-based model and the statistical flood frequency analysis

I am not comfortable with this writing. With two identical methods (say a classical FFA), but with two distributions fitted on two different samples, estimations would be different,
and in that case this difference is completely linked to the uncertainties of the FFA (and mainly the sample uncertainty). But with different methods, these differences can also be produced by discrepancies between methods which should be treated *per se*, in order to assess the method themselves. So the interpretation of the difference should, in my opinion, not only rely on uncertainties.

Page 3, line 35: To better understand the differences between these methods, a sensitivity analysis of the stochastic PQRUT is performed

Here I have somehow the opposite comment from above: this sensitivity analysis of stochastic PQRUT is more about dealing with the uncertainties of stochastic PQRUT, not the differences between methods.

**§2 - Stochastic event-based model**

Page 4, line 18: The study area consists of a set of 20 catchments

A more logical phrasing could be "The study area in Norway, with a dataset of 20 catchments located throughout the whole country"

Page 5, line 6: for Krinsvatn, Lk and the area covered by marsh, M, is more than 10%

In the Table 1, Lk and M values are 9 and 1.1 %, respectively.

Page 5, line 28: the correlation of the method was found to be higher

To which values the results of this disaggregation method have been correlated? Hourly or 3-hourly rainfall observations?

Page 5, line 29: *simply dividing the seNorge data into eight equal parts*

To be clearer, I suggest "simply dividing the seNorge daily data into eight equal 3-hourly values".

Page 5, line 29: *disaggregated to a 1-hour time step using a uniform distribution to match the time resolution of the discharge data, although a 3-hour time step could also*

NHESSD
be used

I don't fully understand this. Was the 3-hour value affected randomnly to one hour or divided into three? Why is it possible to use the 3-hour value with an hourly model?

Page 6, line 1: remotely sensed data

Assimilating remotely sensed data in a hydrological model is not an easy task, especially soil moisture. Any reference to provide that would apply to the context of this paper?

Page 6, line 2: the DDD hydrological model was used

Please provide a reference which introduces this model.

Page 6, line 7: exceeds 0.3 of its (dynamic) capacity

Please define what is a dynamic capacity here.

Page 6, line 11: *the so-called critical duration*

A reference can be provided here to define this concept:

Meynink, W. J., & Cordery, I. (1976). Critical duration of rainfall for flood estimation. Water Resources Research, 12(6), 1209-1214.

Page 6, lines 12-14: In order to determine [...] had to be determined for each catchment

I don't find this sentence useful, considering what is written before and after it.

Page 6, lines 14: flood events over a certain quantile threshold (0.9) were extracted.

On what data this POT sample was extracted: daily or hourly discharges?

Page 6, lines 18-25:

An alternative to this could be to study the correlation between the peak daily value and
the precipitation of that day (which we could call P0), the sum P0 to P-1, P0 to P-2 and so on... When the correlation coefficient stops to increase significantly, it means that the correct length of the "precipitation window" is reached, thus the critical duration is estimated. This is likely to be more robust than studying the correlation between the peak daily discharge and the individual precipitations the days before.

Page 6, lines 22-24: In some catchments (mostly those having snowmelt flood regime), no significant correlation was found between discharge and precipitation

In that case, some processing of the flood is needed, e.g. only considering the "snowfree" seasons, or adding a threshold on the precipitation over the preceding days in the POT selection of floods. This could prevent using an arbitrary duration.

Page 6, line 28: the sequence of the input data must be prescribed for the stochastic simulation

What means "prescribed"? Is it generated? Is it randomly drawn from the observed sequences?

Page 7, line 1: a Generalized Pareto distribution was fitted to the series of selected events

A figure with the corresponding fits and observations for the example catchments would be welcome.

Page 7, line 6: *introduced in section ??*

A paragraph number is lacking.

Page 7, line 7: Using the fitted Generalized Pareto (GP) distribution, precipitation depths were simulated

Does it mean that probabilities where randomly drawn then the corresponding precipitation depths deduced from the fitted GPD? How many events are drawn?
Page 7, line 8: a storm hyetograph was first sampled

How is it sampled? I guess it comes from the hyetographs collection corresponding to the POT selection of precipitation events, but with what consideration to season, intensity, etc.?

Page 7, line 10:

Pi and P are not defined in the disaggregation formula.

Page 7, line 15: Output from DDD model runs

Have the DDD models been calibrated on local data? If yes, some words about the calibration method are welcome. I guess the DDD models are used at daily time-step, is it true?

Page 7, lines 21-25:

The writing is not clear, and neither is the equation of the mixed distribution (what is x?). As far as I understood, it is about randomly switching between a trivariate (discharge, moisture, snow) and a bivariate (discharge, moisture) distribution depending on the probability of having snow on a given season. Is *p* also drawn for the simulation?

Page 7, line 26: The correlation between the observed and simulated variables is shown in Figure 4

Apparently, *sl* is the soil moisture deficit. Contrary to SWE and Qobs which are "observable" variables, *sl* is linked to a model (here DDD). So it should be introduced, in relation with the DDD model structure.

Page 8, line 5: for estimating design floods and safety check floods for dams in Norway

This type of application is perhaps documented in (Andersen, 1983), but this reference is not easily accessible on line, and is written in Norwegian, so a accessible reference documenting this type of application would be welcome.
Page 8, line 14: The general procedures used for the PQRUT calibration are described in Filipova et al.

Some details about this calibration would be welcome, e.g. which flood events sample is considered (is it the same as the one used in §2.2 for critical duration)?

Page 8, lines 15-17: This additional parameter *lp* should be documented in the structure of the PQRUT model presented in the Figure 5. Furthermore, I am not sure that it can be considered as a parameter, more likely it is an internal state variable which vary from event to event.

Page 8, line 18: the value of this parameter was set to the initial soil moisture deficit, estimated using DDD

This is an important assumption: it means that some internal variables of DDD (which ones, this is not documented) are used to estimate another one in PQRUT. This is far from obvious to accept for two very different models, running at different time steps: what has be done to check this "compatibility"?

Page 8, line 23: *Cs is a coefficient accounting for the relation between temperature and snowmelt*

Properties

It is usually called a "degree-day" coefficient (although used at a hourly time step here).

Page 8, line 30: The term under the bar should be "power to k" not be multiplied by k.

Page 9, line3: These simulated events were compared with the POT flood events extracted from the observations

At this point, I don't clearly understand the simulation process. Some lines detailing the simulation process (sequence of random drawings, number of simulation, processing of events, etc.), as well as a diagram, are really necessary to the reader before entering into the analysis of the simulations.

NHESSD
The simulated CDFs look affected by under-sampling above the 500 yr. return period (i.e. not enough simulations of this range), which interrogates the robustness of the 1000 yr. estimations which are assessed in the paper.

Page 9, line 7: large variation in precipitation values

Which duration is considered here? Daily?

The comparison to the 100 yr. precipitation depths estimated thanks to the GP fit evoked in §2.3 would be useful.

Page 9, line 14: even though fully saturated conditions are used in the event-based PQRUT model

I don't understand this: the *lp* variable (variable initial loss) has been introduced in §2.5 to depart from this fully saturated hypothesis.

Page 9, line 16: A sensitivity analysis was performed for the three test catchments

Once again, the detailed protocol of this analysis deserves to be presented for a better understanding of the results. Some information is given in Table 3 but would deserve to be detailed in the text. A more logical "progression" of the different setups could be: 2, 3 (statistical hypothesis on precipitation), then 4 (temporal disaggregation), then 5,6,7 (simple hydrological assumptions) and finally 1 (PQRUT parameters). This would apply for the Table 3, as well as for the writing of §2.7.

Page 9, lines 24-28:

I am not fully convinced by this explanation based on BFI. The sensitivity to initial loss should be linked to the possible values of initial loss in relation with the high quantiles of precipitation. I would be interested by looking at those values (maximum initial loss and 10, 100 and 1000 yr. precipitation) for the three catchments.

Page 9, line 31: In addition, Krinsvatn shows high sensitivity to snowmelt

NHESSD
This is in contradiction with Page 9, line 10 (*for Krinsvatn* [...] *in most cases snowmelt does not contribute to the extreme floods*). Any comment?

Page 10, line 7: Ovrevatn and Horte showed sensitivity (28.9%) to the choice of the

statistical distribution for modelling precipitation

A figure showing the precipitation distribution for each catchment (both observed and modelled by GP, and EXP) would be welcome to illustrate lines 7 to 10.

**§3- Comparison with standard methods**

Page 10, line 27: the standard implementation of the event-based PQRUT method

This is the first mention of such a "standard" implementation. I think this would deserve to be presented at the very beginning of the paper, which proposes a "stochastic PQRUT" being a significant enhancement from the "standard PQRUT". The context of this study would thus be better understood.

Page 10, line 29: the annual maximum series were extracted from the observed daily mean streamflow series

Why not using a GPD with the POT sample of floods extracted for the study of the critical duration?

Page 10, line 31: to obtain instantaneous peak values, the return values were multiplied by empirical ratios, obtained from regression equations

Here I don't understand why the POT flood events extracted from observations (shown in the plots of the Figure 6) has not been used to fit either a GPD, or a GEV after extraction of annual maxima. More comments about this would be welcome.

Page 11, line 11: obtained from growth curves based on the 5-year return period value

If I understand properly, the shape of the design hyetograph is based on the growth curves considered at the 5 yr. return period. Are the ratios between the different
duration values at this return period deduced from empirical distribution, or inferred from a fitted distribution? Later on, this must be scaled to define a 100 or 1000 yr. hyetograph. What precipitation distribution (duration and model) are these extreme values deduced from?

Page 11, line 15: The performance of the three models was validated by using two different tests

In that case, dealing with 100 or 1000 yr. flood estimations, it's more about "comparing different approaches".

Page 11, line 20: As discussed, due to the difficulty in assigning initial conditions for the event-based

PQRUT model

I don't understand this sentence, and to which discussion it refers.

Page 11, line 22: the regional equations were used

Which regional equations? For PQRUT parameters?

Page 11, line 25: equation of QS + *observed probabilities (Qobsi) are calculated using Gringorten positions for the POT series*

The POT series are used here, contrary to the daily (transposed to peak) annual maximum values that have been fitted in the statistical approach. Another option (already mentioned in my remark for page 10, line 29) could be to fit the statistical method on the POT sample, which would have allowed to keep it as a "benchmark" method, given more sense to the comparison presented (or conversely using the "peak-from-daily" observations for the QS calculation).

Page 11, line 30: the results vary between catchments as shown in fig 8

I don't find this figure very useful, the reader is unable to interpret the coloured dots.
An alternative, aside the boxplots, could be some scatter plots (statistical Q100 and Q1000 v/s standard and stochastic PQRUT, statistical QS v/s standard and stochastic PQRUT, etc.).

Page 11, line 32: we can conclude that the performance of the standard PQRUT model is poorer than the performance of the statistical flood frequency analysis and the stochastic PQRUT model

The results which ground this conclusion are not explicitly presented. The only clue given to the reader is the Figure 8 which only presents the distribution of QS scores for FFA and stochastic PQRUT. The results, in terms of QS score as well as confidence interval, should be presented in a table and in an adequate figure.

Page 12, line 3: The violin plots (fig. 9)

See remarks on Figure 9 below.

Page 12, line 7: *Reasons for this may be that higher precipitation intensity or snowmelt is used*

To assess this, the values of the reference hyetographs used in standard PQRUT deserve to be presented and compared to the simulated precipitation values of stochastic PQRUT (like the values of the Table 2 for Q100).

Page 12, line 8: the absolute differences between the two methods are larger in catchments with lower temperature (fig. 9)

I wonder how this can be deduced of illustrated by Figure 9, it is more likely somehow in Figure 10.

Page 12, line 17: which might be due to the uncertainty in estimating the parameters for the GEV distribution

I don't understand this interpretation which appears rather quick and subjective to me.
Page 12, lines 18-21:

This using of the study of Rogger et al. (2012) is off topic for me here, as it is based on Gumbel, whereas the FFA is done here with GEV, which is more flexible.

**§4- Conclusions**

Page 13, line 10-15:

Another modelling option could be to run the event-based simulation with the DDD model, already used for the initial condition. In that case, an hourly version of DDD should also be calibrated (with local observations or regionally), in compatibility with the daily version used for initial conditions. I am not fully aware of the potential difficulties of this, but it would be a more homogeneous approach in terms of hydrological modelling. Any comment about this?

Page 13, line 28: easily incorporate the uncertainty associated with this choice

This is a very good remark: the stochastic process here adequately models a variable which, when represented in a deterministic way (i.e. fixed initial conditions), appears as highly uncertain.

Page 13, line 31: based on an assessment of the uncertainty characterizing the individual methods

This is an interesting suggestion, but it has to be added that a proper expression of uncertainty for a rather sophisticated method like stochastic PQRUT is far from trivial, and is still to be investigated...

**Tables**

Table 1

Units are missing, as well as legend of the columns in the caption.

Table 2:
Units are missing, precipitations could be rounded to the next mm.

For Krinsvatn, the probability to find the Q100 events in one season or the other could be provided.

**Figures**

Figure 3:

Not very informative, re-scaling storm hyetograph is not something difficult to understand. A set of different "typical" hyetographs could instead be presented for the three catchments, ideally illustrating the potential diversity of storm dynamic.

Figure 4:

"for Krinsvatn catchment" could be added in the caption, as well as the number of observed and simulated events.

Figure 6:

The remarkable return periods (10, 100 and 1000 yr.) should be distinguished in the plots (by a bolder vertical line for example).

Figure 7:

There are too many distributions in the plots, their interpretation is not easy. Two plots could be edited for each catchment, having for example only the "calibrated" simulation in common.

An uncertainty band around the "calibrated" simulation would be useful to assess the intrinsic uncertainty of the simulation process.

Figure 8:

See comment of page 11, line 32.

Figure 9:
I am not convinced by the usefulness of the violin plots here considering the limited number of values per scores (20 catchments). Box plots with outliers would have been sufficient and more readable.

Captions of the methods sometimes overlap.

---

## Referee Comment (RC2) · Anonymous Referee #2 · 12 Jul 2018

article  This paper propose a stochastic event-based method for design flood estimation in Norway.  The presented approach is described and the results are compared to a flood frequency analysis and a commonly applied design storm approach based on a single design storm.  The methods are applied in 20 small to meso-scale catchments across Norway to estimate the 100 and 1000 year return period event.  Furthermore, a sensitivity analysis of the model parameters, initial conditions and precipitation input was performed to determine their importance to the resulting flood frequency curve. The results show large differences between the applied methods while the sensitivity analysis shows the importance of the investigated aspects, which helps to explain the large differences.

The study addresses an important topic, which is relevant also for practical applications, as rather subjective user assumptions are overcome by the stochastic approach. The manuscript fits well into the scope of the journal. A good review on the subject is given at the beginning, however, the proposed stochastic event-based method itself should be explained in a more structured way and the writing could be improved, especially in the second part of the manuscript. Furthermore the presentation and description of the results could be enhanced, as they are sometimes not clear for the reader. There is room for improvement at certain points:

**General Comments:**

As I understood, the main purpose of the work is to propose a methodology to overcome the limitations of more commonly applied event based modelling for flood frequency estimations by a stochastic modelling of preconditions, including SWE, and meteorological input. The individual modelling of the different aspects are described in the manuscript, however, it is hard to follow how the different parts are connected. A preceding sub-section with a less detailed step by step explanation of methodology, maybe including a schematic illustration (inputs/ models/ methods / output), could help to better explain the methodology.

For the validation of the disaggregation procedure the disaggregated data were compared against hourly station data. Is this correct? I would be interesting to see how well the disaggregation procedure was performing (For example showing a obs-sim, QQ-plot). It is stated that it works better than equal divisions which is not surprising. What is the advantage of the further equal division to 1h if it is stated that 3-houers are already enough? Further, it is not obvious why the gridded seNorge.no Data are matched to the HIRLAM data if they are in the needed temporal resolution already?

A 1000 years event is extrapolated from daily observation series (length not further

specified). Furthermore the results are then multiplied by empirical factors, to match sub-daily peak flows. I am not aware of the engineering practice in Norway, however, I am not sure about the meaning of the results by this extreme extrapolation and at least this should be critically discussed.

The sensitivity analysis is interesting, however, also confusing including Figure 7 and Table 3. It is not obvious on what basis the percentage difference is calculated. This is also not clear in the follow up comparison of the methods. What exactly is the calibrated model? Also the section misses an explanation of the shaded area which is prominently displayed in Figure 7. Furthermore, the different precipitations settings tested are not well explained. A table, summarizing the different tested aspects, would help to guide the reader.

The Figures and especially the captions should be improved, as they are often not self-explanatory. This includes also missing units, labels and abbreviations.

Maybe consider a professional language proof reading.

**Specific Comments:**

The abbreviation PQRUT, used from beginning (abstract), is not introduced on page 4 or rather page 8. Please declare the meaning of PQRUT first time mentioned.

The characterizations of catchments and chosen abbreviations are introduced on P4 and repeated later (P5, l5) without brackets (e.g. "sparse vegetation over tree line (B)" and "sparse vegetation over tree line B"). Either use brackets throughout the manuscript or only use the abbreviation. Additionally by choosing more self-explanatory abbreviations or using full words (eg. forest; marsh), would be easier to understand, especially in Table 1.

P.5 l.15: The last sentence does not contain important informations and could be omitted

P.6 l.1: The addition ",which can be used for modelling in ungagged basins." could be omitted, as it seems not connected to the procedure.

P.6 l.2: A citation should be added to the DDD model or the corresponding R-package.

P.6 l.11: In my opinion, the meaning of the "critical duration" rather reflects the link between the duration and intensity of precipitation "events" of a certain probability, than to ensure the modelling of the complete flood hydrograph.

P.6 l.32: "individual risk seasons could have been defined". One wonders why it was not done? If not so important for the result, please consider to omit this half sentence.

P.8 l.12-17: Please check grammar and style of the section.

P.8 l.28: Was there a specific reason for using the "Gringorten plotting" position?

P.12 l.3: A more detailed explanation what exactly is analyzed here is missing.

P.12 l.25: Maybe the catchment steepness should be introduced in the section "study area".

P.13 l.25: Why is it peak to volume? I thought it is daily mean to daily max discharge?

Table 1: Missing units. Furthermore, the variables could be sorted and clustered more logically (e.g. temperature and precipitation; Q and AMAX).

Table 3: Why are 100 values sampled? Does T mean the threshold Parameter Trt ?

Figure 2: Labels and units are missing

Figure 3: Labels and units are missing

Figure 7: is confusing because of the large number of different colored lines. Maybe two plots can help to distinguish between the different aspects as for example the precipitation input and other aspects. The legend is confusing as well. GDP was fitted

to what? Y-Axes should start at 0, x-axes missing a label and to be consistent with the rest of the work it should not exceed 1000.

Figure 8: It is impossible to distinguish between 20 colors. Do the colors have any meaning? If they should be recognizable, numbering would be a better option. The numbers could then also be used in Figure 10, so the link between the performance of the model and the results are given.

Figure 9: What exactly is shown in the plots. Please add a more detailed explanation.

Figure 10: The scale "percentage difference" should be unambiguous. The base of the "difference" should be clarified.

**Technical Notes:**

"Figure" and "Table" should start with capitals

Please use the degree symbol e.g. $4^\circ$ C (P.5 l.10, P8 l23 +26,...)

Please use [mm year-1] instead of [mm / year]

P.3 l.22: grammar ", as is often used"

P.5 l.3: Subscript i in ai and Ai ($a_i$ and $A_i$)

P.7 l.6: Missing link to "section"

P.7 l.20: Typo: "multivariat" instead of "mul- tivariat"

P.7 l.23: Whitespace, "values,p,"

P.9 l.1: Whitespace, "where,P"

P.9 l.34: Whitespace, "(29

P.11 l.3: Whitespace, "GL(Generalised..."

P.11 l.10: Typo, "mm//°C"

Check citation "Beven, Keith, . . . 2014"

Check citation "Chow, . . . 1988"

Check citation "Fleig, . . . 2013"

---

## Author Comment (AC1) · 16 Aug 2018

This paper presents a significant enhancement of a Norwegian method for the estimation of extreme floods, based on an event-based rainfall-runoff simulation. It introduces a stochastic process for the assignment of the initial hydrological conditions before the simulated events, as well as for the intensity and the temporal dynamic of the simulated precipitation events. This method is compared to the initial method (which considers only a reference precipitation on given condition), and to a classical FFA. The presented method is interesting, both in terms of methodology and statistical results. It is well explored, with a

[Figure]

**detailed sensitivity analysis.**

We thank the reviewer for the positive comment on the method and the very detailed review of the manuscript.

**However, the paper could be greatly improved by a better writing and more illustrations, particularly about the stochastic PQRUT which deserves a detailed step by step explanation of the simulation procedure (text and diagram), and also the probabilistic models for precipitation.**

We agree that the explanation of the procedure can be improved. In the revised version, we will add a diagram which illustrates how to apply the simulation procedure.

**Regarding the sensitivity analysis, which is very important to understand the key factors and options of this new method, its writing also should be better organized and illustrated. It lacks a basic but important study of the impact of the random drawing (e.g. by performing 100 different simulations) and of the number of the simulated events on the extreme quantiles estimation. The later seems to be an issue here for high return periods.**

These two issues are somewhat related. The effect of the random seed will be minimised if larger number of simulations are used. In addition, increasing the number of simulations will increase the robustness of the extreme quantiles. However, the number of simulations that are needed will depend on the return period of interest. Here we have considered a long return period, 1000-years, so this issue is indeed relevant. In the revised version, we plan to include two additional variables in the sensitivity analysis:

[Figure]

- 100 simulations using different random seeds

- Length of simulation, of up to 100,000 (in increments of 10,000). It is not feasible to consider longer simulations due to computational times. But this results and the possible need for longer simulations will be discussed in the revised text, based on the results of the sensitivity analysis.

**I would recommend a significant revision of this paper, mostly to improve its structure, its writing and the illustrations provided. Detailed comments/suggestions are provided to the authors in that follows.**

We hope that our revisions will significantly improve the writing and the structure of the paper.

**Abstract For those not familiar with PQRUT, it could be added in the abstract that the stochastic PQRUT is an extension/evolution of the " standard" PQRUT routine, applied since many years in Norway (dates and references to be provided). The differences between the estimates can be up to 200% for some catchments, which highlights the uncertainty in these methods This is not a good message for hydrological engineering, a less pessimistic phrasing could be "[. . . ] 200% for some catchments where the uncertainties of the compared methods are high and combine unfavourably".**

The reason, we did not include this information is that we wanted to emphasise that the method can be applied to any catchment with snowmelt/mixed flood regime. But as our study area is in Norway, we will revise the abstract as:

*Traditionally, statistical flood frequency analysis and an event-based model (PQRUT) using a single design storm have been applied in Norway (Midttømme and Pettersson,*

*2011). We here propose a stochastic PQRUT model, as an extension of the standard application of the event-based PQRUT model, by considering different combinations of initial conditions, rainfall and snowmelt, from which a distribution of flood peaks can be constructed. . . .*

*The differences between the estimates can be up to 200% for some catchments where the uncertainties of the compared methods are high and combine unfavourably.*

**§1 – Introduction**

**Page 1, line 14: For example, floods with a 500-year return period are sometimes used to [. . . ] As most of the estimates evoked in the paper are 100 or 1000-yr. floods, and example of the use of such quantiles in Norway could useful.**

We will revise the paragraph as follows:

*The estimation of low-probability floods is required for the design of high-risk structures such as dams, bridges, levees, etc. For example, floods with a 100-year return period are sometimes required for the design of levees, the design and safety evaluation of high-risk dams requires the estimation of flood hydrographs for the 1000-year return period and, in some cases, floods with magnitudes of up to the Probable Maximum Flood (PMF).*

**Page 1, line 17: Flood mapping also usually requires input hydrographs This is also the case for dam safety assessment.**

This sentence will be revised -See answer above

**Page 1, line 21: When longer return periods are needed I guess the author means " longer than the record length", i.e. return period of 100 yr. or above.**

Yes, this is correct. For clarity, this will be changed to:

*When return periods that are longer than the observed record length are needed*

**Page 2, line 6: have been shown to produce average errors between 27 and 70% Please mention on what estimation this error is computed (observed quantiles or estimated ones, of which return period).**

These errors are based on table 1 in the paper by Salinas et al 2013, which provides a comparative assessment of different studies in ungauged basins. The values are calculated using the RMSNE (root mean square normalised error) for Q100 (q100) and we have expressed them as percentage. We will revise as:

*As the physical processes in the catchments are not directly considered in the analysis, estimating the flood quantiles in ungauged basins using regression or geostatistical methods have been shown to produce average RMSNE (root mean square normalised error) between 27 and 70% (Salinas et al 2013), or even higher, for Q100.*

**Page 2, line 32: they are computationally inefficient. . . Another writing could be "[. . . ] they are computationally demanding, as long continuous periods have to be simulated to estimate extreme quantiles".**

Yes, this is in fact a better way to write this.

**Page 3, line 6: millions of rainfall events can be sampled from the MEWP model. . . More exactly, millions of synthetic rainfall events can be generated, assigned to a probability estimated from the MEWP model, and inserted. . .**

Yes, we agree with the revision.

**Page 3, line 21: it requires the generation of a temperature sequence for the event I would add " a temperature sequence for the event, coherent with the simulated rainfall, and a snow water. . . "**

We will include this in the revised version of the manuscript.

**Page 3, line 22: The assumption of a fixed rate of snowmelt [. . . ] and a joint probability model needs to be considered Does it mean that a fixed snowmelt is usually added without consideration to the rest of the variables which will characterize the simulated event? What kind of joint probability model should be added?**

To increase the clarity, these sentences will be rewritten as:

*The assumption of a fixed rate of snowmelt which is based on typical temperatures, as is often used in Norway for the single event-based design method, can introduce bias in the estimates. The joint probability of both rainfall and snowmelt needs to be considered to obtain a probability neutral value (Nathan and Bowles, 1997).*

**Page 24, line 24: SEFM which has been applied in several USGS studies To my knowledge, I am not sure it is USGS (although SEFM is evoked in the USGS Bulletin 17C " Guidelines for Determining Flood Flow Frequency" of 2018), but several application of SEFM for dam safety studies have been delivered to USBR (US Bureau of Reclamation).**

Yes, it should be USBR

**Page 3, line 34: due to the large uncertainty in both the event-based model and the statistical flood frequency analysis I am not comfortable with this writing. With two identical methods (say a classical FFA), but with two distributions fitted on two different samples, estimations would be different, and in that case this difference is completely linked to the uncertainties of the FFA (and mainly the sample uncertainty). But with different methods, these differences can also be produced by discrepancies between methods which should be treated per se, in order to assess the method themselves. So the interpretation of the difference should, in my opinion, not only rely on uncertainties.**

In this case, uncertainty refers to errors due to random errors (aleatory) and to epistemic uncertainty which can be caused by lack of knowledge about the system. For example, the epistemic uncertainty can refer to the choice of statistical distribution for the FFA and for the event-based model - parameter calibration, assumptions about initial conditions. This will be revised as:

*large uncertainty and model assumptions in both the event-based model and the statistical flood frequency analysis procedures*

**Page 3, line 35: To better understand the differences between these methods, a sensitivity analysis of the stochastic PQRUT is performed Here I have somehow the opposite comment from above: this sensitivity analysis of stochastic PQRUT is more about dealing with the uncertainties of stochastic PQRUT, not the differences between methods.**

This is true, the sensitivity analysis will provide a better understanding of the uncertainty. However, the sensitivity analysis can also help us determine the reasons for the differences between the models. For example, the standard PQRUT assumes that fully

saturated conditions are used. By testing the sensitivity of the model to the initial soil moisture deficit, we can check whether assuming fully saturated conditions contributes to the difference between these two methods. We will revise this section as: *In order to understand the uncertainty and the differences with the standard PQRUT model, a sensitivity analysis of the stochastic PQRUT is performed by considering the effect of the initial conditions, model parameters and rainfall intensity on the flood frequency curve.*

**§2 - Stochastic event-based model Page 4, line 18: The study area consists of a set of 20 catchments A more logical phrasing could be " The study area in Norway, with a dataset of 20 catchments located throughout the whole country"**

We will make this revision.

**Page 5, line 6: for Krinsvatn, Lk and the area covered by marsh, M, is more than 10In the Table 1, Lk and M values are 9 and 1.1 %, respectively.**

This is correct, we will revise as:

*for Krinsvatn, the area covered by either Lk or marsh, M, is in total more than 10%*

**Page 5, line 28: the correlation of the method was found to be higher To which values the results of this disaggregation method have been correlated? Hourly or 3-hourly rainfall observations?**

The disaggregated data were compared to the 3-hour observations in the study undertaken by Vormoor and Skaugen to produce the gridded disaggregated data.

**Page 5, line 29: simply dividing the seNorge data into eight equal parts To be clearer, I suggest " simply dividing the seNorge daily data into eight equal 3-hourly values".**

This will be revised as suggested.

**Page 5, line 29: disaggregated to a 1-hour time step using a uniform distribution to match the time resolution of the discharge data, although a 3-hour time step could also be used I don't fully understand this. Was the 3-hour value affected randomly to one hour or divided into three? Why is it possible to use the 3-hour value with an hourly model?**

The rainfall data was then divided into three equal parts, i.e. using a uniform distribution. The PQRUT model can be used with any timestep. Considering that the median catchment size is around 140 km2, a timestep of 3 hours shoud be sufficient for modelling the peak flows.
As the model has previously been calibrated and regionalised using 1 -hour timestep, we decided to use this timestep (instead of 3-hours). A similar comment was raised by reviewer 2 and the section will be revised to include these suggestions as well.

**Page 6, line 1: remotely sensed data Assimilating remotely sensed data in a hydrological model is not an easy task, especially soil moisture. Any reference to provide that would apply to the context of this paper?**

Using the remotely-sensed data currently available is probably not ideal as these data have a coarse spatial resolution (around 20km2). A review of the use of this data is given in Brocca et al (2017). In addition, Sunwoo and Choi (2017) show that remotely

sensed data can improve the predictions of the SCS-CN model. These references will be provided in the revised version.

**Page 6, line 2: the DDD hydrological model was used Please provide a reference which introduces this model.**

Answer: The reference to the DDD model comes earlier in the manuscript, i.e. on p 4.

**Page 6, line 7: exceeds 0.3 of its (dynamic) capacity Please define what is a dynamic capacity here.**
Answer:

In this case, dynamic refers to the concept that the volume of the saturated zone and unsaturated varies in time as explained in the previous sentence. It is probably best to delete this as it is not necessary.

**Page 6, line 11: the so-called critical duration**

**A reference can be provided here to define this concept: Meynink, W. J., Cordery, I. (1976). Critical duration of rainfall for flood estimation. Water Resources Research, 12(6), 1209-1214.**
Answer:

Thanks for this, we will include this citation.

**Page 6, lines 12-14: In order to determine [. . . ] had to be determined for each catchment I don't find this sentence useful, considering what is written before and after it.**
Answer:

We will delete this sentence.

**Page 6, lines 14: flood events over a certain quantile threshold (0.9) were extracted. On what data this POT sample was extracted: daily or hourly discharges?**
Answer:

The sample is based on daily discharge, we will specify the timestep in the revised version.

**Page 6, lines 18-25: An alternative to this could be to study the correlation between the peak daily value and the precipitation of that day (which we could call P0), the sum P0 to P-1, P0 to P-2 and so on. . . When the correlation coefficient stops to increase significantly, it means that the correct length of the " precipitation window" is reached, thus the critical duration is estimated. This is likely to be more robust than studying the correlation between the peak daily discharge and the individual precipitations the days before.**
Answer:

This is an interesting suggestion. In this study, it is important to specify a critical duration in order to capture the "full" rainfall event producing the peak flow (otherwise the peak flow is likely to be underestimated).
We implemented the procedure proposed by reviewer but we did not find much difference between the two methods. In fact, this alternative gives shorter critical durations in some cases (i.e. 1-day for Krinsvatn instead of 48 hours). As our study area consists of small catchments and the shortest window we are using is 1-day, the critical duration is then also 1 day (for 17 out of 20 catchments), rather than being longer.

**Page 6, lines 22-24: In some catchments (mostly those having a snowmelt flood regime), no significant correlation was found between discharge and precipitation In that case, some processing of the flood is needed, e.g. only considering the " snowfree" seasons, or adding a threshold on the precipitation over the pre-**

**ceding days in the POT selection of floods. This could prevent using an arbitrary duration.**
Answer:

Only the snow-free events will be considered.

**Page 6, line 28: the sequence of the input data must be prescribed for the stochastic simulation What means " prescribed"? Is it generated? Is it randomly drawn from the observed sequences?**
Answer:

Yes, we will use "generated" instead.

**Page 7, line 1: a Generalized Pareto distribution was fitted to the series of selected Events A figure with the corresponding fits and observations for the example catchments would be welcome.**
Answer:

We will add a return level plot that shows the fit to the observations for both GP and Exponential distributions.

**Page 7, line 6: introduced in section ??**
Answer:

Section 2.1.2, the reference will be corrected.

**Page 7, line 7: Using the fitted Generalized Pareto (GP) distribution, precipitation depths were simulated Does it mean that probabilities where randomly drawn then the corresponding precipitation depths deduced from the fitted GPD? How many events are drawn?**
Answer:

This is correct, the sentence will be rewritten as:

*The precipitation depths were generated from the fitted GP distribution for each season with 100 000 events. Originally, 40 000 simulations (around 10 000 to 20 000 years) were used, we will now increase this number to 100 000 events.*

**Page 7, line 8: a storm hyetograph was first sampled How is it sampled? I guess it comes from the hyetographs collection corresponding to the POT selection of precipitation events, but with what consideration to season, intensity, etc.?**
Answer:

Yes, it was sampled from the collection of hyetographs, the seasonality was considered but not the precipitation depth. The following revision will be made: *a storm hyetograph was first sampled from the extracted hyetographs for the selected POT events, taking into account seasonality*

**Page 7, line 10: Pi and P are not defined in the disaggregation formula.**
Answer:

We will revise as:

*where Phsim is the simulated 1-hour precipitation intensity, Pdsim is the simulated daily intensity and Pi is the 1 -hour disaggregated SeNorge intensity*

**Page 7, line 15: Output from DDD model runs Have the DDD models been calibrated on local data? If yes, some words about the calibration method are welcome. I guess the DDD models are used at daily time-step, is it true?**
Answer:

Yes, a daily time step was used. The following sentence will be added to the text: *The model was calibrated for the selected catchments at daily timestep using MCMC*

*optimisation routine.*

**Page 7, lines 21-25: The writing is not clear, and neither is the equation of the mixed distribution (what is x ?). As far as I understood, it is about randomly switching between a trivariate (discharge, moisture, snow) and a bivariate (discharge, moisture) distribution depending on the probability of having snow on a given season. Is p also drawn for the simulation?**
Answer:

We agree that this was not clearly described, we will revise this section as follows:

*The probability p for switching between a trivariate and a bivariate distribution is based on the historical data for assessing the probability of SWE higher than 0.*

**Page 7, line 26: The correlation between the observed and simulated variables is shown in Figure 4 Apparently, sl is the soil moisture deficit. Contrary to SWE and Qobs which are " observable" variables, sl is linked to a model (here DDD). So it should be introduced, in relation with the DDD model structure.**
Answer:

The soil moisture deficit is presented in Skaugen and Onof (2013). The soil moisture deficit is the difference between the volume of the unsaturated zone and the volume already present in the soil moisture zone. We will include more information in the revised version:

*It is important to note that simulated values for soil moisture deficit are used however as described in Skaugen and Onof (2013), the model provides realistic values in comparison to measured groundwater levels.*

**Page 8, line 5: for estimating design floods and safety check floods for dams in Norway This type of application is perhaps documented in (Andersen,**

1983), but this reference is not easily accessible on line, and is written in Norwegian, so a accessible reference documenting this type of application would be welcome.
Answer:

A reference to the NVE report will be provided:

*The PQRUT model was used to simulate the streamflow for the selected storm events. The PQRUT model is a simple, event-based, 3-parameter model (fig 5) which is used, amongst other things, for estimating design floods and safety check floods for dams in Norway (Wilson et al. 2011).*

**Page 8, line 14: The general procedures used for the PQRUT calibration are described in Filipova et al. Some details about this calibration would be welcome, e.g. which flood events sample is considered (is it the same as the one used in §2.2 for critical duration)?**
Answer:

In the calibration, the 45 highest flood events were considered. This sample most likely overlaps with the events selected for the critical duration.

**Page 8, lines 15-17: This additional parameter lp should be documented in the structure of the PQRUT model presented in the Figure 5. Furthermore, I am not sure that it can be considered as a parameter, more likely it is an internal state variable which vary from event to event.**
Answer:

We agree in this case, lp can be considered as a state variable. The figure will be updated to include lp.

**Page 8, line 18: the value of this parameter was set to the initial soil moisture deficit, estimated using DDD This is an important assumption: it means that**

**some internal variables of DDD (which ones, this is not documented) are used to estimate another one in PQRUT. This is far from obvious to accept for two very different models, running at different time steps: what has be done to check this " compatibility"?**
Answer:

As we already discussed (see answer to Page 7, line 26 ), the DDD model is able to provide realistic values for soil moisture deficit. As we are interested in the antecedent soil moisture conditions and not the variation of the soil moisture deficit during rainfall event, the timestep is not of such importance.  Other options would be to use soil moisture data from remote sensing or based on antecedent precipitation but these values are much less accurate. In addition, the soil moisture deficit values are not as important (as suggested by the sensitivity analysis) for high return periods.

**Page 8, line 23:  Cs is a coefficient accounting for the relation between temperature and snowmelt Properties It is usually called a " degree-day" coefficient (although used at a hourly time step here).**
Answer:

Both terms are used in literature but, as this is used in hourly time step, we prefer to use temperature index method

**Page 8, line 30:  The term under the bar should be " power to k" not be multiplied by k.**
Answer:

This seems to be correct- multiplied by k.  The return period for the POT events is: $T = \frac{1}{k(1-P)}$ where k- is the number of events and P is the non-exceedance probability

**Page 9, line3:  These simulated events were compared with the POT flood events extracted from the observations At this point, I don't clearly understand**

the simulation process. **Some lines detailing the simulation process (sequence of random drawings, number of simulation, processing of events, etc.), as well as a diagram, are really necessary to the reader before entering into the analysis of the simulations.**
Answer:

The description will be revised (see previous answers).

**The simulated CDFs look affected by under-sampling above the 500 yr. return period (i.e. not enough simulations of this range), which interrogates the robustness of the 1000 yr. estimations which are assessed in the paper.**
Answer:

More simulations (100 000 instead of 40 000) will be used to address this issue.

**Page 9, line 7: large variation in precipitation values Which duration is considered here? Daily?**
Answer:

It is the total depth – in this case 24 or 48 hours, it will be revised as: a large variation in total precipitation depths

**The comparison to the 100 yr. precipitation depths estimated thanks to the GP fit evoked in §2.3 would be useful.**

The suggested figure will be added to the revised version of the paper.

**Page 9, line 14: even though fully saturated conditions are used in the event-based PQRUT model I don't understand this: the lp variable (variable initial loss) has been introduced in §2.5 to depart from this fully saturated**

**hypothesis.**

Most commonly, fully saturated conditions are assumed for the standard PQRUT model. The reason for using the variable lp is to allow us to simulate flood events for which the initial conditions are not fully saturated.

**Page 9, line 16: A sensitivity analysis was performed for the three test catchments Once again, the detailed protocol of this analysis deserves to be presented for a better understanding of the results. Some information is given in Table 3 but would deserve to be detailed in the text. A more logical " progression" of the different setups could be: 2, 3 (statistical hypothesis on precipitation), then 4 (temporal disaggregation), then 5,6,7 (simple hydrological assumptions) and finally 1 (PQRUT parameters). This would apply for the Table 3, as well as for the writing of §2.7**

A similar comment was also raised by reviewer 2. In response to these comments we will revise this section and include a table that illustrates the set up in the revised version of the manuscript.

**Page 9, lines 24-28: I am not fully convinced by this explanation based on BFI. The sensitivity to initial loss should be linked to the possible values of initial loss in relation with the high quantiles of precipitation. I would be interested by looking at those values (maximum initial loss and 10, 100 and 1000 yr. precipitation) for the three catchments.**

This is a good suggestion. This analysis will be included in the revised version.

**Page 9, line 31: In addition, Krinsvatn shows high sensitivity to snowmelt**

**This is in contradiction with Page 9, line 10 (for Krinsvatn [. . . ] in most cases snowmelt does not contribute to the extreme floods). Any comment?**

Krinsvatn shows a high sensitivity to excluding the snow component in the simulation. The reason is that the snowmelt is negative (there is snow accumulation).

**Page 10, line 7: Ovrevatn and Horte showed sensitivity (28.9%) to the choice of the statistical distribution for modelling precipitation A figure showing the precipitation distribution for each catchment (both observed and modelled by GP, and EXP) would be welcome to illustrate lines 7 to 10.**

We will a add return level plot that shows the fit to the observations for both GP and EXP (see answer above ).

**§3- Comparison with standard methods Page 10, line 27: the standard implementation of the event-based PQRUT method This is the first mention of such a " standard" implementation. I think this would deserve to be presented at the very beginning of the paper, which proposes a " stochastic PQRUT" being a significant enhancement from the " standard PQRUT". The context of this study would thus be better understood.**

As of now, the Introduction provides a detailed overview of the methods for estimating extreme floods. Presenting the standard methods used in Norway in the introduction will narrow the scope, as potentially the international interest of this manuscript.

**Page 10, line 29: the annual maximum series were extracted from the observed daily mean streamflow series Why not using a GPD with the POT sample of floods extracted for the study of the critical duration?**

The fitting of a GEV distribution to the AMAX series represents a standard implementation of the flood estimation guidelines in Norway (Midttømme et al. 2011). This is the reason why we used the AMAX series instead of the POT events.

**Page 10, line 31: to obtain instantaneous peak values, the return values were multiplied by empirical ratios, obtained from regression equations Here I don't understand why the POT flood events extracted from observations (shown in the plots of the Figure 6) has not been used to fit either a GPD, or a GEV after extraction of annual maxima. More comments about this would be welcome.**

Much longer series of data are available at daily timestep than at sub-daily timesteps, as technology making sub-daily series widely available was only introduced during the 1980s, whereas many daily records are over 100 years in length. Fitting a GPD distribution to the instantaneous peak flows and using this model to predict the 100 -year return period will involve much higher uncertainty.

**Page 11, line 11: obtained from growth curves based on the 5-year return period value If I understand properly, the shape of the design hyetograph is based on the growth curves considered at the 5 yr. return period. Are the ratios between the different duration values at this return period deduced from empirical distribution, or inferred from a fitted distribution? Later on, this must be scaled to define a 100 or 1000 yr. hyetograph. What precipitation distribution (duration and model) are these extreme values deduced from?**

The Gumbel distribution is used to derive the growth curves, while the ratios between the different durations are derived from an empirical distribution following a procedure developed by NERC in the UK in the 1970s and later applied in Norway, based on

Norwegian data. This section will be revised in the text with references to these procedures given.

**Page 11, line 15: The performance of the three models was validated by using two different tests In that case, dealing with 100 or 1000 yr. flood estimations, it's more about " comparing different approaches".**

Even though the uncertainty is high for these return periods, a check that the data is within the confidence interval can be used as a validation (e.g Lamb et al 2016).

**Page 11, line 20: As discussed, due to the difficulty in assigning initial conditions for the event-based PQRUT model I don't understand this sentence, and to which discussion it refers.**

This refers to the fact that fully saturated conditions are used in the standard implementation of the PQRUT model and will be clarified in the revised text.

**Page 11, line 22: the regional equations were used Which regional equations? For PQRUT parameters?**

Yes, we used the regional equations for the PQRUT parameters. We will make this revision.

**Page 11, line 25: equation of QS + observed probabilities (Qobsi) are calculated using Gringorten positions for the POT series The POT series are used here, contrary to the daily (transposed to peak) annual maximum values that have been fitted in the statistical approach. Another option (already mentioned in my remark for page 10, line 29) could be to fit the statistical method on the**

**POT sample, which would have allowed to keep it as a " benchmark" method, given more sense to the comparison presented (or conversely using the " peak-from-daily" observations for the QS calculation).**

This is a good point; the daily flows will be used for the QS calculation.

**Page 11, line 30: the results vary between catchments as shown in fig 8 I don't find this figure very useful, the reader is unable to interpret the coloured dots. An alternative, aside the boxplots, could be some scatter plots (statistical Q100 and Q1000 v/s standard and stochastic PQRUT, statistical QS v/s standard and stochastic PQRUT, etc.).**

We will add scatterplots in the revised version of the manuscript.

**Page 11, line 32: we can conclude that the performance of the standard PQRUT model is poorer than the performance of the statistical flood frequency analysis and the stochastic PQRUT model The results which ground this conclusion are not explicitly presented. The only clue given to the reader is the Figure 8 which only presents the distribution of QS scores for FFA and stochastic PQRUT. The results, in terms of QS score as well as confidence interval, should be presented in a table and in an adequate figure.**

Thanks for pointing this out, a table will be added.

**Page 12, line 3: The violin plots (fig. 9) See remarks on Figure 9 below.**

Figure 9 shows both violin plots and boxplots (overlayed in gray). In order to increase the readability of the figure, only the boxplots will be plotted.

**Page 12, line 7: Reasons for this may be that higher precipitation intensity or snowmelt is used To assess this, the values of the reference hyetographs used in standard PQRUT deserve to be presented and compared to the simulated precipitation values of stochastic PQRUT (like the values of the Table 2 for Q100).**

A figure will be added to illustrate this, taking into account the reviewer's comments on figure 3.

**Page 12, line 8: the absolute differences between the two methods are larger in catchments with lower temperature (fig. 9) I wonder how this can be deduced of illustrated by Figure 9, it is more likely somehow in Figure 10.**

Apologies for this error and thank you for pointing this out. This should be Fig 10.

**Page 12, line 17: which might be due to the uncertainty in estimating the parameters for the GEV distribution I don't understand this interpretation which appears rather quick and subjective to me.**

We agree, this section will be deleted in the revised manuscript.

**Page 12, lines 18-21: This using of the study of Rogger et al. (2012) is off**

**topic for me here, as it is based on Gumbel, whereas the FFA is done here with GEV, which is more flexible.**

This is true, this is the reason that in the paper we specifically discuss the fact that the Gumbel distribution was used in the study of Rogger et al. (2012). This section will be deleted.

**§4- Conclusions Page 13, line 10-15:**

**Another modelling option could be to run the event-based simulation with the DDD model, already used for the initial condition. In that case, an hourly version of DDD should also be calibrated (with local observations or regionally), in compatibility with the daily version used for initial conditions. I am not fully aware of the potential difficulties of this, but it would be a more homogeneous approach in terms of hydrological modelling. Any comment about this?**

Yes, this is a possibility. However, in large catchments it is not as important to use a subdaily timestep, as the peak and daily flows are similar.

**Page 13, line 28: easily incorporate the uncertainty associated with this choice This is a very good remark: the stochastic process here adequately models a variable which, when represented in a deterministic way (i.e. fixed initial conditions), appears as highly uncertain.**

Yes, this has also been discussed in several other studies.

**Page 13, line 31: based on an assessment of the uncertainty characterizing the individual methods This is an interesting suggestion, but it has to be added that a proper expression of uncertainty for a rather sophisticated**

**method like stochastic PQRUT is far from trivial, and is still to be investigated. . .**

This will be revised as:

*Although it might be difficult to quantify the uncertainty*

**Tables**
**Table 1**
**Units are missing, as well as legend of the columns in the caption. Table 2: Units are missing, precipitations could be rounded to the next mm. For Krinsvatn, the probability to find the Q100 events in one season or the other could be provided.**

**Figures**
**Figure 3:**
**Not very informative, re-scaling storm hyetograph is not something difficult to understand. A set of different " typical" hyetographs could instead be presented for the three catchments, ideally illustrating the potential diversity of storm dynamic.**
**Figure 4:**
**" for Krinsvatn catchment" could be added in the caption, as well as the number of observed and simulated events.**
**Figure 6:**
**The remarkable return periods (10, 100 and 1000 yr.) should be distinguished in the plots (by a bolder vertical line for example).**
**Figure 7:**
**There are too many distributions in the plots, their interpretation is not easy. Two plots could be edited for each catchment, having for example only the " calibrated" simulation in common.**
**An uncertainty band around the " calibrated" simulation would be useful to**

**assess the intrinsic uncertainty of the simulation process.**
**Figure 8:**
**See comment of page 11, line 32.**
**Figure 9: I am not convinced by the usefulness of the violin plots here considering the limited number of values per scores (20 catchments). Box plots with outliers would have been sufficient and more readable.**
**Captions of the methods sometimes overlap.**

Thanks, these revisions and suggestions will be taken into use in the revised manuscript

---

## Author Comment (AC2) · 16 Aug 2018

We would like to thank reviewer 2 for the suggestions. We have provided detailed answers to each of the comments below.

**General Comments: As I understood, the main purpose of the work is to propose a methodology to overcome the limitations of more commonly applied event based modelling for flood frequency estimations by a stochastic modelling of preconditions, including SWE, and meteorological input. The individual modelling of the different aspects are described in the manuscript, however, it**

[Figure]

**is hard to follow how the different parts are connected. A preceding sub-section with a less detailed step by step explanation of methodology, maybe including a schematic illustration (inputs/ models/ methods / output), could help to better explain the methodology.**

A similar comment has been made by reviewer 1. We will include a subsection which precedes the full explanation of the method and describes the general approach, including a diagram which illustrates how the various components are connected.

**For the validation of the disaggregation procedure the disaggregated data were compared against hourly station data. Is this correct? I would be interesting to see how well the disaggregation procedure was performing (For example showing a obs-sim, QQ-plot). It is stated that it works better than equal divisions which is not surprising. What is the advantage of the further equal division to 1h if it is stated that 3-houers are already enough? Further, it is not obvious why the gridded seNorge.no Data are matched to the HIRLAM data if they are in the needed temporal resolution already?**

The HIRLAM data is a hindcast dataset with a spatial resolution of around 10 km2 and a temporal resolution of 3 hours. The gridded seNorge data is obtained by triangulation of the observed rainfall dataseries; it has a spatial resolution of 1km2, and a temporal resolution of 24-hours. As the HIRLAM data has a higher temporal resolution than the seNorge data, the HIRLAM data was used to disaggregate the seNorge data to a 3-hour timestep. The performance, including the validation of the disaggregation procedure, is described in Vormoor and Skaugen (2013).

For the work presented here, the precipitation data were further disaggregated to a 1-hour time step by dividing into three equal parts. This was simply done for convenience, as the PQRUT model has previously been calibrated relative to 1-hour

streamflow data and similar climate input data (i.e. 1-hour data derived from 3-hour data by dividing into three equal parts). A similar comment was raised by reviewer 1, and this section will be revised to include the above clarifications.

**A 1000 years event is extrapolated from daily observation series (length not further specified). Furthermore the results are then multiplied by empirical factors, to match sub-daily peak flows. I am not aware of the engineering practice in Norway, however, I am not sure about the meaning of the results by this extreme extrapolation and at least this should be critically discussed.**

The fitting of an extreme value distribution to estimate the return level for periods longer than the length of a time series is a standard procedure, both in hydrological investigations and in engineering practise. As suggested by the reviewer, the uncertainty of the estimates does increase significantly for longer return periods, relative to the length of record. The length of the daily streamflow series considered here, however, justifies the use of an 'at-site' (cf. a regional) flood frequency analysis as the minimum length is 31 years, while the median is 65 years of data. The following sentences will be added to the text:

*The length of the daily streamflow series justifies the use of at-site flood frequency analysis (Kobierska, et al., 2018); the minimum length is 31 years, while the median is 65 years of data. However, it is expected that the uncertainty will be high when the fitted GEV distribution is extrapolated to 1000-year return period. The 1000-year return period is used here, however, as it is required for dam safety analyses in Norway (e.g. Midttømme, et al., 2011; Table 1). More robust, but potentially less reliable, estimates could be obtained using a 2-parameter Gumbel, rather than a 3-parameter GEV distribution (Kobierska, et al., 2018).*

**The sensitivity analysis is interesting, however, also confusing including Figure 7 and Table 3. It is not obvious on what basis the percentage difference**

[Figure]

**is calculated. This is also not clear in the follow up comparison of the methods. What exactly is the calibrated model? Also the section misses an explanation of the shaded area which is prominently displayed in Figure 7. Furthermore, the different precipitations settings tested are not well explained. A table, summarizing the different tested aspects, would help to guide the reader.**

Thank you for these comments and suggestions. The shaded area represents the simulations based on the 5% and 95% confidence intervals for the regression equations for PQRUT. We will include a table in the revised version of the manuscript.The paragraph that describes the sensitivity analysis will also be revised as follows:

*A sensitivity analysis was performed for the three test catchments, Hørte, Øvrevatn and Krinsvatn, in order to determine the relative importance of the initial conditions, precipitation, and the parameters of PQRUT on the flood frequency curve. To test the sensitivity of the model, we have used several different model runs and calculated the percentage difference of each of these relative to the model simulation, as shown in fig 7. More detailed information on the set up is given in table 3. As these catchments are located in different regions and exhibit different climatic and geomorphic characteristics, we hypothesize that the flood frequency curve will be sensitive to different parameters and hydrological states, as well as local climate and catchment characteristics.*

**The Figures and especially the captions should be improved, as they are often not self-explanatory. This includes also missing units, labels and abbreviations. Maybe consider a professional language proof reading.**

In the revised version of the article, we will provide fuller explanations in the catchment, revise the units, labels and improve the language usage.

**Specific Comments: The abbreviation PQRUT, used from beginning (abstract), is not introduced on page 4 or rather page 8. Please declare the meaning of PQRUT first time mentioned.**

The abbreviation PQRUT comes from P-precipitation, Q- discharge and RUT - routing, this will be explained in the revised manuscript.

**The characterizations of catchments and chosen abbreviations are introduced on P4 and repeated later (P5, I5) without brackets (e.g. "sparse vegetation over tree line (B)" and "sparse vegetation over tree line B"). Either use brackets throughout the manuscript or only use the abbreviation. Additionally by choosing more selfexplanatory abbreviations or using full words (eg. forest; marsh), would be easier to understand, especially in Table 1.**

All of these suggestions will be implemented in the revised manuscript.

**P.5 I.15: The last sentence does not contain important informations and could be omit**

We prefer to keep this sentence as it gives useful information on how the data is derived and increases the reproducibility of the study.

**P.6 I.1: The addition ",which can be used for modelling in ungagged basins." could be omitted, as it seems not connected to the procedure.**

This sentence will be deleted.

**P.6 I.2: A citation should be added to the DDD model or the corresponding R-package.**

The reference is provided earlier in the manuscript, p4.

**P.6 l.11: In my opinion, the meaning of the "critical duration" rather reflects the link between the duration and intensity of precipitation "events" of a certain probability, than to ensure the modelling of the complete flood hydrograph.**

This is a good point, the sentence will be revised to: *When simulating flood response with an event-based model, it is important to specify the so-called critical duration (Meynink and Cordery, 1976) to ensure that the flood peak is correctly modelled. The critical duration is an important factor which effectively links the duration and the intensity of precipitation events of a given probability.*

**P.6 l.32: "individual risk seasons could have been defined". One wonders why it was not done? If not so important for the result, please consider to omit this half sentence.**

This sentence will be deleted. We used this season definition to match the seasonal definition used by the Norwegian Meteorological Institute.

**P.8 l.12-17: Please check grammar and style of the section.**

This section will be revised to also address the issues raised by reviewer 1 as follows:
*The PQRUT model was calibrated for the 45 highest flood events by using the DDS (Dynamically Dimensioned Search) optimization routine (Tolson and Shoemaker, 2007) and the Kling Gupta efficiency (KGE) criterion (Gupta et al., 2009) as the objective function. An additional parameter, lp, was introduced to account for initial*

*losses to the soil zone. The reason for this is that, even though fully saturated conditions are assumed when the model is used to estimate PMF or other extreme floods with low probabilities, the model needs to account for initial losses when actual (more frequent) events are simulated. This procedure is described in more detailed in Filipova et al. (2016 ).*

**P.8 l.28: Was there a specific reason for using the "Gringorten plotting" position?**

The Gringorten plotting positions provide unbiased quantile estimates for the Gumbel distribution. In this case, we don't know the distribution. However, the difference between the plotting positions is usually higher for the low and high quantiles. As reviewer 1 suggests we have increased the number of simulations. This means that differences derived from plotting position formulas will be relatively small when estimating the 1000 -year return period.

**P.12 l.3: A more detailed explanation what exactly is analyzed here is missing.**

The sentence will be revised to:
*A comparison of the stochastic PQRUT with the standard methods for flood estimation shows that there is a large difference between the results of the three methods for both Q100 and Q1000 (fig 9 and 10).*

**P.12 l.25: Maybe the catchment steepness should be introduced in the section "study area".**

Thanks for the suggestion. This will be added to the the section "study area".

**P.13 l.25: Why is it peak to volume? I thought it is daily mean to daily**

**max discharge?**

This refers to converting the daily volume (obtained from the daily mean) to the peak value.

**Table 1: Missing units. Furthermore, the variables could be sorted and clustered more logically (e.g. temperature and precipitation; Q and AMAX).**

Thanks, this will be revised in the revised version of the manuscript.

**Table 3: Why are 100 values sampled? Does T mean the threshold Parameter Trt ?**

For the sensitivity analysis we used 100 samples, as larger number will increase the computational time. We assume that this number is sufficient to calculate the intervals. Trt refers to the parameter of the PQRUT model, thanks for spotting this error.

**Figure 2: Labels and units are missing**

This will be corrected for the revised version of the manuscript.

**Figure 3: Labels and units are missing**

This will be corrected for the revised version of the manuscript.

**Figure 7: is confusing because of the large number of different colored lines. Maybe two plots can help to distinguish between the different aspects as for example the precipitation input and other aspects. The legend is confusing as well. GDP was fitted to what? Y-Axes should start at 0, x-axes missing a label**

**and to be consistent with the rest of the work it should not exceed 1000.**

This issue was also raised by reviewer 1 and the figure will be improved and revised based on the newer, longer simulations.

**Figure 8: It is impossible to distinguish between 20 colors. Do the colors have any meaning? If they should be recognizable, numbering would be a better option. The numbers could then also be used in Figure 10, so the link between the performance of the model and the results are given.**

The colors just represent different catchments but also as reviewer 1 suggests, scatterplots will be used instead in the revised version.

**Figure 9: What exactly is shown in the plots. Please add a more detailed explanation.**

A more detailed explanation of Figure 9 will be included in the revised manuscript.

**Figure 10: The scale "percentage difference" should be unambiguous. The base of the "difference" should be clarified.**

This will be clarified in the figure description in the revised version of the manuscript.

**Technical Notes:**
**"Figure" and "Table" should start with capitals**
**Please use the degree symbol e.g. 4âŮę C (P.5 l.10, P8 l23 +26,. . .)**
**Please use [mm year-1] instead of [mm / year]**
**P.3 l.22: grammar ", as is often used"**
**P.5 l.3: Subscript i in ai and Ai (ai and Ai)**

**P.7 l.6: Missing link to "section"**
**P.7 l.20: Typo: "multivariat" instead of "mul- tivariat"**
**P.7 l.23: Whitespace, "values,p,"**
**P.9 l.1: Whitespace, "where,P"**
**P.9 l.34: Whitespace, "(29**
**P.11 l.3: Whitespace, "GL(Generalised. . ." P.11 l.10: Typo, "mm//ŮȩC"**
**Check citation "Beven, Keith, . . . 2014"**
**Check citation "Chow, . . . 1988"**
**Check citation "Fleig, . . . 2013"**

Thanks, these revisions and suggestions will be taken into use in the revised manuscript.

---

## Author Response (AR1)

**Reply to reviewer 1**

This paper presents a significant enhancement of a Norwegian method for the estimation of extreme floods, based on an event-based rainfall-runoff simulation. It introduces a stochastic process for the assignment of the initial hydrological conditions before the simulated events, as well as for the intensity and the temporal dynamic of the simulated precipitation events. This method is compared to the initial method (which considers only a reference precipitation on given condition), and to a classical FFA. The presented method is interesting, both in terms of methodology and statistical results. It is well explored, with a detailed sensitivity analysis.

We thank the reviewer for the positive comment on the method and the very detailed review of the manuscript.

However, the paper could be greatly improved by a better writing and more illustrations, particularly about the stochastic PQRUT which deserves a detailed step by step explanation of the simulation procedure (text and diagram), and also the probabilistic models for precipitation.

We agree that the explanation of the procedure can be improved. In the revised version, we have added step by step procedure:

- 1. Extract flood events for a given catchment and identify the critical storm duration For each season:
- 2. Aggregate the precipitation data to match the critical duration for the catchment
- 3. Extract POT precipitation events and fit a GP distribution
- 4. Fit probability distributions for the initial discharge, soil moisture deficit and SWE values for the season
- 5. Generate precipitation depth from the fitted GP distribution
- 6. Disaggregate the precipitation depth to a 1 hour time step by matching the dates of the identified POT flood events (from step 3) to dataseries of precipitation with hourly timestep
- 7. Sample a temperature sequence by matching the dates of the identified POT flood events (from step 1) to the dataseries of temperature with hourly timestep
- 8. Sample initial conditions for snow water equivalent (*SWE*), soil moisture deficit and initial discharge from their distributions (step 5), accounting for co-variation using a multivariate normal distribution
- 9. Simulate streamflow values using the calibrated PQRUT model for the sample event
- 10. Repeat steps 6.-9. 100 000 times
- 11. Estimate the annual exceedance probability from the total of 400 000 (100 000 for each season) samples using plotting positions

We have also included a diagram to illustrate the link between the different steps.

Regarding the sensitivity analysis, which is very important to understand the key factors and options of this new method, its writing also should be better organized and illustrated. It lacks a basic but important study of the impact of the random drawing (e.g. by performing 100 different simulations) and of the number of the simulated events on the extreme quantiles estimation. The later seems to be an issue here for high return periods.

These two issues are somewhat related. The effect of the random seed will be minimised if larger number of simulations are used. In addition, increasing the number of simulations will increase the robustness of the extreme quantiles. However, the number of simulations that are needed will depend on the return period of interest. Here we have considered a long return period, 1000-years, so this issue is indeed relevant. We have included two additional variables in the sensitivity analysis:

50 simulations using different random seeds

Length of simulation, of up to 400,000, 100,000 for each season (in increments of 40,000). It is not feasible to consider longer simulations due to computational times.

I would recommend a significant revision of this paper, mostly to improve its structure, its writing and the illustrations provided. Detailed comments/suggestions are provided to the authors in that follows.

We hope that our revisions will significantly improve the writing and the structure of the paper. Abstract

For those not familiar with PQRUT, it could be added in the abstract that the stochastic PQRUT is an extension/evolution of the "standard" PQRUT routine, applied since many years in Norway (dates and references to be provided). The differences between the estimates can be up to 200% for some catchments, which highlights the uncertainty in these methods This is not a good message for hydrological engineering, a less pessimistic phrasing could be "[...] 200% for some catchments where the uncertainties of the compared methods are high and combine unfavourably".

The reason, we did not include this information is that we wanted to emphasise that the method can be applied to any catchment with snowmelt/mixed flood regime. But as our study area is in Norway, we have revised the abstract as:

The estimation of extreme floods is associated with high uncertainty, in part due to the limited length of streamflow records. Traditionally, statistical flood frequency analysis and an event-based model (PQRUT) using a single design storm have been applied in Norway. We here propose a stochastic PQRUT model, as an extension of the standard application of the event-based PQRUT model, by considering different combinations of initial conditions, rainfall and snowmelt, from which a distribution of flood peaks can be constructed. .... The differences between the stochastic PQRUT and the statistical flood frequency analysis are within 50% in most places. However, the differences between the stochastic PQRUT and the standard implementation of the PQRUT model are much higher, especially in catchments with a snowmelt flood regime.

§1 - Introduction Page 1, line 14: For example, floods with a 500-year return period are sometimes used to [...] As most of the estimates evoked in the paper are 100 or 1000-yr. floods, and example of the use of such quantiles in Norway could useful.

We will revise the paragraph as follows:

The estimation of low-probability floods is required for the design of high-risk structures such as dams, bridges, levees, etc. For example, floods with a 100-year return period are sometimes required for the design of levees and the design and safety evaluation of high-risk dams requires the estimation of flood hydrographs for the 1000-year return period and, in some cases, floods with magnitudes of up to the Probable Maximum Flood (PMF). Page 1, line 17: Flood mapping also usually requires input hydrographs This is also the case for dam safety assessment.

This sentence has been revised -See answer above

Page 1, line 21: When longer return periods are needed I guess the author means "longer than the record length", i.e. return period of 100 yr. or above.

Yes, this is correct. For clarity, this will be changed to:

When return periods that are longer than the observed record length are needed, the process requires extrapolation of the fitted statistical distribution. Page 2, line 6: have been shown to produce average errors between 27 and 70% Please mention on what estimation this error is computed (observed quantiles or estimated ones, of which return period).

These errors are based on table 1 in the paper by Salinas et al 2013, which provides a comparative assessment of different studies in ungauged basins. The values are calculated using the RMSNE (root mean square normalised error) for Q100 (q100) and we have expressed them as percentage. We have revised as: As the physical processes in the catchments are usually not directly considered in the analysis, estimating the flood quantiles in ungauged basins using regression or geostatistical methods can produce average RMSNE (root mean square normalised error) values of between 27 and 70% (Salinas et al., 2013), or even higher for the 100-year return period.

Page 2, line 32: they are computationally inefficient. . . Another writing could be "[...] they are computationally demanding, as long continuous periods have to be simulated to estimate extreme quantiles".

Yes, this is in fact a better way to write this.

Page 3, line 6: millions of rainfall events can be sampled from the MEWP model... More exactly, millions of synthetic rainfall events can be generated, assigned to a probability estimated from the MEWP model, and inserted...

Yes, we agree with the revision.

Page 3, line 21: it requires the generation of a temperature sequence for the event I would add " a temperature sequence for the event, coherent with the simulated rainfall, and a snow water. . . "

We have revised this sentence as:

as it requires the generation of a temperature sequence for the event that is consistent with the rainfall sequence used and a snow water equivalent as an initial condition.

**Page 3, line 22: The assumption of a fixed rate of snowmelt [...] and a joint probability model needs to be considered Does it mean that a fixed snowmelt is usually added without consideration to the rest of the variables which will characterize the simulated event? What kind of joint probability model should be added?**

To increase the clarity, these sentences has been rewritten as: The assumption of a fixed rate of snowmelt which is based on typical temperatures, as is often used in Norway for the single event-based design method, can introduce a bias in the estimates. The joint probability of both rainfall and snowmelt needs to be considered to obtain a probability neutral value (Nathan and Bowles, 1997).

Page 24, line 24: SEFM which has been applied in several USGS studies To my knowledge, I am not sure it is USGS (although SEFM is evoked in the USGS Bulletin 17C " Guidelines for Determining Flood Flow Frequency" of 2018), but several application of SEFM for dam safety studies have been delivered to USBR (US Bureau of Reclamation).

Yes, it should be USBR.

Page 3, line 34: due to the large uncertainty in both the event-based model and the statistical flood frequency analysis I am not comfortable with this writing. With two identical methods (say a classical FFA), but with two distributions fitted on two different samples, estimations would be different, and in that case this difference is completely linked to the uncertainties of the FFA (and mainly the sample uncertainty). But with different methods, these differences can also be produced by discrepancies between methods which should be treated per se, in order to assess the method themselves. So the interpretation of the difference should, in my opinion, not only rely on uncertainties.

This sentence has been deleted.

Page 3, line 35: To better understand the differences between these methods, a sensitivity analysis of the stochastic PQRUT is performed Here I have somehow the opposite comment from above: this sensitivity analysis of stochastic PQRUT is more about dealing with the uncertainties of stochastic PQRUT, not the differences between methods.

This is true, the sensitivity analysis will provide a better understanding of the uncertainty. However, the sensitivity analysis can also help us determine the reasons for the differences between the models. For example, the standard PQRUT assumes that fully saturated conditions are used. By testing the sensitivity of the model to the initial soil moisture deficit, we can check whether assuming fully saturated conditions contributes to the difference between these two methods.

We have revised this section as:

In order to understand the uncertainty and the differences with the standard PQRUT model, a sensitivity analysis of the stochastic PQRUT is performed by considering the effect of the initial conditions, model parameters and rainfall intensity on the flood frequency curve.

§2 - Stochastic event-based model Page 4, line 18: The study area consists of a set of 20 catchments A more logical phrasing could be "The study area in Norway, with a dataset of 20 catchments located throughout the whole country"

We have revised this sentence.

Page 5, line 6: for Krinsvatn, Lk and the area covered by marsh, M, is more than 10In the Table 1, Lk and M values are 9 and 1.1 %, respectively.

This is correct, we have revised as: but for Krinsvatn, the Lk is higher and the area covered by marsh, M, is 9Page 5, line 28: the correlation of the method was found to be higher To which values the results of this disaggregation method have been correlated? Hourly or 3-hourly rainfall observations?

The disaggregated data were compared to the 3-hour observations in the study undertaken by Vormoor and Skaugen to produce the gridded disaggregated data. Page 5, line 29: simply dividing the seNorge data into eight equal parts To be clearer, I suggest '' simply dividing the seNorge daily data into eight equal 3- hourly values''.

The sentence has been revised as suggested. Page 5, line 29: disaggregated to a 1-hour time step using a uniform distribution to match the time resolution of the discharge data, although a 3-hour time step could also be used I don't fully understand this. Was the 3-hour value affected randomly to one hour or divided into three? Why is it possible to use the 3-hour value with an hourly model?

The rainfall data was then divided into three equal parts, i.e. using a uniform distribution. The PQRUT model can be used with any timestep. Considering that the median catchment size is around 140 km2, a timestep of 3 hours should be sufficient for modelling the peak flows. As the model has previously been calibrated and regionalised using a 1 -hour timestep, we decided to use this timestep (instead of 3-hours). A similar comment was raised by reviewer 2 and the section was revised to include these suggestions as well:

The HIRLAM atmospheric model for northern Europe has a 0.1 degree resolution (around 10 km2) and we used a temporal distribution of three hours. The HIRLAM data set was first downscaled to match the spatial resolution of the seNorge data and the precipitation of the HIRLAM data was rescaled to match the 24-hour seNorge data (Vormoor and Skaugen, 2013). Then, these rescaled values were used to disaggregate the seNorge data to a 3-hour time resolution. The method was validated against 3-hour observations, and the correlation of the method was found to be higher than that obtained by simply dividing the seNorge data into eight equal 3 -hourly values (Vormoor and Skaugen, 2013). These datasets were further disaggregated to a 1-hour time step by dividing into three equal parts to match the time resolution of the streamflow data.

Page 6, line 1: remotely sensed data Assimilating remotely sensed data in a hydrological model is not an easy task, especially soil moisture. Any reference to provide that would apply to the context of this paper?

Using the remotely-sensed data currently available is probably not ideal as these data have a coarse spatial resolution (around 20km2). A review of the use of this data is given in Brocca et al (2017). This reference has been added to the revised version: Sources of these data can be e.g. remotely sensed data (see, for example, the review provided in Brocca et al. (2017)) or gridded hydrological models.

Page 6, line 2: the DDD hydrological model was used Please provide a reference which introduces this model.

The reference to the DDD model comes earlier in the manuscript, i.e. on p 4. **Page 6, line 7: exceeds 0.3 of its (dynamic) capacity Please define what is a dynamic capacity here.**

In this case, dynamic refers to the concept that the volume of the saturated zone and unsaturated varies in time as explained in the previous sentence. This has been deleted.

Page 6, line 11: the so-called critical duration A reference can be provided here to define this concept: Meynink, W. J. Cordery, I. (1976). Critical duration of rainfall for flood estimation. Water Resources Research, 12(6), 1209-1214. Thanks for this, we have included this citation.

Page 6, lines 12-14: In order to determine [...] had to be determined for each catchment I don't find this sentence useful, considering what is written before and after it.

The sentence has been deleted.

Page 6, lines 14: flood events over a certain quantile threshold (0.9) were extracted. On what data this POT sample was extracted: daily or hourly discharges?

The sample is based on daily discharge, we have revised as: To determine the critical duration, flood events from the daily time series over a quantile threshold (in this case, 0.9) were extracted.

Page 6, lines 18-25: An alternative to this could be to study the correlation between the peak daily value and the precipitation of that day (which we could call P0), the sum P0 to P-1, P0 to P-2 and so on. . . When the correlation coefficient stops to increase significantly, it means that the correct length of the " precipitation window" is reached, thus the critical duration is estimated. This is likely to be more robust than studying the correlation between the peak daily discharge and the individual precipitations the days before.

This is an interesting suggestion. In this study, it is important to specify a critical duration in order to capture the "full" rainfall event producing the peak flow (otherwise the peak flow is likely to be underestimated). We implemented the procedure proposed by reviewer but we did not find much difference between the two methods. In fact, this alternative gives shorter critical durations in some cases (i.e. 1-day for Krinsvatn instead of 48 hours). As our study area consists of small catchments and the shortest window we are using is 1-day, the critical duration is then also 1 day (for 17 out of 20 catchments), rather than being longer.

Page 6, lines 22-24: In some catchments (mostly those having a snowmelt flood regime), no significant correlation was found between discharge and precipitation In that case, some processing of the flood is needed, e.g. only considering the "snowfree" seasons, or adding a threshold on the precipitation over the preceding days in the POT selection of floods. This could prevent using an arbitrary duration.

We have incorporated this and now only the relatively snow-free season (SON) have been considered.

Page 6, line 28: the sequence of the input data must be prescribed for the stochastic simulation What means " prescribed"? Is it generated? Is it randomly drawn from the observed sequences?

Yes, we have revised to "generated".

Page 7, line 1: a Generalized Pareto distribution was fitted to the series of selected Events A figure with the corresponding fits and observations for the example catchments would be welcome.

We have added a return level plot that shows the fit to the observations for both GP and Exponential distributions. **Page 7, line 6: introduced in section ??**

Section 2.1.2, the reference has been corrected.

Page 7, line 7: Using the fitted Generalized Pareto (GP) distribution, precipitation depths were simulated Does it mean that probabilities where randomly drawn then the corresponding precipitation depths deduced from the fitted GPD? How many events are drawn?

This is correct, the sentence will be rewritten as:

The precipitation depths were generated (for 100,000 events) from the fitted GP distribution for each season. Originally, 160 000 (40 000 per season) simulations (around 10 000 to 20 000 years) were used, we have now increased this number to 400 000 events.

Page 7, line 8: a storm hyetograph was first sampled How is it sampled? I guess it comes from the hyetographs collection corresponding to the POT selection of precipitation events, but with what consideration to season, intensity, etc.?

Yes, it was sampled from the collection of hyetographs, the seasonality was considered but not the precipitation depth This has been revised in the manuscript.

**Page 7, line 10: Pi and P are not defined in the disaggregation formula.**

We have revised as:

where Phsim is the simulated 1-hour precipitation intensity, Pdsim is the simulated daily intensity and Pi is the 1 -hour disaggregated SeNorge intensity

Page 7, line 15: Output from DDD model runs Have the DDD models been calibrated on local data? If yes, some words about the calibration method are welcome. I guess the DDD models are used at daily time-step, is it true?

Yes, a daily time step was used. The following sentence has been added to the text The model was calibrated for the selected catchments at a daily timestep using a MCMC routine (Petzoldt, 2010).

Page 7, lines 21-25: The writing is not clear, and neither is the equation of the mixed distribution (what is x ?). As far as I understood, it is about randomly switching between a trivariate (discharge, moisture, snow) and a bivariate (discharge, moisture) distribution depending on the probability of having snow on a given season. Is p also drawn for the simulation?

We agree that this was not clearly described, we have revised this section as follows: The probability p for switching between the trivariate and bivariate distributions is based on the historical data for SWE higher than 0.

Page 7, line 26: The correlation between the observed and simulated variables is shown in Figure 4 Apparently, sl is the soil moisture deficit. Contrary to SWE and Qobs which are "observable" variables, sl is linked to a model (here DDD). So it should be introduced, in relation with the DDD model structure.

The soil moisture deficit is presented in Skaugen and Onof (2013). The soil moisture deficit is the difference between the volume of the unsaturated zone and the volume already present in the soil moisture zone.

Page 8, line 5: for estimating design floods and safety check floods for dams in Norway This type of application is perhaps documented in (Andersen, 1983), but this reference is not easily accessible on line, and is written in Norwegian, so a accessible reference documenting this type of application would be welcome.

A reference to the NVE report is now provided:

The PQRUT model is a simple, event-based, 3-parameter model (Fig. 6) which is used, amongst other things, for estimating design floods and safety check floods for dams in Norway (Wilson et al., 2011).

Page 8, line 14: The general procedures used for the PQRUT calibration are described in Filipova et al. Some details about this calibration would be welcome, e.g. which flood events sample is considered (is it the same as the one used in §2.2 for critical duration)?

In the calibration, the 45 highest flood events were considered. This sample most likely overlaps with the events selected for the critical duration. The sentence has been revised as:

The PQRUT model was calibrated for the 45 highest flood events by using the DDS (Dynamically Dimensioned Search) optimization routine (Tolson and Shoemaker, 2007) and the Kling Gupta efficiency (KGE) criterion (Gupta et al., 2009) as the objective function.

Page 8, lines 15-17: This additional parameter lp should be documented in the structure of the PQRUT model presented in the Figure 5. Furthermore, I am not sure that it can be considered as a parameter, more likely it is an internal state variable which vary from event to event.

We agree in this case, lp can be considered as a state variable. The figure has been updated to include lp.

Page 8, line 18: the value of this parameter was set to the initial soil moisture deficit, estimated using DDD This is an important assumption: it means that some internal variables of DDD (which ones, this is not documented) are used to estimate another one in PQRUT. This is far from obvious to accept for two very different models, running at different

**time steps: what has be done to check this " compatibility"?**

As we already discussed (see answer to Page 7, line 26), the DDD model is able to provide realistic values for soil moisture deficit. As we are interested in the antecedent soil moisture conditions and not the variation of the soil moisture deficit during rainfall event, the timestep is not of such importance. Other options would be to use soil moisture data from remote sensing or based on antecedent precipitation but these values are much less accurate. In addition, the soil moisture deficit values are not as important (as suggested by the sensitivity analysis) for high return periods.

Page 8, line 23: Cs is a coefficient accounting for the relation between temperature and snowmelt Properties It is usually called a "degree-day" coefficient (although used at a hourly time step here).

Both terms are used in literature but, as this is used in hourly time step, we prefer to use temperature index method **Page 8**, **line 30**: The term under the bar should be " power to k" not be multiplied by k.

This seems to be correct- multiplied by k. The return period for the POT events is: T = 1/(k (1-P)) where k- is the number of events and P is the non-exceedance probability

Page 9, line3: These simulated events were compared with the POT flood events extracted from the observations At this point, I don't clearly understand the simulation process. Some lines detailing the simulation process (sequence of random drawings, number of simulation, processing of events, etc.), as well as a diagram, are really necessary to the reader before entering into the analysis of the simulations.

The description has been revised (see previous answers). The simulated CDFs look affected by under-sampling above the 500 yr. return period (i.e. not enough simulations of this range), which interrogates the robustness of the 1000 yr. estimations which are assessed in the paper.

More simulations (400 000 instead of 160 000) have been used to address this issue (also see previous answers).

Page 9, line 7: large variation in precipitation values Which duration is considered here? Daily? It is the total depth – in this case 24 or 48 hours. This is now also explained in the text. The comparison to the 100 yr. precipitation depths estimated thanks to the GP fit evoked in §2.3 would be useful.

A figure (fig 4) that shows the return level plots (which shows 100 -year return period) has been added to the revised version of the paper.

**Page 9, line 14: even though fully saturated conditions are used in the event-based PQRUT model I don't understand this: the lp variable (variable initial loss) has been introduced in §2.5 to depart from this fully saturated hypothesis.**

Most commonly, fully saturated conditions are assumed for the standard PQRUT model. The reason for using the variable lp is to allow us to simulate flood events for which the initial conditions are not fully saturated.

Page 9, line 16: A sensitivity analysis was performed for the three test catchments Once again, the detailed protocol of this analysis deserves to be presented for a better understanding of the results. Some information is given in Table 3 but would deserve to be detailed in the text. A more logical " progression" of the different setups could be: 2, 3 (statistical hypothesis on precipitation), then 4 (temporal disaggregation), then 5,6,7 (simple hydrological assumptions) and finally 1 (PQRUT parameters). This would apply for the Table 3, as well as for the writing of §2.7

A similar comment was also raised by reviewer 2. In response to these comments we have included a table that illustrates the set up in the revised version of the manuscript.

Page 9, lines 24-28: I am not fully convinced by this explanation based on BFI. The sensitivity to initial loss should be linked to the possible values of initial loss in relation with the high quantiles of precipitation. I would be interested by looking at those values (maximum initial loss and 10, 100 and 1000 yr. precipitation) for the three catchments.

This is a good suggestion. We have included this analysis in the revised version: A reason for this is that for Øvrevatn, higher soil moisture conditions are associated with higher rainfall quantiles. For example, for Øvrevatn, precipitation depths with a

1000-year return period are associated with median soil moisture conditions of 37 mm, while for Krinsvatn, it is 30.8 mm and for Hørte, it is 16.7 mm.

**Page 9, line 31: In addition, Krinsvatn shows high sensitivity to snowmelt This is in contradiction with Page 9, line 10 (for Krinsvatn [...] in most cases snowmelt does not contribute to the extreme floods). Any comment?**

Krinsvatn shows a high sensitivity to excluding the snow component in the simulation. The reason is that the snowmelt is negative (there is snow accumulation). The sentence was revised as:

In addition, Krinsvatn shows a high sensitivity to the snowmelt component (21% higher) and also a step change in the frequency curve, even though the soil moisture deficit is higher. This can also be explained by the fact that the snowmelt contribution is negative (there is snow accumulation), as can also be seen in Table 2.

Page 10, line 7: Ovrevatn and Horte showed sensitivity (28.9%) to the choice of the statistical distribution for modelling precipitation A figure showing the precipitation distribution for each catchment (both observed and modelled by GP, and EXP) would be welcome to illustrate lines 7 to 10.

We have a add return level plot that shows the fit to the observations for both GP and EXP (see answer above)

§3- Comparison with standard methods Page 10, line 27: the standard implementation of the event-based PQRUT method This is the first mention of such a "standard" implementation. I think this would deserve to be presented at the very beginning of the paper, which proposes a "stochastic PQRUT" being a significant enhancement from the "standard PQRUT". The context of this study would thus be better understood.

As of now, the Introduction provides a detailed overview of the methods for estimating extreme floods. Presenting the standard methods used in Norway in the introduction will narrow the scope, as potentially the international interest of this manuscript.

Page 10, line 29: the annual maximum series were extracted from the observed daily mean streamflow series Why not using a GPD with the POT sample of floods extracted for the study of the critical duration?

The fitting of a GEV distribution to the AMAX series represents a standard implementation of the flood estimation guidelines in Norway (Midttømme et al. 2011). This is the reason why we used the AMAX series instead of the POT events.

Page 10, line 31: to obtain instantaneous peak values, the return values were multiplied by empirical ratios, obtained from regression equations Here I don't understand why the POT flood events extracted from observations (shown in the plots of the Figure 6) has not been used to fit either a GPD, or a GEV after extraction of annual maxima. More comments about this would be welcome.

Much longer series of data are available at daily timestep than at sub-daily timesteps, as technology making sub-daily series widely available was only introduced during the 1980s, whereas many daily records are over 100 years in length. Fitting a GPD distribution to the instantaneous peak flows and using this model to predict the 100 -year return period will involve much higher uncertainty.

Page 11, line 11: obtained from growth curves based on the 5-year return period value If I understand properly, the shape of the design hyetograph is based on the growth curves considered at the 5 yr. return period. Are the ratios between the different duration values at this return period deduced from empirical distribution, or inferred from a fitted distribution? Later on, this must be scaled to define a 100 or 1000 yr. hyetograph. What precipitation distribution (duration and model) are these extreme values deduced from?

The Gumbel distribution is used to derive the growth curves, while the ratios between the different durations are derived from an empirical distribution following a procedure developed by NERC in the UK in the 1970s and later applied in Norway, based on Norwegian data. This section has been revised: The standard implementation of PQRUT involves using a precipitation sequence that combines different intensities, obtained from growth curves based on the 5-year return period value fitted using a Gumbel distribution while the ratios between the different durations are derived from empirical distribution (Førland, 1992).

**Page 11, line 15: The performance of the three models was validated by using two different tests In that case, dealing with 100 or 1000 yr. flood estimations, it's more about " comparing different approaches".**

Even though the uncertainty is high for these return periods, a check that the data is within the confidence interval can be used as a validation (e.g Lamb et al 2016).

Lamb, R., Faulkner, D., Wass, P. and Cameron, D.: Have applications of continuous rainfall-runoff simulation realized the vision for process-based flood frequency analysis?, Hydrol. Process., 30(14), 2463–2481, doi:10.1002/hyp.10882, 2016.

**Page 11, line 20: As discussed, due to the difficulty in assigning initial conditions for the event-based PQRUT model I don't understand this sentence, and to which discussion it refers.**

This refers to the fact that fully saturated conditions are used in the standard implementation of the PQRUT model. The following sentence was added: The standard implementation of the event-based PQRUT model was not evaluated based on QS as initial conditions could not be assigned for low return periods. As this model is usually used to calculate high quantiles (Q100 or higher), fully saturated conditions are assumed for its implementation.

Page 11, line 22: the regional equations were used Which regional equations? For PQRUT parameters?

Yes, we used the regional equations for the PQRUT parameters. This is now correct in the revision.

Page 11, line 25: equation of QS + observed probabilities (Qobsi) are calculated using Gringorten positions for the POT series The POT series are used here, contrary to the daily (transposed to peak) annual maximum values that have been fitted in the statistical approach. Another option (already mentioned in my remark for page 10, line 29) could be to fit the statistical method on the POT sample, which would have allowed to keep it as a "benchmark" method, given more sense to the comparison presented (or conversely using the "peak-from-daily" observations for the QS calculation).

This is a good point; the daily flows were used for the QS calculation. We have revised the description of the method: In Eq. 5 the observed probabilities (Qobsi) are calculated using Gringorten positions for the peak AMAX series that were derived from the daily values. The modelled probabilities that correspond to the observed events are calculated by using the statistical flood frequency analysis and the Stochastic PQRUT model, as described previously.

Page 11, line 30: the results vary between catchments as shown in fig 8 I don't find this figure very useful, the reader is unable to interpret the coloured dots. An alternative, aside the boxplots, could be some scatter plots (statistical Q100 and Q1000 v/s standard and stochastic PQRUT, statistical QS v/s standard and stochastic PQRUT, etc.).

We have included a figure that shows the QS for each catchment.

Page 11, line 32: we can conclude that the performance of the standard PQRUT model is poorer than the performance of the statistical flood frequency analysis and the stochastic PQRUT model The results which ground this conclusion are not explicitly presented. The only clue given to the reader is the Figure 8 which only presents the distribution of QS scores for FFA and stochastic PQRUT. The results, in terms of QS score as well as confidence interval, should be presented in a table and in an adequate figure.

In addition to the figure that shows the QS, we have included a figure that shows the number of models that are within the confidence interval for Q100 for each catchment.

Page 12, line 3: The violin plots (fig. 9) See remarks on Figure 9 below.

Figure 9 shows both violin plots and boxplots (overlayed in gray). In order to increase the readability of the figure, only the boxplots are now plotted.

Page 12, line 7: Reasons for this may be that higher precipitation intensity or snowmelt is used To assess this, the values of the reference hyetographs used in standard PQRUT deserve to be presented and compared to the simulated precipitation values of stochastic PQRUT (like the values of the Table 2 for Q100).

The results for Q1000 obtained from the stochastic PQRUT are now lower (after increasing the number of simulations). Page 12, line 8: the absolute differences between the two methods are larger in catchments with lower temperature (fig. 9) I wonder how this can be deduced of illustrated by Figure 9, it is more likely somehow in Figure 10.

Apologies for this error, the figure numbers have now been updated.

Page 12, line 17: which might be due to the uncertainty in estimating the parameters for the GEV distribution I don't understand this interpretation which appears rather quick and subjective to me. We agree, this section has been deleted in the revised manuscript.

Page 12, lines 18-21: This using of the study of Rogger et al. (2012) is off topic for me here, as it is based on Gumbel, whereas the FFA is done here with GEV, which is more flexible.

This is true, this is the reason that in the paper we specifically discuss the fact that the Gumbel distribution was used in the study of Rogger et al. (2012). This section has been deleted.

§4- Conclusions Page 13, line 10-15: Another modelling option could be to run the event-based simulation with the DDD model, already used for the initial condition. In that case, an hourly version of DDD should also be calibrated (with local observations or regionally), in compatibility with the daily version used for initial conditions. I am not fully aware of the potential difficulties of this, but it would be a more homogeneous approach in terms of hydrological modelling. Any comment about this?

Yes, this is a possibility. However, in large catchments it is not as important to use a subdaily timestep, as the peak and daily flows are similar.

Page 13, line 28: easily incorporate the uncertainty associated with this choice This is a very good remark: the stochastic process here adequately models a variable which, when represented in a deterministic way (i.e. fixed initial conditions), appears as highly uncertain.

Yes, this has also been discussed in several other studies.

Page 13, line 31: based on an assessment of the uncertainty characterizing the individual methods This is an interesting suggestion, but it has to be added that a proper expression of uncertainty for a rather sophisticated method like stochastic PQRUT is far from trivial, and is still to be investigated...

This sentence has been deleted.

Tables Table 1 Units are missing, as well as legend of the columns in the caption. Table 2: Units are missing, precipitations could be rounded to the next mm. For Krinsvatn, the probability to find the Q100 events in one season or the other could be provided.

The two tables have now been revised.

Figures Figure 3: Not very informative, re-scaling storm hyetograph is not something difficult to understand. A set of different " typical" hyetographs could instead be presented for the three catchments, ideally illustrating the potential diversity of storm dynamic. A new figure was added. Figure 4: " for Krinsvatn catchment" could be added in the caption, as well as the number of observed and simulated events.

The figure has been updated.

Figure 6: The remarkable return periods (10, 100 and 1000 yr.) should be distinguished in the plots (by a bolder vertical line for example).

The figure has been updated.

Figure 7: There are too many distributions in the plots, their interpretation is not easy. Two plots could be edited for each catchment, having for example only the " calibrated" simulation in common. An uncertainty band around the "

calibrated" simulation would be useful to assess the intrinsic uncertainty of the simulation process.

The figure has been updated. Figure 8: See comment of page 11, line 32.

This will be added.

Figure 9: I am not convinced by the usefulness of the violin plots here considering the limited number of values per scores (20 catchments). Box plots with outliers would have been sufficient and more readable. Captions of the methods sometimes overlap.

**Reply to reviewer 2**

General Comments: As I understood, the main purpose of the work is to propose a methodology to overcome the limitations of more commonly applied event based modelling for flood frequency estimations by a stochastic modelling of preconditions, including SWE, and meteorological input. The individual modelling of the different aspects are described in the manuscript, however, it is hard to follow how the different parts are connected. A preceding sub-section with a less detailed step by step explanation of methodology, maybe including a schematic illustration (inputs/ models/ methods / output), could help to better explain the methodology.

A similar comment has been made by reviewer 1. We have added a step by step description and also included a diagram to illustrate the link between the different steps.

For the validation of the disaggregation procedure the disaggregated data were compared against hourly station data. Is this correct? I would be interesting to see how well the disaggregation procedure was performing (For example showing a obs-sim, QQ-plot). It is stated that it works better than equal divisions which is not surprising. What is the advantage of the further equal division to 1h if it is stated that 3-houers are already enough? Further, it is not obvious why the gridded seNorge.no Data are matched to the HIRLAM data if they are in the needed temporal resolution already?

The HIRLAM data is a hindcast dataset with a spatial resolution of around 10 km2 and a temporal resolution of 3 hours. The gridded seNorge data is obtained by triangulation of the observed rainfall dataseries; it has a spatial resolution of 1km2, and a temporal resolution of 24-hours. As the HIRLAM data has a higher temporal resolution than the seNorge data, the HIRLAM data was used to disaggregate the seNorge data to a 3-hour timestep. The performance, including the validation of the disaggregation procedure, is described in Vormoor and Skaugen (2013).

For the work presented here, the precipitation data were further disaggregated to a 1-hour time step by dividing into three equal parts. This was simply done for convenience, as the PQRUT model has previously been calibrated relative to 1-hour streamflow data and similar climate input data (i.e. 1-hour data derived from 3-hour data by dividing into three equal parts). A similar comment was raised by reviewer 1, and this section will be revised to include the above clarifications.

We have revised the paragraph as:

The HIRLAM atmospheric model for northern Europe has a 0.1 degree resolution (around 10 km2) and we used a temporal distribution of three hours. The HIRLAM data set was first downscaled to match the spatial resolution of the seNorge data and the precipitation of the HIRLAM data was rescaled to match the 24-hour seNorge data (Vormoor and Skaugen, 2013). Then, these rescaled values were used to disaggregate the seNorge data to a 3-hour time resolution. The method was validated against 3-hour observations, and the correlation of the method was found to be higher than that obtained by simply dividing the seNorge data into eight equal 3 -hourly values (Vormoor and Skaugen, 2013). These datasets were further disaggregated to a 1-hour time step by dividing into three equal parts to match the time resolution of the streamflow data.

A 1000 years event is extrapolated from daily observation series (length not further specified). Furthermore the results are then multiplied by empirical factors, to match sub-daily peak flows. I am not aware of the engineering practice in Norway, however, I am not sure about the meaning of the results by this extreme extrapolation and at least this

**should be critically discussed.**

The fitting of an extreme value distribution to estimate the return level for periods longer than the length of a time series is a standard procedure, both in hydrological investigations and in engineering practise. As suggested by the reviewer, the uncertainty of the estimates does increase significantly for longer return periods, relative to the length of record. The length of the daily streamflow series considered here, however, justifies the use of an 'at-site' (cf. a regional) flood frequency analysis as the minimum length is 31 years, while the median is 65 years of data. The following sentences will be added to the text:

In addition, the length of the daily streamflow series justifies the use of at-site flood frequency analysis (Kobierska et al., 2017); the minimum length is 25 years, while the median is 65 years of data. However, it is expected that the uncertainty will be high when the fitted GEV distribution is extrapolated to a 1000-year return period. The 1000-year return period is used here, however, as it is required for dam safety analyses in Norway (e.g. Midttømme, et al., 2011; Table 1). More robust, but potentially less reliable, estimates could be obtained using a 2-parameter Gumbel, rather than a 3-parameter GEV distribution (Kobierska et al., 2017).

The sensitivity analysis is interesting, however, also confusing including Figure 7 and Table 3. It is not obvious on what basis the percentage difference is calculated. This is also not clear in the follow up comparison of the methods. What exactly is the calibrated model? Also the section misses an explanation of the shaded area which is prominently displayed in Figure 7. Furthermore, the different precipitations settings tested are not well explained. A table, summarizing the different tested aspects, would help to guide the reader.

Thank you for these comments and suggestions. The shaded area represents the simulations based on the 5% and 95% confidence intervals for the regression equations for PQRUT. We have included a table to describe the set up and we have also revised the paragraph that describes the sensitivity analysis as follows:

A sensitivity analysis was performed for the three test catchments, Hørte, Øvrevatn and Krinsvatn, in order to determine the relative importance of the initial conditions, precipitation, the parameters of PQRUT, the effect of the random seed and length of simulation on the flood frequency curve. To test the sensitivity of the model, we have used several different model runs and calculated the percentage difference of each of these model runs relative to the standard model setup, as shown in Fig.8. More detailed information on the set up is given in Table 3. As these catchments are located in different regions and exhibit different climatic and geomorphic characteristics, we hypothesize that the flood frequency curve will be sensitive to different parameters and hydrological states, as well as local climate and catchment characteristics. The results are summarised in Table 4.

The Figures and especially the captions should be improved, as they are often not self-explanatory. This includes also missing units, labels and abbreviations. Maybe consider a professional language proof reading.

In the revised version of the article, we have tried to improve the language usage.

Specific Comments: The abbreviation PQRUT, used from beginning (abstract), is not introduced on page 4 or rather page 8. Please declare the meaning of PQRUT first time mentioned.

The abbreviation PQRUT comes from P-precipitation, Q- discharge and RUT -routing, this has been updated.

The characterizations of catchments and chosen abbreviations are introduced on P4 and repeated later (P5, I5) without brackets (e.g. "sparse vegetation over tree line (B)" and "sparse vegetation over tree line B"). Either use brackets throughout the manuscript or only use the abbreviation. Additionally by choosing more selfexplanatory abbreviations or using full words (eg. forest; marsh), would be easier to understand, especially in Table 1.

The abbreviation are now explained in the table caption. P.5 1.15: The last sentence does not contain important informations and could be omit

We prefer to keep this sentence as it gives useful information on how the data is derived and increases the reproducibility of the study.

**P.6 l.1: The addition ",which can be used for modelling in ungagged basins." could be omitted, as it seems not connected to the procedure.**

This sentence has been deleted.

P.6 l.2: A citation should be added to the DDD model or the corresponding R-package.

The reference is provided earlier in the manuscript, p4.

P.6 l.11: In my opinion, the meaning of the "critical duration" rather reflects the link between the duration and intensity of precipitation "events" of a certain probability, than to ensure the modelling of the complete flood hydrograph.

This is a good point, the sentence will be revised to:

When simulating flood response with an event-based model, it is important to specify the so-called critical duration (Meynink, W. J., Cordery, I. 1976) to ensure that the flood peak is correctly modelled. The critical duration is an important factor which effectively links the duration and the intensity of precipitation events of a given probability.

**P.6 1.32: "individual risk seasons could have been defined". One wonders why it was not done? If not so important for the result, please consider to omit this half sentence.**

This sentence has been deleted. We used this season definition to match the seasonal definition used by the Norwegian Meteorological Institute.

**P.8 l.12-17: Please check grammar and style of the section.**

This section has been revised to also address the issues raised by reviewer 1 as follows.

The PQRUT model was calibrated for the 45 highest flood events by using the DDS (Dynamically Dimensioned Search) optimization routine (Tolson and Shoemaker, 2007) and the Kling Gupta efficiency (KGE) criterion (Gupta et al., 2009) as the objective function. An additional variable, the soil deficit, lp, was introduced to account for initial losses to the soil zone. The reason for this is that, even though fully saturated conditions are assumed when the model is used to estimate PMF or other extreme floods with low probabilities, the model needs to account for initial losses when actual (more frequent) events are simulated. This procedure is described in more detail in Filipova et al. (2016). In addition, regional values can be used in ungauged or poorly catchments (Andersen et al., 1983; Filipova et al., 2016).

**P.8 l.28: Was there a specific reason for using the "Gringorten plotting" position?**

The Gringorten plotting positions provide unbiased quantile estimates for the Gumbel distribution. In this case, we don't know the distribution. However, the difference between the plotting positions is usually higher for the low and high quantiles. As reviewer 1 suggests we have increased the number of simulations. This means that differences derived from plotting position formulas will be relatively small when estimating the 1000 -year return period.

**P.12 l.3: A more detailed explanation what exactly is analyzed here is missing.**

The sentence will be revised to: A comparison of the stochastic PQRUT with the standard methods for flood estimation shows that there is a large difference between the results of the three methods for both Q100 and Q1000 (Fig. 12 and 13).

P.12 l.25: Maybe the catchment steepness should be introduced in the section "study area".

We have now added the catchment steepness to the "study area" section.

P.13 l.25: Why is it peak to volume? I thought it is daily mean to daily max discharge?

This refers to converting the daily volume (obtained from the daily mean) to the peak value.

Table 1: Missing units. Furthermore, the variables could be sorted and clustered more logically (e.g. temperature and precipitation; Q and AMAX).

Thanks, the units have now been included.

Table 3: Why are 100 values sampled? Does T mean the threshold Parameter Trt ?

For the sensitivity analysis we used 50 samples, as larger number will increase the computational time. We assume that this number is sufficient to calculate the intervals. Trt refers to the parameter of the PQRUT model, thanks for spotting this error.

**Figure 2: Labels and units are missing**

This has been corrected for the revised version of the manuscript. Figure 3: Labels and units are missing

This has been corrected for the revised version of the manuscript.

Figure 7: is confusing because of the large number of different colored lines. Maybe two plots can help to distinguish between the different aspects as for example the precipitation input and other aspects. The legend is confusing as well. GDP was fitted to what? Y-Axes should start at 0, x-axes missing a label and to be consistent with the rest of the work it should not exceed 1000.

This issue was also raised by reviewer 1 and the figure will be improved and revised based on the newer, longer simulations. Figure 8: It is impossible to distinguish between 20 colors. Do the colors have any meaning? If they should be recognizable, numbering would be a better option. The numbers could then also be used in Figure 10, so the link between the performance of the model and the results are given.

The colors just represent different catchments but also as reviewer 1 suggests. We have replaced this figure with a boxplot that shows the performance of the methods at each catchment.

Figure 9: What exactly is shown in the plots. Please add a more detailed explanation.

The figure has now been replaced. Instead, now we are using boxplots to show the differences in the performance of the three methods.

**Figure 10: The scale "percentage difference" should be unambiguous. The base of the "difference" should be clarified.**

We have revised the description of the figure as:

[revised manuscript text omitted]
 . The methods give different results in many of the catchments due to the large uncertainty in both the event-based model and the statistical flood frequency analysis. To better understand the differences between these methods, a sensitivity analysis of the stochastic PQRUT is performed by considering the effect of the initial conditions, model parameters and rainfall intensity on the flood frequency curve. for the 100- and 1000 -year return
- 20 period.

**2 Stochastic event-based flood model**

The stochastic event-based model proposed here involves the generation of several hydrometeorological variables: precipitation depth and sequence, temperature the temperature sequence during the precipitation event, antecedent discharge, the initial discharge, and the antecedent soil moisture conditions and antecedent snow water equivalent. A simple 3-parameter flood model PORUT (Andersen et al., 1983) is used to simulate the streamflow hydrograph for a set of randomly generated selected

- 25 model PQRUT (Andersen et al., 1983) is used to simulate the streamflow hydrograph for a set of randomly generated selected conditions based on the hydrometeorological variables hydrometeorological variables. After this procedure is completed 100,000 times for each of the four seasons, the results are combined and a flood frequency curve is constructed from all of the simulations using their plotting positions. As the method requires initial values for soil moisture and snow water equivalent, i.e variables which generally cannot be sampled directly from climatological data and which depend on the sequence of pre-
- 30 cipitation and temperature over longer periods, the Distance Distribution Dynamics (DDD) hydrological model (Skaugen and Onof, 2014) was calibrated and run for a historical period to produce a distribution of possible values for testing the approach. The method (also shown in Fig. 1) can be outlined in summary form as follows:

- 1. Extract flood events for a given catchment and identify the critical storm duration For each season:
- 2. Aggregate the precipitation data to match the critical duration for the catchment
- 3. Extract POT precipitation events and fit a GP distribution
- 5 4. Fit probability distributions for the initial discharge, soil moisture deficit and SWE values for the season
  - 5. Generate precipitation depth from the fitted GP distribution
  - 6. Disaggregate the precipitation depth to a 1 hour time step by matching the dates of the identified POT flood events (from step 3) to dataseries of precipitation with hourly timestep.
  - 7. Sample a temperature sequence by matching the dates of the identified POT flood events (from step 1) to the dataseries
- 10

20

- of temperature with hourly timestep
  - 8. Sample initial conditions for snow water equivalent (*SWE*), soil moisture deficit and initial discharge from their distributions (step 5), accounting for co-variation using a multivariate normal distribution
  - 9. Simulate streamflow values using the calibrated PQRUT model for the sample event
  - 10. Repeat steps 6.-9. 100 000 times
- 11. Estimate the annual exceedance probability from the total of 400 000 (100 000 for each season) samples using plotting positions

The study area and data requirements for the proposed method are described in section 2.1, while section 2.2 describes the method for determining the critical duration, and section 2.3 and 2.4 describe the generation of antecedent conditions and meteorological data series. The hydrological model is presented in section 2.5and, the method for constructing the flood frequency curve is outlined in section 2.6 and the sensitivity analysis is presented in section 2.7.

**2.1 Study Area and Data requirements**

**2.1.1 Catchment selection and available streamflow data**

The study area consists of a set in Norway, consisting of a dataset of 20 catchments located throughout Norway (fig 2)the whole country, is shown in Fig 2. All catchments have at least 10 years of hourly discharge data, and in all cases the length

25 of the daily flow record is considerably longer than 10 years. All selected catchments are members of the Norwegian Bench Mark dataset (Fleig, 2013), which ensures that the data series are unaffected by significant streamflow regulation and have discharge data of sufficiently high quality suitable for the analyses of flood statistics. The catchment size was restricted to small and medium-sized catchments (maximum area is 854 km2), as the structure of the 3-parameter PQRUT model does not take into account all of the storage processes within the catchment which possibly the longer-term storage processes which can contribute to delaying runoff the runoff response during storm events. Previous applications of PQRUT in Norway indicate that this shortcoming is most problematic for larger catchments. Discharge datasets with both daily and hourly time steps were obtained from the national archive of streamflow data held by NVE (https://www.nve.no/). The catchments were delineated

- 5 and their geomorphological properties were extracted using the NEVINA tool:http://nevina.nve.no, except for *Q*, which was calculated using the available streamflow data and *P*, which was calculated using available gridded data (further details are given in 2.1.2 below). In order to illustrate the application of the method, we have selected three catchments which can be considered representative for different flood regimes in Norway: Krinsvatn in western Norway, Øvrevatn in northern Norway and Hørte in southern Norway (fig-Fig. 2).
- 10 Table 1 summarises the climatological and geomorphological properties of these three catchments, including: area (A in km2), mean annual runoff (Q in mm /yearyear-1), mean annual precipitation (P in mm /yearyear-1), mean elevation (*Hm50*), percent of percentage forest-covered area (*For*), percent of percentage marsh-covered area (M), percent percentage area with sparse vegetation above three tree line (B), 'effective' lake percent percentage (*Lk*)and, catchment steepness (*Hl*) and the mean annual temperature in the catchment (*Temp*). The effective lake percent (*Lk*) is used to describe the ability of water bodies to
- 15 attenuate peak flows such that lake areas which are closer to the catchment outlet have a higher weight than those near the catchment divide. It is calculated as  $\frac{\sum A_i \times a_i}{A^2} \times 100$ , where  $aig_i$  is the area of lake i,  $Ai_{-1}$  is the catchment area upstream of lake i and A is the total catchment area. The dominant land cover for Krinsvatn and Øvrevatn is sparse vegetation over tree line*B*, while the land cover for Hørte is mainly forest*For*. The effective lake percent *Lk* is insignificant for Hørte and Øvrevatn, but for Krinsvatn, the *Lk* is higher and the area covered by marsh, *M*, is more than 109%. The catchment Krinsvatnsteepness
- 20 (HI) (defined as (Hm75-Hm25)/L, where L is the catchment length and Hm25 and Hm75 are the 25 and 75 quantiles of the catchment elevation) is highest for Hørte (18.7 m/km) and lowest for Krinsvatn (5.4 m/km). The catchment Krinsvatn, being located near the western coast of Norway, has a much higher mean annual precipitation (*P*), i.e. an average of 2354 mm /year, compared to year-1, in comparison with Hørte (1261mm/year1261 mm year-1) and Øvrevatn (1558 mm /yearyear-1). The dominant flood regime for Krinsvatn is primarily rainfall-driven high flows, as the catchment is located in a coastal area
- and is characterised by high precipitation values and an average annual temperature of around  $4^{0}_{\sim}$  C. The highest observed floods, however, also have a contribution from snowmelt. The season of the AMAX (annual maxima flood) is the winter period, i.e. December – February, although high flows can occur throughout the year. Hørte has a mixed flood regime with most of the AMAX flood events in the period September–November, but in some years annual flood events occur in the period March–May and are associated with rainfall events during the snowmelt season. Øvrevatn has a predominantly snowmelt flood
- 30 regime with most AMAX flood events occurring in the period June August, due to the lower temperatures in the region such that precipitation falls as snow during much of the year. The eatchments were delineated and their geomorphological properties were extracted using the NEVINA tool:http://nevina.nve.no, except for *Q*, which was calculated using the available streamflow data and *P*, which was calculated using available gridded data (further details are given in 2.1.2 below).

**2.1.2 Available meteorological data**

Data for temperature and precipitation with daily time resolution were obtained from seNorge.no. This dataset is derived by interpolating station data on a 1 km2 grid and is corrected for wind losses and elevation (Mohr, 2008). In addition, meteorological data with a sub-daily time step is needed for calibrating the PQRUT model, as many of the catchments have fast response times.

- 5 For this, precipitation and temperature data with a three-hour resolution, representing a disaggregation of the 24-hour gridded seNorge.no data using the HIRLAM hindcast series (Vormoor and Skaugen, 2013), were used. The HIRLAM atmospheric model for northern Europe has a 0.1 degree resolution (around 10 km2) and we used a temporal distribution of three hours. The HIRLAM data set was first downscaled to match the spatial resolution of the seNorge data (Vormoor and Skaugen, 2013). The and the precipitation of the HIRLAM data was rescaled to match the 24-hour seNorge data , and (Vormoor and Skaugen, 2013).
- 10 . Then, these rescaled values were used to disaggregate the seNorge data to a 3-hour time resolution. The method was validated against 3-hour observations, and the correlation of the method was found to be higher than that obtained by simply dividing the seNorge data into eight equal parts 3 -hourly values (Vormoor and Skaugen, 2013). These datasets were further disaggregated to a 1-hour time step using a uniform distribution by dividing into three equal parts to match the time resolution of the discharge data, although a 3-hour time step could also be used streamflow data.

**15 2.1.3 Initial conditions**

The stochastic PQRUT method requires time series of soil moisture deficit, *SWE* and initial discharge. These data series are used to generate initial conditions, which serve as input in construct probability distribution functions for generating initial conditions for the event-based PQRUT modelsimulations. Sources of these data can be e.g. remotely sensed data (see, for example, the review provided in Brocca et al. (2017)) or gridded hydrological models, which can be used for modelling in

- 20 ungauged basins. In this study, the DDD hydrological model was used to simulate these data series. The DDD model is a conceptual model that includes snow, soil moisture and runoff response routines and is calibrated for individual catchments using a parsimonious set of model parameters. The snowmelt routine of DDD model uses a temperature-index method and accounts for snow storage and melting for each of 10 equal area elevation zones. The soil moisture routine is based on one dynamic storage reservoir, in which we find both the saturated- and the unsaturated zone, having capacities which vary in time.
- 25 The flow percolates to the saturated zone if the water content in the unsaturated zone exceeds 0.3 of its (dynamic) capacity. The response routine includes routing of the water in the saturated zone using a convolution of unit hydrographs which are based on the distribution of distances to the nearest river channel within the catchment and from the distribution of distances within the river channel.

**2.2 Critical Duration**

30 When simulating flood response with an event-based model, it is important to specify the so-called critical duration (Meynink and Cordery, to ensure that the complete flood hydrograph is modelledflood peak is correctly modelled. The critical duration is an important guantity which effectively links the duration and the intensity of precipitation events of a given probability. In order to

determine the length of the precipitation input series producing the most extreme flows, a critical duration for storm events, defined as the duration that results in the highest observed peak value, had to be determined for each catchment. To determine the critical duration, flood events over a certain from the daily time series over a quantile threshold (in this case, 0.9) were extracted. The POT (peak over threshold) flood events were considered to be independent if they were separated by at least

- 5 seven days of lower values than values below the threshold. The day with the maximum value (peak ) of the peak value of streamflow was then identified for each event. The peak values were tested for correlations with the precipitation on the day of the peak flow and on days -1, -2 and -3 before the peak. The critical duration was determined as the number of days in which the correlation between the precipitation and the streamflow was higher than 0.25. This threshold value was selected because it gave realistic durations for the catchments in the study area. At firstAs an alternative approach, the critical duration was also
- 10 set to equal the number of days for which the correlation was significant at p=0.01. This method resulted, however, which howeverresulted in very long durations, in some cases. A possible reason for this is that if there are only a few observations, even relatively low Pearson correlation coefficients can produce statistically significant p-values. In some catchments (mostly those having a snowmelt flood regime), no significant correlation was found between discharge and precipitation, and in this case the critical duration was fixed to 24 hours determined by considering only flood events in the September-November (SON)
- 15 season in which most events are caused by rainfall (in this case, during the autumn). If the critical duration was more than one day, the precipitation was aggregated to the critical duration by applying a moving window to the data series. For Hørte and Øvrevatn, the critical duration was set to found to be 24 hours and for Krinsvatn to it was found to be 48 hours (fig-Fig. 3).

**2.3 Precipitation and temperature sequence generation**

In addition to the critical duration of the event, the sequence of the input data must be prescribed generated for the stochastic simulation. Snowmelt can be important in the catchments considered in this study, so both the sequence of precipitation and temperature must be considered. In order to account for seasonality, the meteorological data series were first split into standard seasons: DJF, MAM, JJA and SON. In this way, we ensure that more homogeneous samples are used to fit the statistical distributions. Although the season at risk could have been defined for each catchment individually (e.g. Paquet et al., 2013), the standard season definition was used for all catchments. Precipitation events over a threshold (POT events) were identified in the

- 25 24h precipitation data precipitation dataseries and a Generalized Pareto distribution was fitted to the series of selected events. In order to select choose a threshold value for event selection, two criteria were used: 1) the threshold must be higher than the 0.93 quantile, and 2) the number of selected events must be between two to and three per season. Although other methods for threshold selection exist, such as the use of mean life residual plots, the described method gives adequate is much simpler to apply and gives acceptable results (e.g. Coles, 2001). The selected threshold varied between the 0.93 to 0.99 quantiles,
- 30 depending on the season and catchment. In addition, storm hyetographs and the exponential distribution is often fitted to POT events, as it can give more robust results than the GP distribution. Figure 4 shows the return levels calculated from the GP and Exponential distributions and the empirical return levels and demonstrates that it is appropriate to use these models. In this case, we have preferred to use the GP due to the inclusion of the shape parameter for describing the behaviour of the highest quantiles. A exponential distribution, however, could also be used, as could a compound weather pattern-based distribution

such as the MEWP distribution (e.g. Garavaglia et al., 2010; Blanchet et al., 2015). In addition, a temperature sequence with a 1-hour time resolution were was identified from the disaggregated seNorge data, introduced in section ??? 2.1.2, and extracted for each POT events. Using the fitted Generalized Pareto (GP) distribution, event. The precipitation depths were simulated and the storm-generated (for 100,000 events) from the fitted GP distribution for each season. Storm hyetographs were used to

5 disaggregate the precipitation values as follows: a storm hyetograph was first sampled (fig 9) from the extracted hyetographs for the selected POT precipitation events (by matching the dates of the selected POT precipitation events to the disaggregated seNorge datataseries), taking into account seasonality, and the ratios between the 1-hour and the total precipitation for the event were calculated according to:

$$P_h sim = \frac{P_i}{sum(\underline{PP_i})} P_d sim \tag{1}$$

10 where Phsim is the simulated 1-hour precipitation intensity Pdsim is the daily intensity simulated daily intensity and Pi is the 1-hour disaggregated SeNorge intensity. The calculated ratios were then used to rescale the simulated values (fig 9).

**2.4 Antecedent snow water equivalent, streamflow and soil moisture deficit conditions**

In order to determine the underlying distribution for various antecedent conditions, the relevant quantities were extracted from simulations using based on the DDD hydrological model of Skaugen and Onof (2014). The model was calibrated for the

- 15 selected catchments at a daily timestep using a MCMC routine (Petzoldt, 2010). Output from DDD model runs were used to extract values for the initial streamflow, snow water equivalent (*SWE*) and soil moisture deficit, prior to the seasonal at the onset of the previously selected seasonal flood POT events. It is important to note that simulated values for the soil moisture deficit are used. However as described in Skaugen and Onof (2014), the model provides realistic values in comparison with measured groundwater levels. The POT event series used for this is the same as that used for identifying the critical duration
- 20 (described in section 2.2).

After extracting the initial conditions, the correlation between the variables was tested for each season for each catchment. As the correlation between the variables is in most cases significant, the variables were jointly simulated using a truncated multivariate multivariate normal distribution. In order to achieve a normality for the marginal marginal distributions, the SWE and the discharge were log-transformed. In the spring and summer, the SWE is often very low or 0 in some catchments. If the

25 proportion of non-zero values, *pp*, was greater than 0.3 (around 15 observations), the values were simulated using a mixed distribution as:

$$F(x) = pG_1(x) + (1-p)G_2(x)$$
(2)

where  $G_1$  and  $G_2$  represent represents the multivariate normal distribution with discharge, soil moisture deficit and *SWE* as variables and (denoted as x) and  $G_2$  the bivariate normal distribution for the discharge and soil moisture, respectively..., The

[revised manuscript text omitted]

**2.7 Sensitivity analysis**

A sensitivity analysis was performed for the three test catchments, Hørte, Øvrevatn and Krinsvatn, in order to determine the relative importance of the initial conditions, precipitation<del>and</del>, the parameters of PQRUT, the effect of the random seed and length of simulation on the flood frequency curve. To test the sensitivity of the model, we have used several different model runs and calculated the percentage difference of each of these model runs relative to the standard model setup, as shown in Fig.8. More detailed information on the set up is given in Table 3. As these catchments are located in different regions and exhibit different climatic and geomorphic characteristics, we hypothesize that the flood frequency curve will be sensitive to different parameters , and hydrological states, precipitation, snow and catchment characteristics well as local climate and catchment characteristics. The results are presented in figure 8 and summarised in table summarised in Table 4.

- 5 Considering the effect of the initial conditions, using fully saturated conditions results in the slight overestimation for all catchments of flood values, as expected, and the impact is higher at lower return periods. In addition, Øvrevatn shows higher sensitivity (around 30% for *Q1000*) to the initial soil moisture conditions than the other two catchments. A possible explanation is that the baseflow index (*BFI*) is higher (*BFI*=0.6) for Øvrevatn, than the *BFI* for Krinsvatn (*BFI*=0.4) and Hørte (*BFI*=0.5). This indicates that the runoff at Øvrevatn is less responsive to the rainfall input. Similarly, a sensitivity analysis by
- 10 Svensson et al. (2013) shows that high sensitivity of floods to soil moisture deficit is more present for permeable catchments, where the *BFI* is also high. 
[revised manuscript text omitted]

**3.2 Discussion**

- 15 A comparison of the three methods stochastic PQRUT with the standard methods for flood estimation shows that there is a large difference between the results of the different methods (fig-three methods for both *Q100* and *Q1000* (Fig. 12 and 13). The violin plots (figboxplots (Fig. 12) show that the stochastic PQRUT method gives slightly lower results on average than the standard PQRUT model for *Q100* and *Q1000*. This is probably due to assuming fully saturated conditions when applying the standard PQRUT for *Q100*, which might not be realistic for some catchments. For example, the results for the initial conditions
- 20 for the three catchments, presented in section 2.6, show that the soil moisture deficit is larger than 0. However, when the results are compared for 0 for *Q1000, the stochastic PQRUT gives slightly higher results. Reasons for this may be that higher precipitation intensity or snowmelt is used. Q100.* Furthermore, the absolute differences between the two methods are larger in catchments with lower temperature (figFig. 12). This indicates that the performance of the standard PQRUT model is worse in catchments with a snowmelt flood regime, which might be due may be due either to the difficulty in determining snowmelt
- 25 contribution or to the poorer performance of the regional parameters in catchments with a snowmelt flow regime. Although showing the same it shows a similar pattern, the standard PQRUT model, implemented using calibrated parameters results in much less spread than the implementation using the regionalised parameters, when compared to both the GEV distribution and the stochastic PQRUT model. This means that the hydrological model can introduce a large amount of uncertainty, as also indicated by the sensitivity analysis described in section 2.7 and previous results presented by Brigode et al. (2014).
- 30 The difference differences between the stochastic PQRUT model and GEV is the GEV fits are much smaller than the difference differences between the standard PQRUT and GEV model and the GEV fits, even when calibrated parameters are used . In general, the stochastic PQRUT model gives higher values than the GEV distribution, which might be due to the uncertainty in estimating the parameters for the GEV distribution. For example, the study by Rogger et al. (2012) shows that the

flood frequency analysis based on fitting a Gumbel distribution to AMAX series underestimates high flows in catchments with a high storage capacity, where a step change in the flood frequency curve occurs. The results of the study by Rogger et al. (2012) can be explained by the fact that the Gumbel distribution, which has a shape parameter of 0 and so is not as flexible as the GEV distribution. In this study we find very low correlation between the difference between the stochastic PQRUT and the GEV

- 5 distribution and catchments with high values of the effective lake index or the percentage of the catchment covered by marsh (fig 13), where we would expect that the storage capacity is higher. In addition, the for the PQRUT modelling. The differences are larger (i.e. the stochastic PQRUT results are lower, as shown in fig Fig. 13) in western Norway where P and Q are higher and for eatchments with higher eatchment steepness Hl (defined as (Hm75-Hm25)/L, where L is the catchment length and Hm25 and Hm75 are the 25 and 75 quantiles of the catchment elevation)steeper catchments, i.e. with a higher value of Hl. A
- 10 reason for this might be that the empirical ratios that are used to convert daily to peak flows in these catchments are inaccurate and possibly too high.

Similarly to the violin plots, fig 13 boxplots, Fig. 13 also shows that the results of the stochastic PQRUT closely match the GEV distribution fits with differences within 50% for most locations. There is no clear spatial pattern in the differences between estimates based on the GEV distribution and on the standard PQRUT model, except for the catchments in mid-Norway.

15 i.e. Trøndelag (including catchment Krinsvatn), where the GEV distribution produces higher results. However, a much larger sample of catchments is needed to assess whether there is a spatial pattern in the performance of the methods.

**4 Conclusions**

In this article, we have presented a stochastic method for flood frequency analysis based on a Monte Carlo simulation to generate rainfall hyetographs and temperature series to drive a snowmelt estimation, along with the corresponding initial conditions.

- 20 A simple rainfall-runoff model is used to simulate discharge, and plotting positions are used to calculate the final probabilities. In this way, we can generate thousands of flood events and use the empirical distribution instead of extrapolating a statistical distribution fitted to the observed events. The approach thereby gives significant insights into the various combinations of factors that can produce floods with long return periods in a given catchment, including combinations of factors that are not necessarily well represented in observed flow series. It is thus a very useful complement to statistical flood frequency analysis
- 25 and can be particularly beneficial in catchments with shorter streamflow series compared to the precipitation record as well as in ungauged catchments.

In order to apply the method, we assume that the precipitation and temperature series are not significantly correlated with the initial conditions, which allows us to simulate them as independent variables. Although we have not performed a statistical analysis, the independence between the flood-precipitation events and the initial conditions has been verified by e.g.

30 Paquet et al. (2013) (Paquet et al., 2013). Due to the considerable seasonal variation in the initial conditions, seasonal distributions were used. In addition to obtaining more homogeneous samples, this allows one to check for a check of the seasonality of the flood events, which can be of interest in catchments with a mixed flood regime. In this study, we have used a GP distribution to model the extreme precipitation. However, if only shorter precipitation dataseries are available, the exponential distribution

or even regional frequency analysis methods may provide more robust results. A limitation of the method is that PQRUT can only be used for small and medium-sized catchments, since its three parameters cannot take into account the spatial variation of spatial variation in the snowmelt and soil saturation conditions within the catchment. However, for the catchments presented in this study (all with a catchment area under 850 km2), the model produces relatively good fits to the observed peaks, even

5 though it uses a very limited number of parameters. For example, a semi-distributed temperature-index snowmelt model, such as the one used in HBV (Sælthun, 1996), may improve the results in some catchments, though this would also increase the amount of data required.

In this study, initial conditions based on simulations using a hydrological model (DDD) were used. However, in other applications, initial conditions may be based on remotely-sensed data, or on the output of gridded hydrological models. This

- 10 is particularly important for the application of the method in ungauged basins This requires that this model is calibrated at each catchment. Considering the results of the sensitivity analysis, the quality of the initial conditions is not as important as that of the precipitation data for the estimation of extreme floods (with return periods higher than 100 years). This means that if no other data is available, the output of gridded hydrological model could be considered -as a source of this input data. Alternatively, remotely sensed data can be used for soil moisture and the snow water equivalent while regional values for the
- 15 initial discharge can be derived. This for example, can be an option in ungauged basins.

The stochastic PQRUT model was applied to 20 catchments, located in different regions of Norway and was compared with the results of the statistical flood frequency analysis and the event based PQRUT model. This comparison shows that there are large differences between the methods. Major sources of uncertainty for the flood frequency analysis are the use of short data series and the empirical peak to volume ratios that were used to calculate the instantaneous flow. There is also uncertainty in the

- 20 event-based rainfall-runoff simulation method because of difficulties in assigning the initial conditions and in calibrating the rainfall-runoff model. This first of these is a reason why the use of a stochastic model is important, as it can simulate multiple initial conditions and easily incorporate the uncertainty associated with this choice. event-based PQRUT method which is today used in standard practice. Due to the high uncertainty in estimating extreme floods, the application of different methods most often the different methods produces differing results, as in often the case in practical applications. However, in this work
- 25 we have shown that the stochastic PQRUT model gives estimates which generally are more similar to those obtained using a statistical flood frequency analysis based on the observed annual maximum series than are estimates obtained using a standard implementation of PQRUT. As it is not possible to test the reliability of estimates for the 500- or 1000-year flood (due to length of the observed streamflow series relative to the return period of interest), the use of alternative methods for flood estimation, including stochastic simulations such as presented here, is an essential component of flood estimation in practice. A possible
- 30 way forward is to consider estimates based on different methods by calculating a weighted average of the various estimates , in which the weighting is based on an assessment of the uncertainty characterizing the individual methods .

[revised manuscript text omitted]

- 5 Ries, K. G.: The national streamflow statistics program: A computer program for estimating streamflow statistics for ungaged sites, in: Hydrologic Analysis and Interpretation Section A, Statistical Analysis, p. 37, 2007.
  - Rogger, M., Kohl, B., Pirkl, H., Viglione, A., Komma, J., Kirnbauer, R., Merz, R., and Blöschl, G.: Runoff models and flood frequency statistics for design flood estimation in Austria - Do they tell a consistent story?, Journal of Hydrology, 456-457, 30–43, https://doi.org/10.1016/j.jhydrol.2012.05.068, 2012.
- 10 Sælthun, N. R.: The "Nordic" HBV Model. Description and documentation of the model version developed for the project Climate Change and Energy Production, NVE Publication 7, Norwegian Water Ressources and Energy Administration, Oslo, p. 26, 1996.
  - Salazar, S., Salinas, J. L., García-bartual, R., and Francés, F.: A flood frequency analysis framework to account flood-generating factors in Western Mediterranean catchments, in: STAHY2017, September, p. 2017, 2017.

Salinas, J. L., Laaha, G., Rogger, M., Parajka, J., Viglione, A., Sivapalan, M., and Blöschl, G.: Comparative assessment of predictions in

- 15 ungauged basins Part 2: Flood and low flow studies, Hydrol. Earth Syst. Sci, 17, 2637–2652, https://doi.org/10.5194/hess-17-2637-2013, www.hydrol-earth-syst-sci.net/17/2637/2013/, 2013.
  - Schaefer, M. and Barker, B.: Stochastic Event Flood Model (SEFM), in: Mathematical models of small watershed hydrology and applications, edited by Singh, V. P. and Frevert, D., chap. 20, p. 950, Water Resources Publications, Colorado, USA, 2002.

Skaugen, T.: Studie av Skilltemperatur for snø ved hjelp samlokalisert snøpute, nedbør og temperaturdata, Tech. rep., NVE, Oslo, 1998.

- 20 Skaugen, T. and Onof, C.: A rainfall-runoff model parameterized from GIS and runoff data, Hydrological Processes, 28, 4529–4542, https://doi.org/10.1002/hyp.9968, 2014.
  - Svensson, C., Kjeldsen, T. R., and Jones, D. a.: Flood frequency estimation using a joint probability approach within a Monte Carlo framework, Hydrological Sciences Journal, 58, 8–27, https://doi.org/10.1080/02626667.2012.746780, http://www.tandfonline.com/doi/abs/10. 1080/02626667.2012.746780, 2013.
- 25 Tolson, B. A. and Shoemaker, C. A.: Dynamically dimensioned search algorithm for computationally efficient watershed model calibration, Water Resources Research, 43, https://doi.org/10.1029/2005WR004723, 2007.
  - Vormoor, K. and Skaugen, T.: Temporal Disaggregation of Daily Temperature and Precipitation Grid Data for Norway, Journal of Hydrometeorology, 14, 989–999, https://doi.org/10.1175/JHM-D-12-0139.1, http://journals.ametsoc.org/doi/abs/10.1175/JHM-D-12-0139.1, 2013.
     Wilson, D., Fleig, A., Lawrence, D., Hisdal, H., Petterson, L., and Holmqvist, E.: A review of NVE's flood frequency estimation procedures,

30 9, 2011.

---

## Referee Report (RR1)

**Referee Report of « A stochastic event-based approach for flood estimation in catchments with mixed rainfall/snowmelt flood regimes» (NHESS-2018-174, version 2)**

**General comments:**

This paper presents a significant enhancement of a Norwegian method for the estimation of extreme floods, based on an event-based rainfall-runoff simulation. It introduces a stochastic process for the assignment of the initial hydrological conditions before the simulated events, as well as for the intensity and the temporal dynamic of the simulated precipitation events. This method is compared to the initial method (which considers only a reference precipitation on given condition), and to a classical FFA.

The presented method is interesting, both in terms of methodology and statistical results. It is well explored, with a detailed sensitivity analysis.

Most of the issues raised during the review of the initial manuscript have been addressed by the authors, with new and/or complementary information provided, which now allows to better understand the method and the sensitivity analysis.

However, in some sections the writing remains perfectible, and some figures would deserve to be improved to ease their reading and their interpretation.

I would then recommend a minor revision of this paper, based on the detailed comments/suggestions provided to the authors in that follows.

**Detailed comments/questions:**

Quoted sentences are written in italic.

Page 4, line 4: *Australian Rainfall and Runoff 2016 (?)* Reference is missing.

Page 4, lines 4-5: *this method has not yet been established in Norway* Should be "this kind of methods has not yet..."

Page 4, line 24 to page 5, line 7:

This summary is not easy to read. A block diagram (or an enhanced version of the Figure 1) would better illustrate the process, especially the seasonal and stochastic loops. The "calibration" steps (1 to 4) should be better distinguished from the "simulation" steps.

In step 10, I think that the steps 5 to 9 (instead of step 6 to 9) are repeated 100 000 times.

Page 5, line 24: Reference/link to NEVINA tool should be put into the "Reference" section.

Page 8, line 14: we have preferred to use the GP due to the inclusion of the shape parameter Two of the seasonal GP distribution for Hørte (SON and JJA) show a bounded asymptotic behaviour. According to the authors, is this an "acceptable" hypothesis for extreme flood estimation? What about coercing the shape parameter to be positive or null only?

Page 9, equation 2:

This equation should rewritten properly: x is a vector of parameters (and consequently to be written with uppercase), with 3 components for G1, and 2 for G2.

Page 9, lines 14-19:

The solution based on copulas could be evoked at the beginning of the section (line 2?) as it introduces the use of the multivariate normal distribution.

Page9, line 29: *was calibrated for the 45 highest flood events* Is it a fixed number for each catchment, whatever may be the length of discharge record?

Page 11, line 6: *More detailed information on the set up is given in Table 3* This detailed information would better fit into the section §2.7. See specific comments on Table 3.

Page 11, line 11: A high sensitivity to the shape of the hyetograph was also found by Alfieri et al. (2008). More details about this reference (context, what kind of catchment, connection to the present work) would be welcome to increase its relevance.

Page 11, line 15: A high sensitivity to the parameters of the rainfall model was also described by Svensson et al. (2013) Same remark as above. Page 11, line 18: using a higher quantile for the threshold leads to selecting fewer events, leading to a higher degree of uncertainty in the GP fit.

It's the same for all the catchments (higher quantile -> less selected events). This doesn't explain why Øvrevatn is more sensible to this.

Page 11, line 20 to 22: Krinsvatn seems more sensible to the parameters of the PQRUT model. Any comment on that?

**Page 11, line 26 to 28: A reason for this [...] and for Horte, it is 16.7mm.**

In this analysis, I guess that when *soil moisture conditions* are evoked, it is in fact the *soil moisture deficit*. Please check the writing, and also the captions of Table 2.

I don't clearly understand what is explained in these sentences. A possibly better explanation for the sensitivity of Øvrevatn to initial soil conditions can be provided by considering the ratio between the maximum soil moisture deficit and the mean precipitation, both provided in the Table 2: for Krinsvatn, Hørte and Øvrevatn, these ratios are 33%, 26% and 65% respectively, meaning that for Øvrevatn, the maximum potential interception of precipitation is relatively two to three times higher than for the other catchments.

Page 11, line 34: *and also a step change in the frequency curve* I don't see to what this "step change" refers in the frequency curves of the Figure 8.

Page 12, line 33: the duration of the storm event was assumed to be the same as that used for the stochastic PQRUT model

I suppose it is the critical duration? If yes, it is worth mentioning it.

Page 13, Equation 5:

The quantile score should be noted QS, as it is in the text above the equation. The index I, going from 1 to the number of years of record should be mentioned under/above the  $\Sigma$ .

Page 13, line 29: the results of test 2 indicate that both the GEV distribution and the stochastic PQRUT provide similar fits to observed quantiles

This not obvious considering the Figure 11, where the stochastic PQRUT show more variant but also higher QS scores than GEV. A simple boxplot of the QS scores would be more informative than the barplot of the Figure 11.

Page 15, line 2: *Due to the considerable seasonal variation in the initial conditions...* And also (and mostly) in the extreme rainfall distribution.

**Pages 17 to 19:**

Some references are incomplete (journal name is missing): Blanchet, Gräler, Nathan, Wilson.

Figure 1:

As suggested in the comments above, a simple, vertical block diagram would better illustrate the process, especially the seasonal and stochastic loops. The "calibration" steps (1 to 4) should be better distinguished from the "simulation" steps.

**Figure 2:**

The "zoom maps" for the three catchments will likely be unreadable in the final document if the size of the figure is reduced. Furthermore, I am not sure they provide an interesting information.

**Figure 4:**

I assume the duration of these precipitations is the critical duration of each catchment. If yes, it deserves to be mentioned, whether in the axis name or in the caption of the figure.

The catchments should be ordered in the same way they are in the rest of the manuscript, i.e. Hørte, Krinsvatn and Øvrevatn.

Figure 5: Units are missing.

**Figure 7:**

The "peak discharge" mention should appear whether in the axis name or in the caption of the figure.

**Figure 8:**

In order to "spare space", the name of the catchments could only be mentioned once in the top of each "column".

Bolder lines could be used, colours are difficult to distinguish on very fine lines.

Legend which are specific to each "setup" should appear within the corresponding box. To this end, a "portrait" setup could be chosen for this page.

**Figure 9:**

I am unable to interpret the left plot. I am not sure that violin plots used in this context are useful. Ten different stochastic hyetographs could be presented instead of it, to illustrate their potential diversity.

**Figure 10:**

Once again, this plot looks odd to me. Why not presenting a simple matrix [catchment x models] with check crosses where the Q100 falls within the confidence interval of the GP?

The reference to the three example catchments should be written in bold, and identified in the caption.

**Figure 11:**

As mentioned above, simple boxplot of the QS scores for both methods would be more informative, and also a scatter plot QS(flood frequency) v/s QS(Stoch. PQRUT). The three example catchments should also be outlined.

**Table 1:**

In the caption of the columns, the units could be written below the caption for a better readability.

**Table 2:**

Column and row caption should be written in bold characters.

The width of the columns could be uniformized.

I am not sure that it is worth providing both mean and median values. I would chose the median.

The P100 fitted by GPD (for the season at risk) would be worth to be added in the table. The critical duration should also be mentioned, at least in the Pmean caption (i.e. Pmean 48h). "Soil moisture" should be replaced by "soil moisture deficit".

Table 3:

Column and row caption should be written in bold characters.

The different setups presented in the table should be organized in the same order than they are in the text and in the Figure 8.

Table 4:

As for Table 3, the different setups presented in the table should be organized in the same order than they are in the text and in the Figure 8, and "blocks" of setups should be named a) b) c) etc. as they are in Figure 8.

The setups should be described in the text, not summarized in the captions here.

The percentage values could be rounded to the next integer.

The 10/100/1000 quantiles could be separated more clearly (bold line) in the Table.

---

## Author Response (AR2)

**Reply to reviewer 1**

General comments: This paper presents a significant enhancement of a Norwegian method for the estimation of extreme floods, based on an event-based rainfall-runoff simulation. It introduces a stochastic process for the assignment of the initial hydrological conditions before the simulated events, as well as for the intensity and the temporal dynamic of the simulated precipitation events. This method is compared to the initial method (which considers only a reference precipitation on given condition), and to a classical FFA. The presented method is interesting, both in terms of methodology and statistical results. It is well explored, with a detailed sensitivity analysis. Most of the issues raised during the review of the initial manuscript have been addressed by the authors, with new and/or complementary information provided, which now allows to better understand the method and the sensitivity analysis. However, in some sections the writing remains perfectible, and some figures would deserve to be improved to ease their reading and their interpretation. I would then recommend a minor revision of this paper, based on the detailed comments/suggestions provided to the authors in that follows. Detailed comments/questions: Quoted sentences are written in italic.

**Page 4, line 4: Australian Rainfall and Runoff 2016 (?) Reference is missing.**

We have now included the missing reference.
**Page 4, lines 4-5: this method has not yet been established in Norway Should be "this kind of methods has not yet..."**
This has now been revised as suggested.
**Page 4, line 24 to page 5, line 7: This summary is not easy to read. A block diagram (or an enhanced version of the Figure 1) would better illustrate the process, especially the seasonal and stochastic loops. The "calibration" steps (1 to 4) should be better distinguished from the "simulation" steps. In step 10, I think that the steps 5 to 9 (instead of step 6 to 9) are repeated 100 000 times.**

We have revised the figure by adding numbers and using different numbers and symbols for the steps.
**Page 5, line 24: Reference/link to NEVINA tool should be put into the "Reference" section.**

The reference has been provided. **Page 8, line 14: we have preferred to use the GP due to the inclusion of the shape parameter Two of the seasonal GP distribution for Hørte (SON and JJA) show a bounded asymptotic behaviour. According to the authors, is this an "acceptable" hypothesis for extreme flood estimation? What about coercing the shape parameter to be positive or null only?**

In the paper we also comment that it is possible to use exponential distribution so that would be an option in these cases.
**Page 9, equation 2: This equation should rewritten properly: x is a vector of parameters (and consequently to be written with uppercase), with 3 components for G1, and 2 for G2.**

We have defined the vector X in the text- we prefer not to rewrite the equation as we have not yet introduced the notation.
**Page 9, lines 14-19: The solution based on copulas could be evoked at the beginning of the section (line 2?) as it introduces the use of the multivariate normal distribution.**

We have just mentioned the copula as one of the alternatives as we don't actually use copulas even though it is true that the multivariate normal can be expressed by using the Gaussian copula distribution.
**Page9, line 29: was calibrated for the 45 highest flood events Is it a fixed number for each catchment, whatever may be the length of discharge record?**

This has been revised in the text.
**Page 11, line 6: More detailed information on the set up is given in Table 3 This detailed information would better fit into the section §2.7. See specific comments on Table 3.**

See answer below.

**Page 11, line 11: A high sensitivity to the shape of the hyetograph was also found by Alfieri et al. (2008). More details about this reference (context, what kind of catchment, connection to the present work) would be welcome to increase its relevance.**

We have added the following sentence: A high sensitivity to the shape of the hyetograph was also found in Alfieri et al. (2008). Their study shows that using rectangular hyetograph results in a significant underestimation of the flood peak while the Chicago hyetograph (e.g. Chow et al., 1988), where the peak is in the middle of the event, resulted in overestimation.

**Page 11, line 15: A high sensitivity to the parameters of the rainfall model was also described by Svensson et al. (2013) Same remark as above.**

We have revised this as: A high sensitivity to the parameters of the rainfall model was also described by Svensson et al. (2016), who tested the sensitivity of a stochastic event based model applied to four small to medium-sized catchments in the UK.

**Page 11, line 18: using a higher quantile for the threshold leads to selecting fewer events, leading to a higher degree of uncertainty in the GP fit. It's the same for all the catchments (higher quantile -> less selected events). This doesn't explain why Øvrevatn is more sensible to this.**

This sentence has now been deleted.

**Page 11, line 20 to 22: Krinsvatn seems more sensible to the parameters of the PQRUT model. Any comment on that?**

In order to check the sensitivity to the parameters of the PQRUT model, we have sampled values from the 5% and 95 % confidence interval of the regional regression equations that can be used to estimate these parameters in ungauged catchments. The performance of these parameters for low probability events is not satisfactory as we can see from fig 8 and the sensitivity is higher. This might be due to the interactions between the parameters -K1,K2 and Trt values as these are predicted independently.

**Page 11, line 26 to 28: A reason for this [...] and for Horte, it is 16.7mm. In this analysis, I guess that when soil moisture conditions are evoked, it is in fact the soil moisture deficit. Please check the writing, and also the captions of Table 2.**

Yes, this is correct -we have fixed this.

**I don't clearly understand what is explained in these sentences. A possibly better explanation for the sensitivity of Øvrevatn to initial soil conditions can be provided by considering the ratio between the maximum soil moisture deficit and the mean precipitation, both provided in the Table 2: for Krinsvatn, Hørte and Øvrevatn, these ratios are 33%, 26% and 65% respectively, meaning that for Øvrevatn, the maximum potential interception of precipitation is relatively two to three times higher than for the other catchments.**

This is maybe due to misunderstanding the reviewer's comments. Here, we have extracted the median soil moisture deficit, which was associated with rainfall events with return periods of 1000 years. It is true, that it makes more sense to present the maximum soil moisture and the mean precipitation. We have now revised this sentence as suggested.

**Page 11, line 34: and also a step change in the frequency curve I don't see to what this "step change" refers in the frequency curves of the Figure 8.**

Agreed, this referred to the fact that there was higher percentage change for the higher return periods, especially for the previous set of results that were based on smaller number of simulations.

**Page 12, line 33: the duration of the storm event was assumed to be the same as that used for the stochastic PQRUT model I suppose it is the critical duration? If yes, it is worth mentioning it.**

Yes, this is correct -we have fixed this.

**Page 13, Equation 5: The quantile score should be noted QS, as it is in the text above the equation. The index I, going from 1 to the number of years of record should be mentioned under/above .**

We have now corrected the equation.

**Page 13, line 29: the results of test 2 indicate that both the GEV distribution and the stochastic PQRUT provide similar fits to observed quantiles This not obvious considering the Figure 11, where the stochastic PQRUT show more variant but also higher QS scores than GEV. A simple boxplot of the QS scores would be more informative than the barplot of the Figure 11.**

We prefer to keep the barplots as they show the performance at each catchment.

**Page 15, line 2: Due to the considerable seasonal variation in the initial conditions… And also (and mostly) in the extreme rainfall distribution.**

That is a good point – we have made this addition.

**Pages 17 to 19: Some references are incomplete (journal name is missing): Blanchet, Graler, Nathan, Wilson.**

Thank you for spotting this – these have now been fixed.

**Figure 1: As suggested in the comments above, a simple, vertical block diagram would better illustrate the process, especially the seasonal and stochastic loops. The "calibration" steps (1 to 4) should be better distinguished from the "simulation" steps. The "zoom maps" for the three catchments will likely be unreadable in the final document if the size of the figure is reduced. Furthermore, I am not sure they provide an interesting information.**

The inset plots have now been removed. **Figure 4: I assume the duration of these precipitations is the critical duration of each catchment. If yes, it deserves to be mentioned, whether in the axis name or in the caption of the figure. The catchments should be ordered in the same way they are in the rest of the manuscript, i.e. Hørte, Krinsvatn and Øvrevatn.**

We have now specified the duration and also rearranged the plot.

**Figure 5: Units are missing.**

We have now specified the units in the figure caption.

**Figure 7: The "peak discharge" mention should appear whether in the axis name or in the caption of the figure.**

We have added this to the figure caption.

**Figure 8: In order to "spare space", the name of the catchments could only be mentioned once in the top of each "column". Bolder lines could be used, colours are difficult to distinguish on very fine lines. Legend which are specific to each "setup" should appear within the corresponding box. To this end, a "portrait" setup could be chosen for this page.**

We prefer to label each figure.

**Figure 9: I am unable to interpret the left plot. I am not sure that violin plots used in this context are useful. Ten different stochastic hyetographs could be presented instead of it, to illustrate their potential diversity.**

We have now updated the plot using lines.

**Figure 10: Once again, this plot looks odd to me. Why not presenting a simple matrix [catchment x models] with check crosses where the Q100 falls within the confidence interval of the GP? The reference to the three example catchments should be written in bold, and identified in the caption.**

We prefer to keep the figure as is- the barplots are easier to read than using crosses.

**Figure 11: As mentioned above, simple boxplot of the QS scores for both methods would be more informative, and also a scatter plot QS(flood frequency) v/s QS(Stoch. PQRUT). The three example catchments should also be outlined.**

See answer above

**Table 1: In the caption of the columns, the units could be written below the caption for a better readability.**

We would rather have the units on the same line in order to save space.

**Table 2: Column and row caption should be written in bold characters. The width of the columns could be uniformized. I am not sure that it is worth providing both mean and median values. I would chose the median. The P100 fitted by GPD (for the season at risk) would be worth to be added in the table. The critical duration should also be mentioned, at least in the Pmean caption (i.e. Pmean 48h). "Soil moisture" should be replaced by "soil moisture deficit".**

**Table 3: Column and row caption should be written in bold characters. The different setups presented in the table should be organized in the same order than they are in the text and in the Figure 8.**

**Table 4: As for Table 3, the different setups presented in the table should be organized in the same order than they are in the text and in the Figure 8, and "blocks" of setups should be named a) b) c) etc. as they are in Figure 8. The setups should be described in the text, not summarized in the captions here. The percentage values could be rounded to the next integer. The 10/100/1000 quantiles could be separated more clearly (bold line) in the Table.**

The tables have been updated.

**Reply to reviewer 2**

**Suggestions for revision or reasons for rejection (will be published if the paper is accepted for final publication) General Comments: The revised manuscript has been improved, especially by the revised presentation of the results and by adding a more detailed description of the proposed procedure itself (including flow chart). The presented results and suggested method is, in my opinion, worth publishing as the proposed method is clearly advantages in contrast to rather subjective single storm event applications. Nevertheless, I still have the feeling that the presentation of the manuscript could be enhanced. I'm not a native speaker myself, however, I think the reading flow could be further improved for example by splitting very long sentences. The representation of variables and units, especially subscript and superscript, should be checked throughout manuscript. Furthermore, the presentation of the study areas should be enhanced and the added flow-char has to be revised, as in my opinion it is one core element of the manuscript. Some specific suggestions are given below. Page and line numbers correspond to the author response version.**

**Specific Comments:**

**P1, l13: "in most places" Do you mean catchments? estimates?**

Yes, this refers to catchments. This has now been corrected.
**P1, l14-15: This sentence is not clear to me. The difference between the PQRUT estimates or the difference to the statistical estimate?**

In this sentence, we refer to: the differences between the stochastic PQRUT and the standard implementation of the PQRUT model Although similarly the differences are also large between the standard implementation of the PQRUT and the flood frequency analysis
**P5, l1-15: Enumeration and corresponding Figure 1. I think the now more detailed description including the Figure helps to understand the procedure. Nevertheless there is room for further improvement. The Figure is a bit chaotic and should be polished. A few suggestions are: • add the numbers of the steps in the manuscript to the corresponding steps in the Figure • use colors of better contrast • use straight arrows were possible • use different representation for different things, such as the processing steps and the labels "each season" and repeat "100 000 times" • it is not obvious what exactly is repeated 100 000 times • maybe add input data for the processing steps**

We have revised the figure by adding numbers and using different numbers and symbols for the steps
**P 6, l5: The web address should be in brackets**

The website is now given in the reference list.
**P6, l12: The numbers should be superscript for units, please check subscript and superscript in the manuscript accordingly.**

The units are now fixed.
**P 9, l10: Phsim, Pdsim, Pi should be identical to the Formula; Please check the consistent presentation of variables and units.**

We have now corrected the notation.
**P10, l12: "amongst other things," among other models? Be more precise or delete the term.**

The PQRUT model is a simple, event-based, 3-parameter model (Fig. 6) which is used for a few different applications, including estimating design floods and safety check floods for dams in Norway (Wilson et al., 2011).
**P14, l18: Delete whitespace between degree and Celsius P15, l8: "the median is approximately" instead of "is around"? P15. l24: I think it should be "which may be either due to" P17, l24: Typo "is" instead of "in" (as is of-**

**ten the case)**

We have fixed these in the new revised version.

 **Figure 1.: See comment on L1-15 P5. Figure 2: The catchment representations, look like google map screenshots. I think it would be an improvement to show the 3 catchments with a rather simplified representation of their river network and topography, including the gauging station. This can easily be done by open source data (e.g. srtm dem etc.) and open source software (e.g. Q-GIS etc.). The catchment should be displayed larger.**

We have now removed the inset maps.

Figure 12: Caption "Pqrut" "PQRUT"

[revised manuscript text omitted]